**Technical Report**

# Intron-encoded cistronic transcripts for minimally invasive monitoring of coding and non-coding RNAs

**Dong-Jiunn Jeffery Truong** [1,2,4], **Niklas Armbrust** [1,2,4], **Julian Geilenkeuser** [1,2], **Eva-Maria Lederer** [1,2], **Tobias Heinrich Santl**[1,2], **Maren Beyer**[1,2], **Sebastian Ittermann**[3], **Emily Steinmaßl**[1,2], **Mariya Dyka**[1,2], **Gerald Raffl** [1,2], **Teeradon Phlairaharn** [1,2], **Tobias Greisle** [3], **Milica Živanić** [1,2], **Markus Grosch** [3], **Micha Drukker** [3] & **Gil Gregor Westmeyer** [1,2] ✉

Despite their fundamental role in assessing (patho)physiological cell states, conventional gene reporters can follow gene expression but leave scars on the proteins or substantially alter the mature messenger RNA. Multi-time-point measurements of non-coding RNAs are currently impossible without modifying their nucleotide sequence, which can alter their native function, half-life and localization. Thus, we developed the intron-encoded scarless programmable extranuclear cistronic transcript (INSPECT) as a minimally invasive transcriptional reporter embedded within an intron of a gene of interest. Post-transcriptional excision of INSPECT results in the mature endogenous RNA without sequence alterations and an additional engineered transcript that leaves the nucleus by hijacking the nuclear export machinery for subsequent translation into a reporter or effector protein. We showcase its use in monitoring interleukin-2 (*IL2*) after T cell activation and tracking the transcriptional dynamics of the long non-coding RNA (lncRNA) *NEAT1* during CRISPR interference-mediated perturbation. INSPECT is a method for monitoring gene transcription without altering the mature lncRNA or messenger RNA of the target of interest.

Many processes dynamically regulate gene expression in vivo, such as the epigenetic state, transcriptional activity and post-transcriptional processes (alternative splicing, the length of the polyA tail and RNA interference).

Less than 2% of the human genome codes for proteins, and high-throughput technologies such as next-generation sequencing have spurred extensive examination of the non-coding genome[1]. Among the various non-protein-coding transcripts, RNAs longer than 200 nucleotides are classified as long non-coding RNAs (lncRNAs).

Many of these transcripts carry out various functions, such as transcriptional regulation, (re)organization of the nuclear architecture, scaffolding or modulation of enzymatic activities[2]. Dysregulation of lncRNAs is involved in various diseases, such as cancer and neurodegenerative diseases[3].

Current genetically encoded reporter systems allow monitoring of the gene expression of selective promoters. However, they all modify the endogenous products either at the protein or mature RNA level unless the entire promoter region coupled to a reporter is duplicated

[1]Institute for Synthetic Biomedicine, Helmholtz Zentrum München, Neuherberg, Germany. [2]Department of Chemistry and TUM School of Medicine, Technical University of Munich, Munich, Germany. [3]Institute of Stem Cell Research, Helmholtz Zentrum München, Neuherberg, Germany. [4]These authors contributed equally: Dong-Jiunn Jeffery Truong, Niklas Armbrust. ✉e-mail: gil.westmeyer@tum.de

into a safe harbour, which takes it out of the context of the endogenous genomic locus[4,5].

While chimeric fusions of a reporter moiety to a protein of interest constitute the most drastic modification of the primary amino acid sequence[6], commonly used 2A skipping peptides (for example, P2A and T2A) also leave scars in the resultant proteins. These alterations may influence the protein's turnover rate due to its modified termini (N-end rule)[7] or its subcellular distribution since many localization sequences are terminally encoded (signal peptides, N-myristoylation, C-prenylation motifs and glycosylphosphatidylinositol anchors).

One way to leave the protein of interest unmodified is to indirectly label it via a nanobody fused to a fluorescent protein. However, only a few nanobodies reliably work when expressed inside cells (intrabodies), and controlling the stoichiometry between the target and intrabody is challenging. Placing an internal ribosome entry site (IRES) into the 3′ untranslated region (UTR) leaves the protein sequence unchanged. However, since the 3′ UTR impacts messenger RNA (mRNA) stability and contains the transcript's zip code, such modifications can be detrimental[8].

As for non-coding genes of interest (GOIs), the most commonly used genetic method is based on RNA aptamers inserted into the UTR of a target mRNA such that an aptamer-binding protein fused to a (fluorescent) reporter can colocalize. However, sufficient labelling usually requires large insertions of up to 24 aptamer repeats which are likely to interfere with the target mRNA localization, half-life and function[9,10]. Thus, aptamers with a decreased affinity for aptamer-binding proteins were developed to minimize alterations to the mRNA half-life[11]. Similarly, deploying programmable RNA-binding proteins (RBPs) such as dCas13 for this task is conceivable but limited by its relatively low affinity to singular target sequences[12,13].

Another approach relies on a multistep process that begins with the release of a single guide RNA (sgRNA) array from the host RNA. The system includes two transfer RNA (tRNA) sequences flanking an sgRNA inserted into the 3′ UTR of a GOI[14]. Upon RNase P- or RNase Z-mediated excision of the sgRNA, a dCas9–sgRNA complex forms. Subsequently, this complex binds to the minimal promoter of an additionally integrated reporter locus and recruits multiple transcriptional transactivator units to trigger the expression of a fluorescent protein downstream of the minimal promoter. However, while the synthetic transcription factor provides amplification, the multi-component process is relatively slow and requires a separate genomic insertion of a bio-orthogonal CRISPR reporter module for each GOI. Furthermore, incorporating the tRNA-flanked sgRNA arrays into the 3′ UTR of a target gene might alter its processing and stability since the resulting host transcript will not have a poly(A) tail[15].

Currently, hybridization methods such as RNA fluorescence in situ hybridization, non-spatially resolved quantitative reverse transcription PCR (RT-qPCR) or RNA sequencing are still the dominant modes for measuring non-coding RNA. However, they are consumptive and thus limited to single-time-point measurements.

Therefore, we set out to develop a minimally invasive reporter system that allows for longitudinal monitoring of the expression of non-coding or coding genes without modifying their protein or mature RNA sequence.

We hypothesized that intronic information could be made actionable in cells by mimicking viral motifs to trigger nuclear RNA export instead of intron degradation. Such an intron-encoded synthetic programmable extranuclear cistronic transcript (INSPECT) could then be translated into an effector protein of choice in a 5′ cap-independent manner or exported out of the cell as an RNA reporter via engineered cell export.

Here, we report on the development, optimization and application of INSPECT for tracking the expression of the cytokine interleukin-2 (IL-2) and the lncRNAs *GUARDIN* and *NEAT1* via minimally invasive, multi-time-point measurements.

## Results

### Development of INSPECT

The core elements of the INSPECT concept comprise an optimized splice donor and acceptor site to ensure fast and efficient excision of the intron-encoded information. In addition, cap-independent translation and cap- and polyA-independent nuclear export elements were also inserted to enable the expression of protein-coding transcripts from an otherwise degraded intron.

First, we generated a reporter system with an exonic firefly luciferase[16] (FLuc) driven by a constitutive *Pgk1* promoter to optimize INSPECT components. Into this surrogate exon, we inserted an artificial intron based on the modified rabbit β-globin (rb*HBB*) intron 1 (ref. [17]) with an embedded NanoLuc luciferase[18] (NLuc) (Fig. 1a,b).

A carboxy (C)-terminal PEST degradation sequence was added to both luciferases (that is, FLuc and NLuc) to ensure a high dynamic range of the reporter system by preventing cytosolic accumulation[19]. After transient transfection, both luciferases were read out independently in a dual-luciferase assay. The FLuc signal indicates correct splicing, whereas its reduction or absence may be caused by cryptic splice sites within the intronic sequence. NLuc complementarily indicates the cumulative efficiency of nuclear export and cap-independent translation of the intron-encoded reporter (Fig. 1a,b).

In our initial experiments, we inserted different RNA elements from murine and primate retroviruses, which have been reported to be responsible for the nuclear–cytosolic export of the viral RNA genome, into the intron downstream of the reporter's coding sequence (CDS)[20,21].

Since it is critical to detect and eliminate missplicing, every insertion of a genetic element into the INSPECT construct was evaluated in terms of splicing efficiency by comparing the FLuc signal relative to the control construct without nuclear export elements. Each experiment also included a FLuc control equipped with a minimal intron without additional elements (Fig. 1a,b).

The reporter system also features an optional Cre recombinase-dependent switch that renders the target gene inoperative (that is, it effectively generates a knockout (KO)) and can be combined with the wide variety of existing Cre recombinase systems[22]. The KO cassette also comprises three inverted triple poly(A) sites (simian virus 40 (SV40) late[23], r*HBB*[24] and a synthetic poly(A) site[25]) flanked by two pairs of heterospecific *loxP* sites[26]. Upon Cre activity, the inverted polyA sequence is flipped together with an inverted upstream splice acceptor into its active sense direction. Similar to gene traps[27], the splice acceptor induces a cryptic splice event by trapping the original splice donor site of INSPECT to ensure the usage of the three poly(A) sites for premature transcriptional termination leading to KO of the host gene (Fig. 1c).

Initial testing of several viral nuclear export elements revealed that two constitutive transport elements (CTEs) from the Mason–Pfizer monkey virus, arranged as a tandem repeat downstream of the splice donor site at the 5′ end of the intronic sequence, enables efficient nuclear export of the intron-encoded NLuc while preserving FLuc expression (Fig. 1d; $P < 0.0001$; one-way analysis of variance (ANOVA) with Bonferroni multiple comparisons test (MCT); full statistical results are given in Supplementary Table 1).

Increasing the number of repeats to four (CTE$_4$) negatively influenced the exonic signal (FLuc) and was thus not investigated further (Fig. 1d; $P < 0.0001$; one-way ANOVA with Bonferroni MCT). In contrast, the same tandem CTE elements (CTE$_2$) negatively influenced the FLuc signal when inserted in the 3′ position (Fig. 1d; $P < 0.0001$; one-way ANOVA with Bonferroni MCT), probably via cryptic splice donor-like motifs within the CTE motifs. Therefore, these were eliminated in CTE* or replaced by a different sequence from another virus strain, including the deletion of cryptic splice donor sites (CTE**) (Fig. 1d)[21].

Tandem insertion of CTE** (CTE$_2$**) was also efficient in driving intron-encoded NLuc–PEST expression while having no detrimental effects on FLuc (Fig. 1d; $P > 0.9999$; one-way ANOVA with Bonferroni MCT).

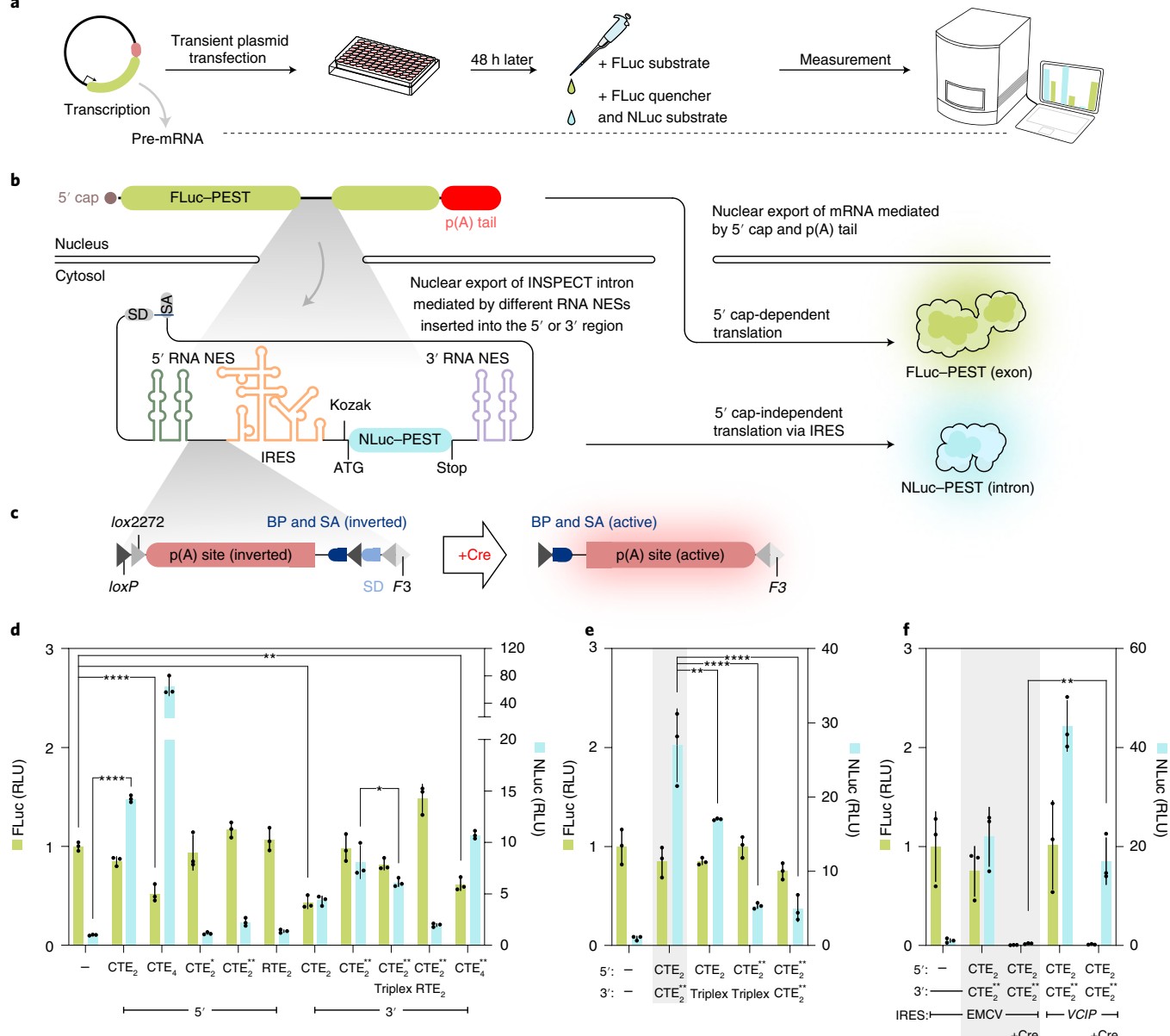

**Fig. 1 | Nested dual-luciferase system for optimizing nuclear export, RNA stability and 5′ cap-independent translation of INSPECT. a**, General workflow for optimization of the INSPECT reporter system via dual-luciferase assay. **b**, The synthetic intron was nested within a FLuc–PEST CDS on a plasmid system driven by the mouse *Pgk1* promoter. Furthermore, the translational unit IRES_NLuc–PEST was inserted into the artificial intron, which is composed of two highly efficient splice sites (splice donor (SD) and splice acceptor (SA)) for the insertion of additional genetic elements for nuclear export and cap-independent translation at the 5′ and 3′ end. Consequently, the NLuc signal represents effective nuclear export and translation of the intron-encoded cistronic transcript, while detection of the FLuc signal indicates correct splicing of the exonic sequence. NES, nuclear export signal; p(A), poly(A). **c**, The reporter system also features a Cre recombinase-inducible KO switch encoded by an inverted triple poly(A) signal flanked by two heterospecific *loxP* pairs. Upon transfection of Cre recombinase, the poly(A) sites are inverted, together with an upstream splice acceptor leading to KO of the gene. BP, branch point; SA, splice acceptor. **d,e**, Results of the dual-luciferase assay schematized in **a**, for different variants of nuclear export and RNA stabilization elements inserted at either the 5′ site or the 3′ site (relative to IRES_NLuc–PEST) as tandem

repeats (the numbers of repeats are indexed as subscripts) (**d**), or at both insertion sites in parallel (**e**). RLU values are normalized to the negative control without nuclear export elements. CTE, from the Mason–Pfizer monkey virus; CTE*, a variant of CTE with cryptic splice donor-like motifs eliminated; CTE**, a variant of CTE with cryptic splice donor-like motifs replaced by a different sequence from another virus strain; RTE, m26 mutant of an RNA transport element with homology to rodent intracisternal A particles; triplex: triple helix-forming RNA from mouse *MALAT1* lncRNA for 3′ end stabilization. **f**, The combination of 5′ CTE$_2$ with 3′ CTE$_2$** was compared in the context of different IRES sequences from either ECMV or the human gene *VCIP*. 'Cre:' indicates the co-transfection of a plasmid expressing Cre recombinase, which recognizes the heterospecific *loxP* and *lox2272* to activate the KO switch (see schematic in **c**). In **e,f**, the grey shading indicates the INSPECT reporter of choice for further experiments. In **d–f**, the bars represent the mean of three biological replicates, with error bars representing s.d. Selected results of Bonferroni MCT after one-way ANOVA analysis are shown (*$P < 0.05$; **$P < 0.01$; ****$P < 0.0001$). For clarity, not all statistical comparisons are graphically presented, but full statistical results are given in Supplementary Table 1.

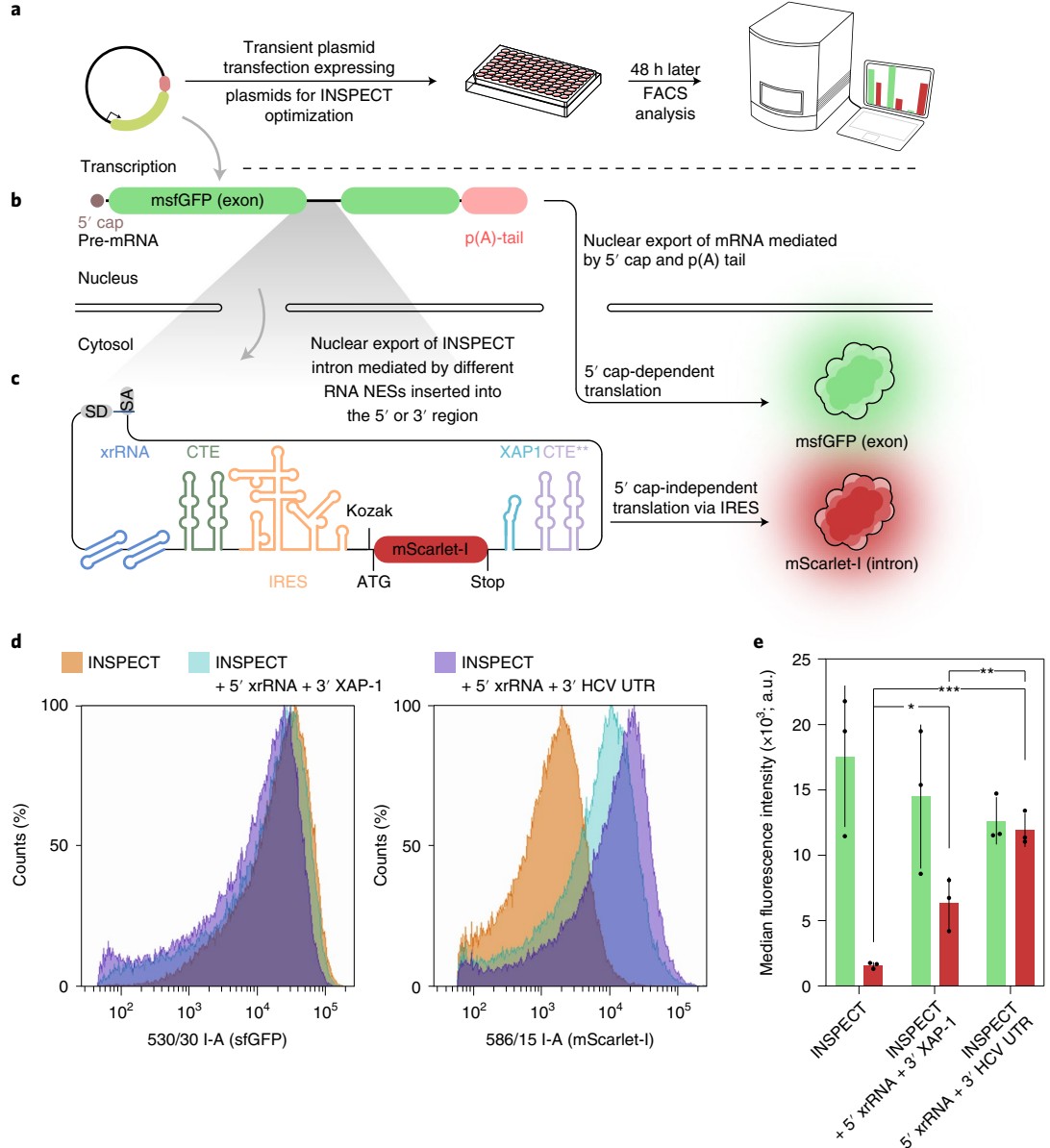

**Fig. 2 | Further optimization of nuclear export, RNA stability and 5′ cap-independent translation of fluorescent INSPECT reporter modules. a**, The INSPECT system was tested by transient transfection of HEK293T cells, followed by FACS analysis 48 h post-transfection. The effect of different genetic elements on the ability to express proteins from an intron (that is, the cumulative effect of nuclear export of the intron and translational efficiency of the intron-encoded protein) was validated by mScarlet-I fluorescence (readout at 586 nm), while detection of msfGFP signal indicated correct splicing of the exonic sequence (readout at 530 nm). **b,** A synthetic intron was nested within an msfGFP CDS (green fluorescence) on a plasmid system driven by the strong mammalian CAG promoter. **c,** In addition, an intron-encoded translational unit, IRES−mScarlet-I (red fluorescence; 3,795 bp from the splice donor to the splice acceptor)

was inserted into the artificial intron already equipped with the INSPECT elements and offers the opportunity to insert additional genetic elements for nuclear export or RNA stability at the 5′ and 3′ end to enhance the INSPECT system. **d**, Representative results of FACS analysis readout at 530 nm (msfGFP; exonic signal; left) and 586 nm (mScarlet-I; intronic signal; right) of HEK293T cells transfected with the indicated INSPECT versions, including additional modifications. **e**, Median fluorescence intensity of cells transfected with the indicated constructs in **d**. Green represents msfGFP (exonic signal) and red represents mScarlet-I (intronic signal). The bars represent the median of three biological replicates with error bars representing s.d. Statistical significance was determined by one-way ANOVA with Bonferroni MCT (*$P < 0.05$; **$P < 0.01$; ***$P < 0.001$). Full statistical results are provided in Supplementary Table 1.

Additional insertion of a 3′ stabilizing element (triple helix)[28,29] reduced NLuc expression (Fig. 1d; $P = 0.0357$; one-way ANOVA with Bonferroni MCT). Similar to the integration at the 5′ end, the insertion of four CTE** (CTE$_4$**) at the 3′ end diminished the FLuc signal (Fig. 1d; $P = 0.0014$; one-way ANOVA with Bonferroni MCT).

Based on the results from the modifications at the 5′ and 3′ end, it was expected and confirmed that combinations of 5′ CTE$_2$ and 3′ CTE$_2$** were strong drivers of intron-encoded NLuc−PEST compared with other combinations (Fig. 1e; $P = 0.0042$, $P < 0.0001$ and $P < 0.0001$; one-way ANOVA with Bonferroni MCT) without detrimental effects on the exonic expression of FLuc−PEST (Fig. 1e; $P = 0.6631$; one-way ANOVA with Bonferroni MCT).

Next, we compared the ability of different IRES elements to drive 5′ cap-independent translation in combination with the tandem CTE nuclear export elements.

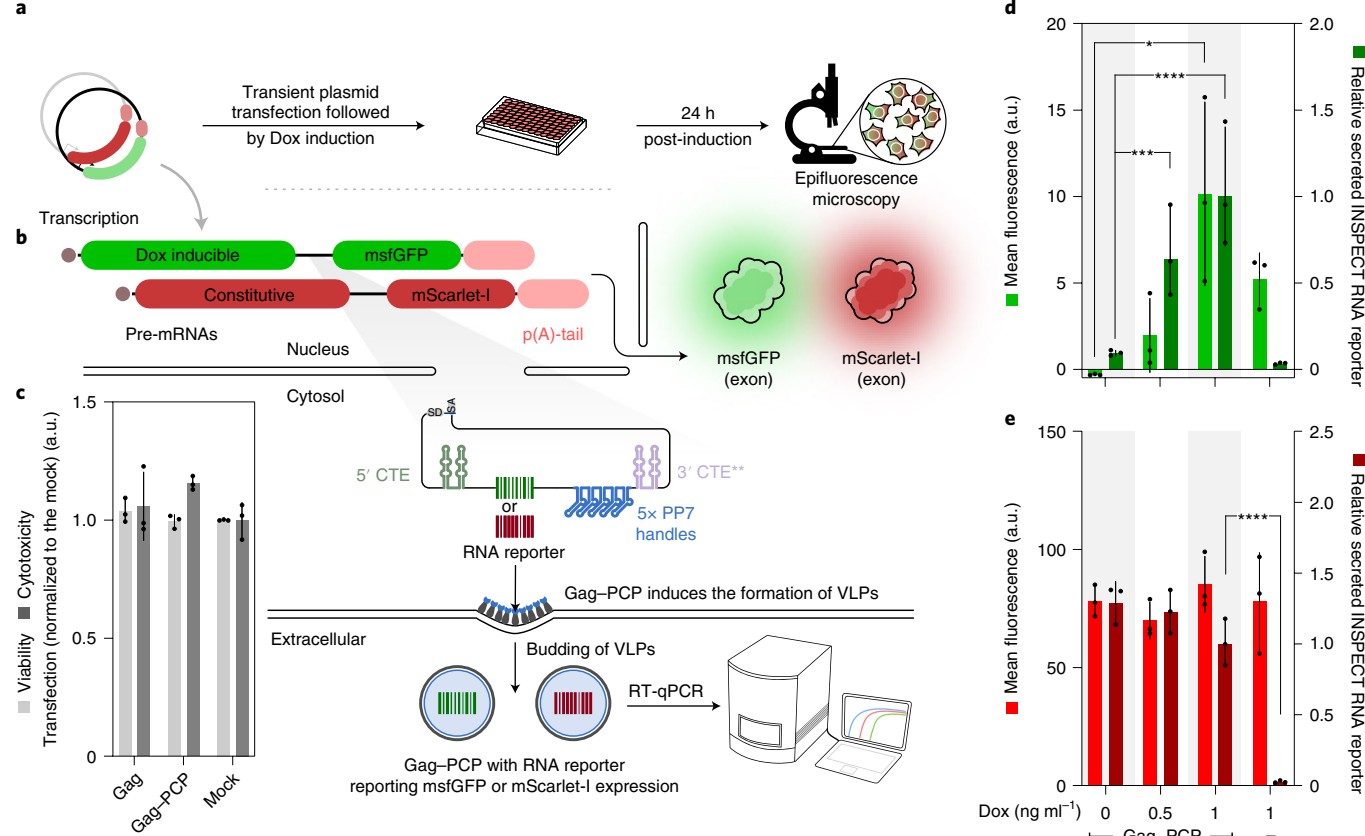

**Fig. 3 | INSPECT reporter enables modular readout of coding genes using RNA reporters. a,** To demonstrate the alternative use of INSPECT transcripts as secreted RNA barcode reporters, a Dox-inducible construct was generated expressing msfGFP together with an INSPECT reporter module. **b,** Instead of the translational unit with an IRES–reporter CDS, INSPECT codes for an RNA reporter followed by a 5× PP7 aptamer motif. As a reference, mScarlet-I was constitutively expressed from a second plasmid together with a second RNA reporter. To secrete PP7-tagged RNA into the extracellular environment via virus-like particle (VLP) budding, a synthetic chimera made of HIV-1 gag polyprotein and PCP was used. **c,** A viability/cytotoxicity assay was performed 48 h after transient transfection with an expression plasmid coding for mouse *Pgk1* promoter-driven HIV-1 gag. The bars represent the mean of *n* = 3 biological replicates with error bars indicating s.d. **d,e,** Quantification of the fluorescence signals of msfGFP (**d**) and mScarlet-I (**e**) and the corresponding INSPECT RNA reporters 48 h post-transfection and 24 h post-induction with the indicated Dox concentrations. The mean fluorescence intensity was corrected accounting for background fluorescence. INSPECT RNA reporter levels were normalized to the maximum induction at 1 ng ml$^{-1}$ Dox. The bars represent the mean of three biological replicates with the error bars representing s.d. Statistical significance was determined by one-way ANOVA with Bonferroni multiple comparisons test (*$P$ < 0.05; ***$P$ < 0.001; ****$P$ < 0.0001). A full statistical analysis is provided in Supplementary Table 1. Fluorescence microscopy images are shown in Supplementary Fig. 1.

Given contradictory reports regarding the efficiency of different IRES variants[19,30,31], we tested IRES sequences from encephalomyocarditis virus (EMCV) or vascular endothelial growth factor and type 1 human collagen-inducible protein (VCIP). Since the IRES elements are located upstream of the corresponding CDS of the INSPECT reporter, it is crucial to exclude cryptic promoter activity, which could lead to false-positive detection of host gene activity. To test for this, we used the Cre-inducible KO switch upstream of the IRES insertion site (Fig. 1c). After Cre-mediated inversion, complete ablation of expression should occur unless cryptic promoter activities within the IRES are present.

Analysis of the reporter system equipped with EMCV-IRES, NLuc, tandem 5′ CTE and tandem 3′ CTE** revealed no cryptic promoter activity or abnormal RNA splicing (FLuc expression) and showed a reliable NLuc signal 20-fold over INSPECT$_{control}$. We also tested the translational efficiency of *VCIP* IRES, which gave an NLuc signal with sufficient FLuc expression. However, Cre-mediated transcription termination upstream of the IRES revealed strong cryptic promoter activity of *VCIP* IRES and thus precluded its application (Fig. 1f; *P* = 0.0038; two-tailed unpaired *t*-test).

In conclusion, EMCV-IRES offered the highest translation rate and was thus chosen, together with tandem wild-type CTE repeats at the 5′ insertion site and a tandem CTE** repeat at the 3′ insertion site, as the bioluminescent INSPECT reporter of choice for subsequent experiments (grey shading in Fig. 1e,f).

## INSPECT reporter modules

In addition to bioluminescent INSPECT readout, we were also interested in establishing fluorescent proteins as INSPECT reporter modules. To optimize INSPECT for these reporters with intrinsically lower signals, we generated a construct that encodes monomeric superfolder GFP (msfGFP) as a surrogate exon, into which an INSPECT module encoding mScarlet-I is inserted (Fig. 2a–c). We found that the addition of an exonuclease-resistant motif at the 5′ end (exoribonuclease-resistant RNA (xrRNA))[32], with either a nuclear export motif (export aptamer (XAP-1))[33] or a translational enhancer motif (3′ UTR of Hepatitis C virus (HCV))[34], could boost the intron-encoded mScarlet-I expression by around five- to tenfold (Fig. 2d,e). At the same time, the exonic msfGFP fluorescence maintained its level of fluorescence, indicating unchanged splicing behaviour.

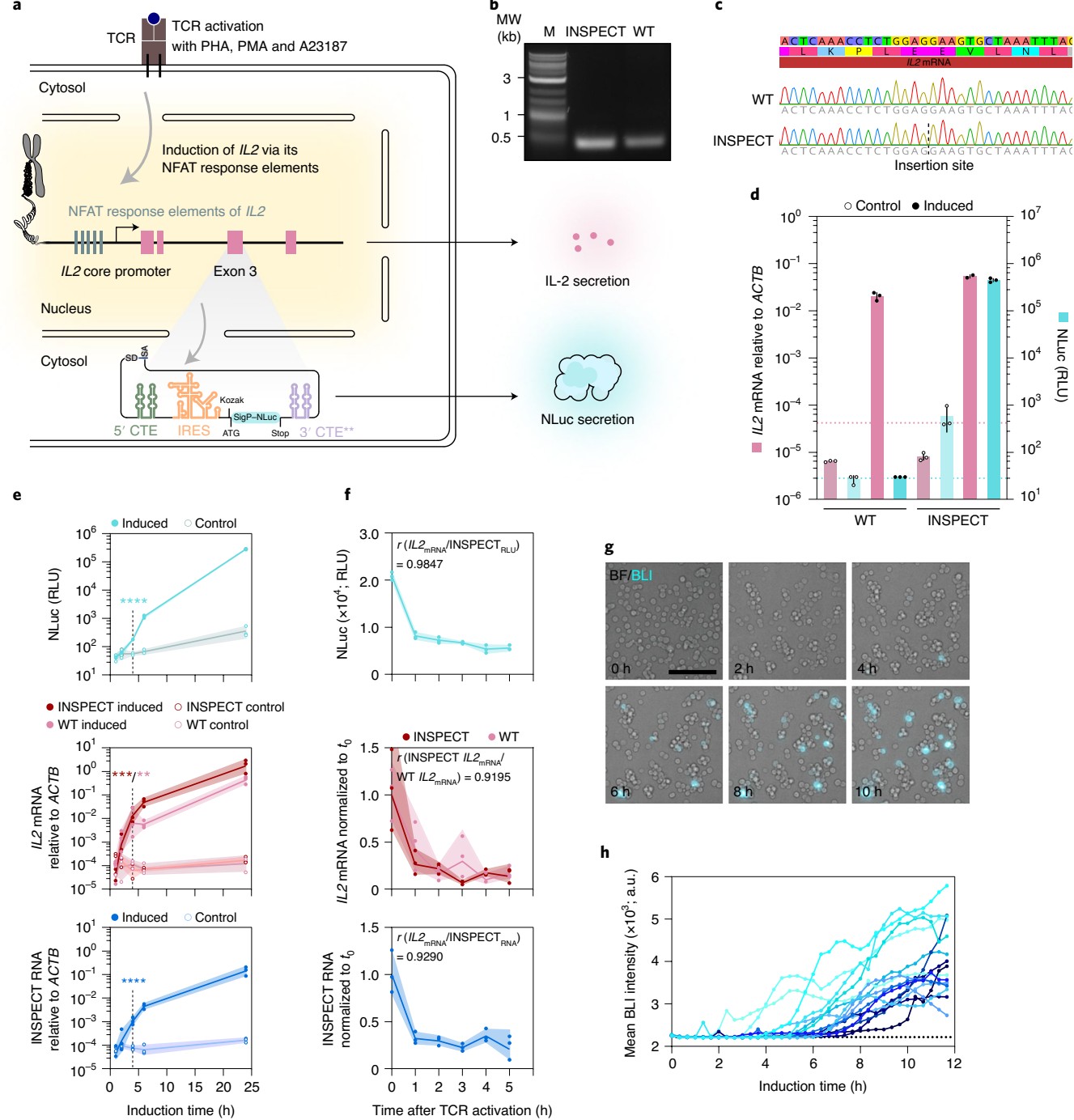

**Fig. 4 | Single-step knock-in of the INSPECT reporter enables minimally invasive monitoring of gene expression. a**, Schematic of TCR stimulation with the tripartite mixture of PHA, PMA and the $Ca^{2+}$ ionophore (Br)-A23187. The subsequent induction of *IL2* was read out via $INSPECT_{SigP-NLuc}$ knocked into exon 3 of the NFAT-controlled *IL2* locus in Jurkat E6.1 cells. **b**, Exon-spanning RT-PCR to indicate the scarless splicing isoform of *IL2* mRNA from wild-type (WT) and $INSPECT_{IL2:SigP-NLuc}$ cells. MW, molecular weight. **c**, Sanger sequencing of the RT-PCR bands of *IL2*. **d**, Relative *IL2* mRNA levels for wild-type and $INSPECT_{IL2:SigP-NLuc}$ Jurkat E6.1 cells 16 h after NFAT induction with 1 ng $ml^{-1}$ PMA, 1 μg $ml^{-1}$ PHA and 0.1 μM (Br)-A23187. Ct values were normalized to *ACTB* mRNA levels. The bars represent three biological replicates (except $n = 2$ for $INSPECT_{IL2:SigP-NLuc}$, induced) with error bars representing s.d. The dotted lines represent the autoluminescence RLU level (turquoise) and the no reverse transcriptase (RT) control (pink). **e**, Top, time courses of the luminescence signal of secreted NLuc from unmodified or $INSPECT_{IL2:SigP-NLuc}$ Jurkat E6.1 cells after TCR activation (0.5 ng $ml^{-1}$ PMA and 0.5 μg $ml^{-1}$ PHA). Corresponding mRNA levels of *IL2* (middle) and INSPECT (bottom) were quantified by RT-qPCR and normalized to the *ACTB* expression

strength. Statistical significance was determined by two-tailed, unpaired *t*-test for the induced versus the control condition after 4 h of induction (**$P < 0.01$; ***$P < 0.001$; ****$P < 0.0001$). **f**, Top, time courses of NLuc luminescence after TCR activation for 6 h (0.5 ng $ml^{-1}$ PMA and 0.5 μM A23187) followed by media exchange containing a cell-permeable NLuc substrate and a cell-impermeable NLuc inhibitor to suppress the extracellular signal. Corresponding mRNA levels of *IL2* (middle) and INSPECT (bottom) are shown relative to *ACTB* and normalized to the time point of media exchange. *r* indicates the Pearson correlation coefficient. The lines in **e** and **f** connect three biological replicates ± s.d. (shading). Full statistical results are provided in Supplementary Table 1. **g**, Bioluminescence microscopy images (BLI; cyan) overlayed on the brightfield channel (BF; grey) acquired after induction with 0.5 ng $ml^{-1}$ PMA, 0.5 μg $ml^{-1}$ PHA and 0.1 μM A23187 from $INSPECT_{IL2:SigP-NLuc}$ Jurkat cells using a long-lasting, cell-permeable NLuc substrate (endurazine) combined with a cell-impermeable inhibitor. Scale bar, 100 μm. **h**, Time courses of luminescence signals from selected cells after bulk TCR activation ($n = 16$ cells that were responsive to NFAT induction). The dotted line indicates the autoluminescence baseline. Full frames are shown in Supplementary Fig. 1.

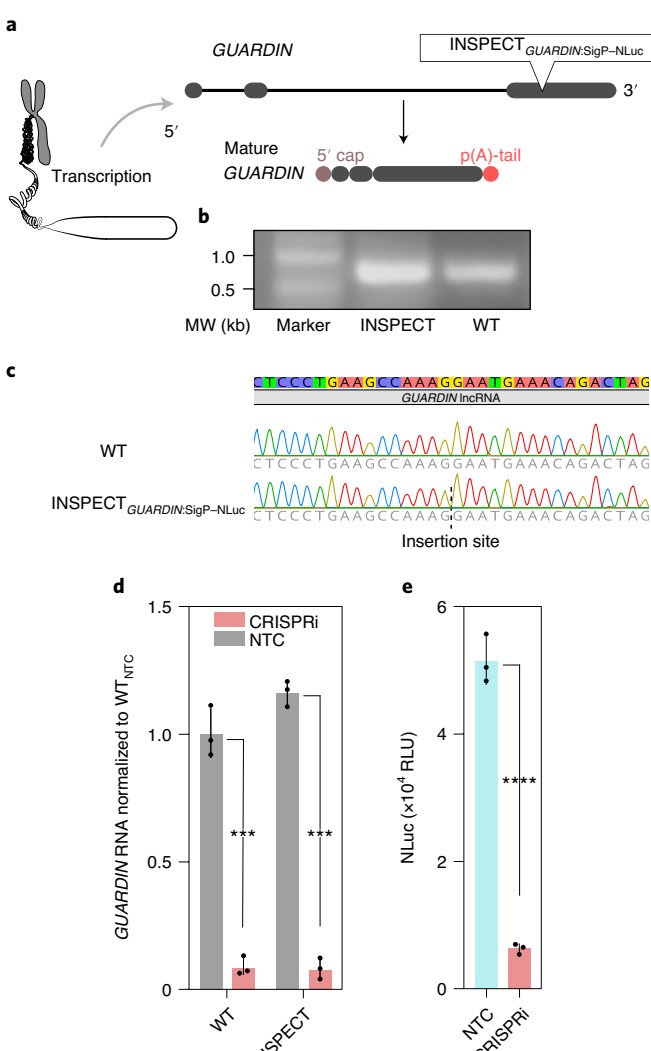

**Fig. 5 | Tagging of the lncRNA *GUARDIN* with the INSPECT reporter system.** **a**, INSPECT$_{SigP–NLuc}$ was inserted into exon 3 of the lncRNA *GUARDIN* via CRISPR–Cas9. **b**, Semi-quantitative RT-PCR analysis reveals the splice isoforms of *GUARDIN* in wild-type and INSPECT$_{GUARDIN:SigP–NLuc}$ HEK293T cells. **c**, Sanger sequencing of the cDNA reveals sequence integrity of the *GUARDIN* locus. **d**,**e**, *GUARDIN* RNA levels and corresponding bioluminescence signals (RLU) of INSPECT$_{GUARDIN:SigP–NLuc}$, measured from the supernatant 72 h after transfection with plasmids for CRISPRi of the *GUARDIN* locus. CRISPRi was achieved via plasmids encoding a dCas9–transcriptional repressor fusion with two sgRNAs against the TSS. The medium was exchanged 24 h before the measurement to reset the secreted luciferase signal. The bars represent the means of three biological replicates with the error bars representing s.d. Selected results of Bonferroni MCT after one-way ANOVA are shown for comparison among the control conditions without CRISPRi to check the effect of the insertion sites. NTC, non-targeting control. The results of two-tailed unpaired *t*-tests are shown for each clone for CRISPRi (****$P < 0.0001$; ***$P < 0.001$). Full statistical results are given in Supplementary Table 1.

To further expand the multiplexing capabilities of INSPECT, we next sought to directly harness the extranuclear intronic RNA as an RNA reporter that can be secreted from the cells for sequence-based readout. We thus generated two constructs with surrogate exons encoding either doxycycline (Dox)-inducible msfGFP or constitutively expressed mScarlet-I (Fig. 3a,b). We encoded a distinct RNA barcode for each condition followed by five PP7 aptamers to provide a high-affinity binding handle for PP7 bacteriophage coat proteins (PCPs).

To secrete the INSPECT RNA barcode reporter, we co-expressed virus-like particles forming chimera (Gag–PCP), derived from human immunodeficiency virus type 1 (HIV-1) gag polyprotein. We inactivated their zinc finger domains, replaced them with PCP and confirmed that their expression had no negative impact on cell viability (Fig. 3c). 24 h post-induction with Dox, msfGFP was expressed in a Dox-dependent manner, while mScarlet-I expression remained relatively stable (Fig. 3d). Analysis of the supernatants from the same cells via RT-qPCR showed the corresponding Dox-dependent increase in transcript levels for the specific INSPECT RNA barcodes. Expectedly, omitting Gag–PCP did not yield any INSPECT transcripts in the supernatant (Fig. 3e).

## Monitoring T cell receptor-driven cytokine *IL2* expression

To demonstrate the advantages of INSPECT over conventional gene reporters that modify the protein and/or the mRNA sequences (Extended Data Fig. 1a), we first focused on the cytokine IL-2, which is strongly induced by activation of the T cell receptor (TCR) mediated by the family of calcium-responsive nuclear factor of activated T cells (NFAT) transcription factors. This signal transduction allows T cells to rapidly express high levels of effector cytokines in response to antigen stimulation, thereby promoting the differentiation of CD4⁺ T cells into defined effector T cells[35,36]. Furthermore, IL-2 is a crucial secreted cytokine for regulatory T cell survival and function[37].

We inserted the optimized INSPECT reporter system, equipped with nuclear export elements, EMCV-IRES and a secreted NLuc reporter, into *IL2* exon 3 (INSPECT$_{IL2:SigP–NLuc}$) of Jurkat E6.1 T lymphocytes (Fig. 4a).

First, we showed that INSPECT$_{IL2:SigP–NLuc}$ was correctly spliced out of its host transcript *IL2* after pharmacological TCR activation (with phytohaemagglutinin (PHA) and phorbol 12-myristate 13-acetate (PMA)) by RT-PCR (Fig. 4b) and sequencing to confirm that the resulting mRNA contained the correct sequence (Fig. 4c).

Furthermore, the wild-type Jurkat E6.1 cells and INSPECT$_{IL2:SigP–NLuc}$ reporter line both showed strong induction of *IL2* mRNA over three orders of magnitude, which was tracked well by the secreted NLuc reporter signal (Fig. 4d). Similarly, the accumulation of IL-2 protein in the supernatant after induction yielded $39.8 \pm 0.5$ ng ml$^{-1}$ for wild-type cells and $67 \pm 1.8$ ng ml$^{-1}$ for INSPECT$_{IL2:SigP–NLuc}$; Supplementary Table 1), while baseline protein concentrations were below the detection limit of the enzyme-linked immunosorbent assay (ELISA).

Next, we set out to monitor the TCR-driven (0.5 ng ml$^{-1}$ PMA and 0.5 µg ml$^{-1}$ PHA) increase in *IL2* expression over a period of 24 h. After 4 h of TCR induction, we could reliably detect accumulation of the secreted NLuc reporter by repeatedly sampling the supernatant (two-tailed unpaired *t*-test; $P < 0.0001$), which correlated well with *IL2* mRNA levels of INSPECT$_{IL2:SigP–NLuc}$ ($P = 0.0008$; two-tailed unpaired *t*-test) and unmodified Jurkat E6.1 cells ($P = 0.0091$; two-tailed unpaired *t*-test; Fig. 4e).

To follow the decrease of *IL2* expression levels after T cell activation, we stimulated TCRs for 6 h (0.5 ng ml$^{-1}$ PMA and 0.5 µM A23187). We then changed to media containing a cell-permeable NLuc substrate in combination with a cell-impermeable NLuc inhibitor, such that the luminescence signal selectively reflected the levels of endoplasmic reticulum/Golgi-resident intracellular NLuc during its secretion process while extracellular NLuc enzymes were inhibited.

The resulting NLuc signal dropped rapidly within 1 h in line with *IL2* mRNA levels in unmodified Jurkat E6.1 cells (Pearson's $r = 0.985$; $P = 0.0003$; Fig. 4f). The same protocol also allowed us to obtain single-cell resolution in bioluminescence microscopy (Fig. 4g), which showed that *IL2* expression could be detected as early as 4 h post-TCR activation with a substantial variability up to response times of ~8 h across the population (Fig. 4h).

To show that INSPECT is compatible with a range of reporter modules, we replaced NLuc with the complex multipass transmembrane sodium iodide symporter (NIS) (SLC5A5; 1,932 bp), which is selectively expressed in thyroid gland cells but can be ectopically expressed to

accumulate the radioisotope $^{131}I^-$ that can be detected using gamma counters or SPECT[38-40] (Extended Data Fig. 2a–c).

A linear relationship was observed between the counts per minute and the IL-2 protein levels in the supernatant of three independent INSPECT$_{IL2:NIS}$ Jurkat E6.1 cell clones, as determined by ELISA before and after TCR activation. Unmodified Jurkat E6.1 cells showed comparable IL-2 levels in the supernatant after induction, while the counts per minute remained at baseline (Extended Data Fig. 2d).

## INSPECT reporter for the lncRNA GUARDIN

Next, we assessed the ability of INSPECT to monitor lncRNAs, as they can currently only be measured via consumptive staining or sequence-based transcript analysis. As a first-target lncRNA, we chose GUARDIN, which plays a role in maintaining genome integrity[41] and exhibits low expression levels (~13% of lncRNA NEAT1_v2 and ~6% of total NEAT1 expression levels; Extended Data Fig. 3) in HEK293T cells (Fig. 5a-c).

The secreted NLuc signal after CRISPR interference (CRISPRi) targeting still gave a signal intensity of two magnitudes over the background (Fig. 5d,e), indicating INSPECT's high sensitivity for transcripts with weak expression.

## Minimally invasive monitoring of the structural lncRNA NEAT1

To challenge the INSPECT system even more with a complex architectural lncRNA, we next selected NEAT1 as a target lncRNA of interest since it is organized as subnuclear bodies in the interchromatin space of mammalian cells close to nuclear speckles. Since these so-called paraspeckles can be structurally analysed in detail by single-molecule FISH (smFISH), we reasoned that any perturbation from INSPECT insertion could be assessed with high sensitivity. NEAT1 was shown to have relevance for modulating replication, the stress response, tumour formation or human embryonic stem cell (hESC) differentiation[42,43]. NEAT1 is transcribed into two distinct isoforms (short NEAT1_v1 and long NEAT1_v2) via alternative 3′ processing[44,45]. However, only the long NEAT1_v2 isoform is crucial for paraspeckle formation. It binds to various RBPs (for example, the Drosophila behaviour human splicing family NONO, SFPQ, FUS and TDP-43 (ref. [46])).

We thus inserted two INSPECT$_{SigP-NLuc}$ modules into different insertion sites of the NEAT1 locus, such that the first insertion site (INSPECT$_{NEAT1\_total:SigP-NLuc}$) allows monitoring of both NEAT1 isoforms in parallel (that is, v1 and v2), while the second exclusively reports NEAT1_v2 (INSPECT$_{NEAT1\_v2:SigP-NLuc}$) (Fig. 6a,b).

The impact of INSPECT's insertion on the paraspeckle assembly was analysed via smFISH, with established probes against the NEAT1 5′ segment to quantify the paraspeckles in cell lines harbouring the different insertion sites compared with the parental wild-type control (Fig. 6c). The fraction of cells containing paraspeckles and the

number of paraspeckles per cell in the two INSPECT$_{NEAT1}$ lines were indistinguishable from the wild type (Fig. 6d,e). We also verified that the nuclear abundance of NEAT1 lncRNA and the cytosolic/nuclear ratio was non-significantly different between wild-type and INSPECT$_{NEAT1}$ cells (Fig. 6f,g, $P > 0.6987$, one-way ANOVA with Bonferroni MCT). Furthermore, RT-PCR and Sanger sequencing of the junction revealed no sequence change for both insertion sites (Extended Data Fig. 4a,b).

In comparison, an alternative INSPECT insertion site for NEAT1_v2 showed some alteration in the number of paraspeckles per cell and the abundance of NEAT1 in the nuclear fraction; therefore it was not chosen for subsequent experiments (Extended Data Fig. 5).

The INSPECT system also offers the possibility to render the target gene inoperative via a KO switch. Cells with Cre recombinase-activated KO switch in NEAT1 revealed the near-complete absence of positive smFISH stainings, indicating successful KO of the NEAT1 locus (Fig. 1c and Extended Data Fig. 6).

Next, we modulated NEAT1 promoter activity by CRISPR interference (CRISPRi)[47] to show that NLuc activity reflects NEAT1 promoter activity, as intended. For this purpose, we fused the Krüppel-associated box (KRAB) transcriptional repression domain from the ZIM3 protein[48] to catalytically inactive dCas9 (D10A and H840A)[49]. INSPECT$_{NEAT1\_total:SigP-NLuc}$ HEK293T and INSPECT$_{NEAT1\_v2:SigP-NLuc}$ HEK293T cells were transfected with the ZIM3-KRAB–dCas9 fusion and a mixture of sgRNAs targeting the upstream region close to the NEAT1 transcription start site (TSS). Supernatant NLuc activity was significantly decreased compared with a mix of control sgRNAs targeting an unrelated MAPT promoter (Fig. 6h,i). We created an additional INSPECT$_{NEAT1\_v2:NLuc}$ cell line lacking the signal peptide (SigP) to prevent NLuc secretion, such that NEAT1_v2 expression could be monitored with single-cell resolution by bioluminescence microscopy without the necessity for an extracellular NLuc inhibitor. After live cell imaging, we fixed the cells and confirmed that cells with higher bioluminescence signals also contained more paraspeckles via smFISH (Fig. 6j).

We subsequently knocked in INSPECT$_{NEAT1\_v2:SigP-NLuc}$ into H9 hESCs in which NEAT1 is induced upon differentiation[43,50]. We confirmed that the H9 cells carrying INSPECT$_{NEAT1\_v2:SigP-NLuc}$ showed all specific lineage markers for the three germlines, similar to unmodified H9 cells (Extended Data Fig. 7a,b).

Compared with their undifferentiated state, NEAT1 was induced most strongly in cells during differentiation to mesodermal fate, followed by endo- and ectodermal fate (Extended Data Fig. 7c; $P = 0.0052$, $P < 0.0001$ and $P < 0.0001$; one-way ANOVA with Bonferroni MCT) in line with NEAT1 levels measured by RT-qPCR (Extended Data Fig. 7d; $P = 0.0039$ and $P < 0.0001$; one-way ANOVA with Bonferroni MCT).

These results jointly demonstrate the capability of INSPECT to report the transcriptional activity of non-coding genes in a minimally

**Fig. 6 | INSPECT enables minimally invasive monitoring of the expression of the lncRNA NEAT1. a**, INSPECT was inserted via CRISPR–Cas9 into different sites of the lncRNA NEAT1. The lncRNA NEAT1 is transcribed into a short (NEAT1_v1) and a long RNA isoform (NEAT1_v2), where the latter is essential for the formation of paraspeckles containing several RBPs. The 5′ insertion (INSPECT$_{NEAT1\_total:SigP-NLuc}$) is present in both isoforms, while the second insertion after the alternative poly(A) site (INSPECT$_{NEAT1\_v2:SigP-NLuc}$) exclusively reports long isoform expression. **b,** Elements of INSPECT coding for secreted or cytosolic NLuc. **c**, Top, representative images of the DAPI-stained and smFISH-probed (against NEAT1) channel for each insertion site, shown as a merged overlay. Bottom, signals (ROIs) from the images above that were identified as paraspeckles (+) inside the nucleus (encircled) for automated quantification. Scale bar, 10 μm. NUC, nucleus. **d,e**, Percentage of cells with paraspeckles (**d**) and paraspeckle count per cell (**e**; red bars represent the median of the distribution) in selected clonal cell lines containing INSPECT$_{SigP-NLuc}$ at the indicated insertion sites (n indicates the number of cells; see **c** for representative images). A full statistical analysis is provided in Supplementary Table 1. **f**, Comparison of NEAT1

abundance relative to MALAT1 in the nucleus after INSPECT insertion at different insertion sites of NEAT1. **g**, Cytosolic/nuclear ratio of NEAT1 RNA. The bars in **f** and **g** represent the geometric mean ± geometric s.d. Full statistical analysis is provided in Supplementary Table 1. **h**, RLUs obtained from the supernatant of INSPECT$_{SigP-NLuc}$ cells with the indicated insertion sites 72 h after transfection with CRISPRi plasmids against NEAT1. The CRISPRi plasmid mixture encodes a dCas9–transcriptional repressor fusion chimera with three sgRNAs against the NEAT1 TSS (promoter). The medium was exchanged 24 h before measurement to reset the secreted luciferase signal. **i**, Fold-change of NEAT1 expression in wild-type HEK293T cells after CRISPRi perturbation. The bars in **h** and **i** represent three biological replicates with the error bar representing s.d. (***$P < 0.001$ and ****$P < 0.0001$; two-tailed unpaired t-test). Full statistical results are provided in Supplementary Table 1. **j**, Left, detection of the NLuc signal of INSPECT$_{NEAT1\_v2:NLuc}$ with single-cell resolution by bioluminescence microscopy. The respective images were obtained 16 h after seeding. Right, after imaging, the cells underwent fixation and smFISH against NEAT1 to detect paraspeckles. Scale bar, 10 μm. Uncropped images are shown in Supplementary Fig. 1.

invasive manner without interfering with their native sequence or structure.

## Discussion

In this work, we show that an intron can be repurposed as a synthetic host-promoter-dependent cistron by exploiting natural

mechanisms to export foreign nucleic acids from the host's nucleus, stabilize unprotected RNAs and enhance translational efficiency without introducing scars on the transcriptional or translational level.

We systematically tested permutations and combinations of the respective functional elements and identified an optimized variant

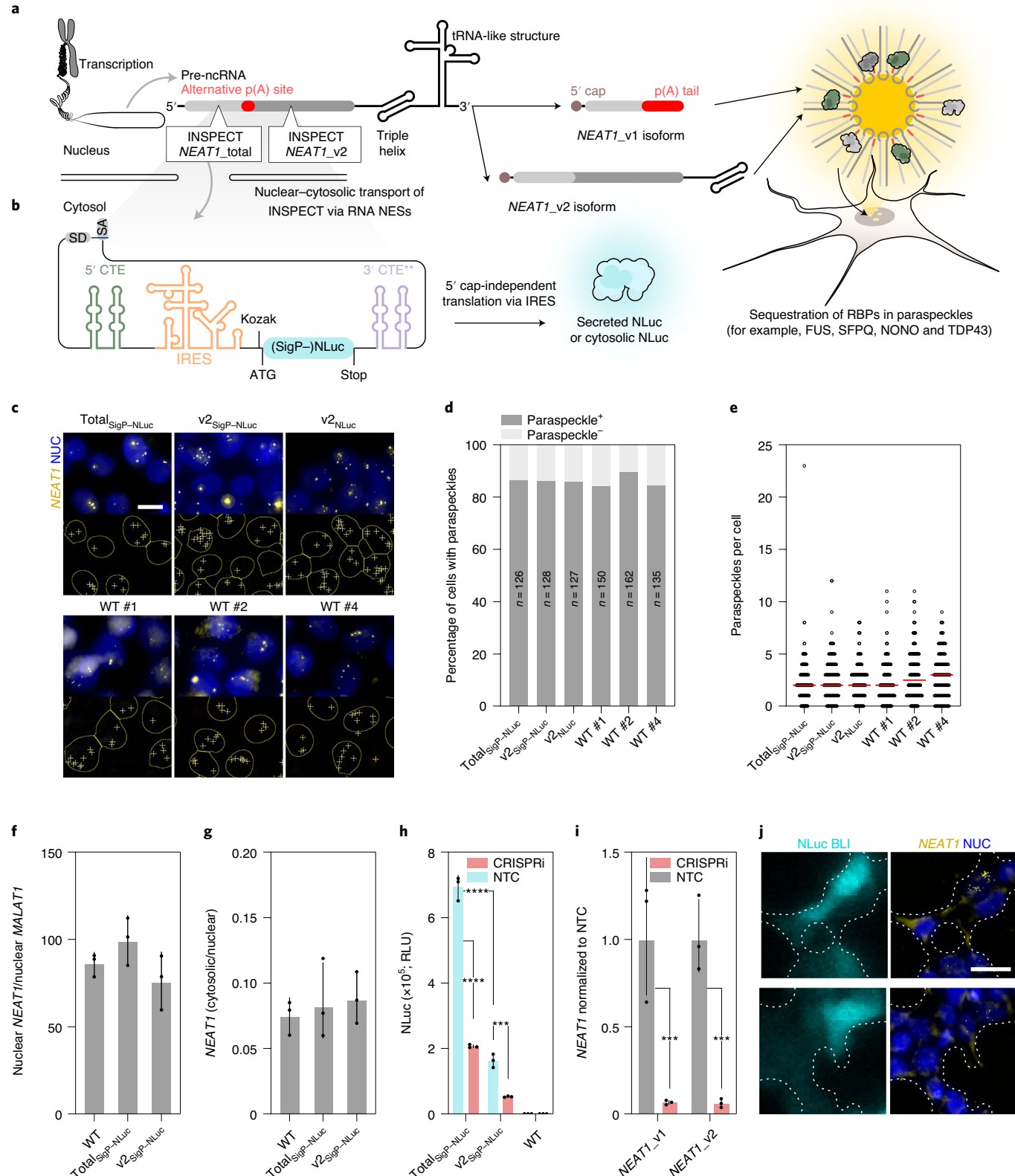

with very efficient intron-encoded reporter expression depending on the host gene's promoter activity.

We showed that INSPECT can be used to track gene expression changes of both coding and non-coding genes in a minimally invasive fashion.

INSPECT monitored the expression of the lncRNAs *NEAT1* and *GUARDIN* with high sensitivity and was capable of detecting low basal levels and tracking expression changes over time.

Compared with consumptive sequence analyses or staining techniques, INSPECT can allow for multi-time-point and multi-well assessment of modulators of lncRNA expression for correlation with resulting phenotypes[51]. While INSPECT is currently not designed to interact directly with lncRNA molecules to, for example, localize them, INSPECT could also be used to intron encode complex effectors for regulating cell function via guide RNAs or the expression of effector proteins of choice, including selection mechanisms.

INSPECT can also aid in monitoring coding genes without altering the mature mRNA sequence and thus the protein sequence. This feature may be of particular advantage for valuable cell systems in which safety concerns are a priority, such as therapeutic immune cells. For example, INSPECT could monitor the *IL2* locus for temporospatial mapping of engineered T cell activation, ideally with a highly sensitive method such as positron emission tomography or single-photon-emission computed tomography and without introducing any immunogenic protein product, as in the example of the NIS reporter.

Analysis of secreted INSPECT transcripts by RT-qPCR as opposed to a reporter protein signal can enable multiplexing capabilities of INSPECT. By integrating RNA barcodes for molecular timestamps, one could also imagine time-resolved transcript monitoring of GOIs[52].

Furthermore, the Cre-dependent KO switch integrated into INSPECT can complement promoter-dependent interrogation with tissue-specific or developmental-stage-dependent KOs of coding and non-coding genes.

To facilitate the application of INSPECT to other GOIs, we provide optimized protocols that allow for the insertion of INSPECT reporter modules and establishing respective monoclonal reporter cell lines within ~8 weeks. We recommend testing several insertion sites to ensure that no context-dependent cryptic splicing occurs, as was observed in one alternative insertion site in *NEAT1* (Extended Data Fig. 5).

INSPECT is a generalizable and modular method to render intron-encoded genetic information actionable in cells via synthetic cistronic transcripts, overriding lariat degradation and hijacking the cellular processes for nuclear export, cap-independent translation and optional secretion from the cell.

We thus anticipate versatile applications for INSPECT in minimally invasive studies of (non-coding) gene expression and mammalian cell engineering.

## Online content

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

## Methods

Please note that the general cloning procedures for generating INSPECT reporter cell lines are similar to our protocols published for EXSISERS[53]. A detailed exemplary protocol can be found on Protocol Exchange[54].

### Molecular cloning

**Bacterial strains.** Chemically competent *Escherichia coli* cells (NEB Stable) were used for the transformation with plasmid DNA. Carbenicillin (Carl Roth) was used at a final concentration of 100 μg ml⁻¹ for antibiotic selection. All bacterial cells were grown in lysogeny broth medium or agar plates, including the respective antibiotics.

**Bacterial transformation.** Chemical transformations were performed by adding 1–3 μl DNA from Gibson Assembly or ligation reaction to thawed, ice-cold, chemically competent cells (50 μl). After incubation on ice for 30 min, heat shock was performed at 42 °C for 30 s, followed by a 5-min incubation step on ice before finally adding 450 μl of room temperature SOC Medium (NEB). Cells were then plated on agar plates containing the appropriate types and concentrations of antibiotics and were incubated overnight at 37 °C.

**DNA agarose gel electrophoresis.** DNA fragments were separated by electrophoresis at 110 V for 30–50 min on a TAE-buffered 1% (wt/vol) agarose gel (Agarose Standard; Carl Roth) with 1:10,000 SYBR Safe stain (Thermo Fisher Scientific). Before loading, samples were mixed with Gel Loading Dye (Purple, 6X; NEB). Plus DNA Ladder (1 kb; NEB) was loaded in parallel for size determination.

**DNA digestion with restriction endonucleases.** Plasmid DNA was digested with restriction enzymes (NEB) according to the manufacturer's protocol in a reaction volume of 40–50 μl containing 1–3 μg plasmid DNA. Digested plasmid DNA fragments were separated by size via agarose gel electrophoresis and subsequently purified using a Monarch DNA Gel Extraction Kit (NEB).

**Gibson assembly and ligation reactions.** Concentrations of DNA fragments were measured with a spectrophotometer (NanoDrop 1000; Thermo Fisher Scientific). Ligations were performed with 30–100 ng plasmid backbone DNA (an ori-containing DNA fragment) in a 20 μl reaction volume at room temperature, with 1:1–3 backbone:insert molar ratios, using T4 DNA Ligase (Quick Ligation Kit; NEB) for 5–10 min. Isothermal Gibson assemblies were performed in a 15 μl reaction volume with 1:1–1:5 backbone:insert molar ratios and 50–75 ng backbone DNA, using NEBuilder HiFi DNA Assembly Master Mix (2×; NEB) for 20–60 min at 50 °C.

**PCR.** Oligodeoxyribonucleotides (single-stranded DNA primer; IDT) were resolubilized in nuclease-free water to 100 μM. PCR amplification from plasmid or genomic DNA was performed with Platinum SuperFi II PCR Master Mix (Thermo Fisher Scientific), Q5 Hot Start High-Fidelity 2X Master Mix or 5× High-Fidelity DNA Polymerase and 5× GC Enhancer (NEB) according to the manufacturer's protocol. After PCR, samples were gel purified from gel electrophoresis using a Monarch DNA Gel Extraction Kit (NEB).

For downstream T7 Endonuclease I assays, PCR was performed using LongAmp Hot Start Taq 2X Master Mix (NEB). Primers are listed in Supplementary Table 2.

**Plasmid DNA purification and Sanger sequencing.** Transformed *E. coli* clones were picked from agar plates using sterile pipette tips, inoculated in 2 ml SOC Medium with appropriate antibiotics in a ventilated falcon tube and incubated at 37 °C for 6–8 h. For sequencing or molecular cloning, plasmid DNA was extracted using the Monarch Plasmid Miniprep Kit (NEB) according to the manufacturer's protocol. Correct clones were inoculated in 100 ml lysogeny broth medium

containing proper antibiotics overnight at 37 °C. Plasmid DNA was purified with a Plasmid Maxi Kit (QIAGEN).

Plasmid DNA was sent for Sanger sequencing (GENEWIZ) and analysed by Geneious Prime (Biomatters) sequence alignments.

### Mammalian cell culture

**Cell lines and cultivation.** Cells were cultured at 37 °C under an atmosphere of 5% $CO_2$ and saturated $H_2O$.

HEK293T cells (ECACC 12022001; Sigma–Aldrich) were cultured in advanced Dulbecco's modified Eagle medium (Gibco; Thermo Fisher Scientific) containing 10% foetal bovine serum (Gibco; Thermo Fisher Scientific), GlutaMAX (Gibco; Thermo Fisher Scientific), 100 μg ml⁻¹ Penicillin–Streptomycin (Gibco; Thermo Fisher Scientific), 10 μg ml⁻¹ Piperacillin (Sigma–Aldrich) and 10 μg ml⁻¹ Ciprofloxacin (Sigma–Aldrich). Cells were split at 90% confluence by medium removal, a washing step with Dulbecco's phosphate-buffered saline (DPBS; Gibco; Thermo Fisher Scientific) and cell detachment for 5–10 min with room temperature Accutase solution (Gibco; Thermo Fisher Scientific). Accutase was subsequently inactivated by adding a 300% excess of pre-warmed culture medium. Cells were then transferred into a new T25/T75 flask at an appropriate density or counted and plated in a 96- or 6-well format for plasmid transfection.

Jurkat E6.1 cells (ECACC 88042803) were maintained in RPMI 1640 media containing glutamine (Gibco; Thermo Fisher Scientific) supplemented with 10% foetal bovine serum (Gibco; Thermo Fisher Scientific) and 100 μg ml⁻¹ Penicillin–Streptomycin (Gibco; Thermo Fisher Scientific). The cells were passaged at a density of ~900,000 cells per ml⁻¹ by removing 90% of the cell suspension and adding pre-warmed RPMI 1640 media.

Human embryonic stem cells (H9; WiCell Research Institute) were routinely cultured in StemMACS iPS-Brew XF (Miltenyi Biotec) and passaged using StemMACS Passaging Solution (Miltenyi Biotec) on tissue culture-treated plates (Sigma–Aldrich) coated with Matrigel (Thermo Fisher Scientific) diluted 1:100 in Dulbecco's modified Eagle medium/F-12 (Thermo Fisher Scientific).

**Activation of IL-2 expression in Jurkat E6.1 cells.** A total of 500,000 cells were centrifuged (3 min; 300 RCF) and resuspended in 2 ml RPMI media supplemented with 1 ng ml⁻¹ PMA and 1 μg ml⁻¹ PHA, if not otherwise indicated. Induction stress could be reduced by using 0.5 ng ml⁻¹ PMA and 0.5 μg ml⁻¹ PHA. The NFAT response could be accelerated by adding 0.1 μM calcium ionophore (Br)-A23187, leading to rapid *IL2* expression.

**Cell fractionation.** For separation of the cytoplasmic and nuclear RNA, cells were fractionated. After removal of the cultivation media, cells were rinsed with DPBS (Gibco; Thermo Fisher Scientific) and incubated in Accutase solution (Gibco; Thermo Fisher Scientific) for 10 min. The detached cells were collected from the T25 flask (1,000,000 cells were seeded 24 h before collection) and centrifuged for 3 min at 300 RCF and 4 °C. Cells were resuspended in 750 μl DPBS (Gibco; Thermo Fisher Scientific) and centrifuged again (300 RCF at 4 °C). All further manipulations with the samples were performed on ice. Cell pellets were resuspended in 750 μl ice-cold mild lysis buffer (50 mM Tris, 100 mM NaCl, 5 mM $MgCl_2$ and 0.5% Nonidet P-40, pH 8) and incubated for 5 min to allow lysis of the cell membrane only.

The lysate was centrifuged for 2 min (18,000 RCF at 4 °C) and the supernatant was carefully transferred to the fresh single-packaged Eppendorf reaction tube. The pelleted nuclei were resuspended in 750 μl mild lysis buffer. Subsequently, 1.5 μl proteinase K (20 mg ml⁻¹; Thermo Fisher Scientific) and 4 μl 40% sodium dodecyl sulfate solution were added to both fractions, which were vortexed vigorously before incubation for 15 min at 37 °C.

Finally, the fractions were centrifuged at 18,000 RCF for 2 min to clear impurities from the samples. RNA isolation was carried out as

described in the RNA analysis section using the NEB Monarch Total RNA Miniprep Kit according to the protocol.

**Differentiation of H9 embryonic stem cells.** Wild-type H9 stem cells and H9 stem cells carrying the INSPECT reporter system at the *NEAT1* locus (INSPECT$_{NEAT1\_v2:SigP-NLuc}$) were differentiated into ectoderm, mesoderm and endoderm using the STEMdiff Trilineage Differentiation Kit (Stem Cell Technologies) on 24-well plates according to the manufacturer's instructions. For the differentiation into each lineage, 12 wells were seeded, allowing sampling of the supernatant and RNA purification in triplicates for each time point. Samples were analysed after 0, 2, 3 and 5 d for mesoderm and endoderm, or 7 d for ectoderm. Each day, 55 µl supernatant was sampled for bioluminescence quantification. Additionally, after each measurement, cells were counted with a Countess 3 Automated Cell Counter (Invitrogen) for normalization of the luminescence signals. Subsequently, RNA was isolated for RT-qPCR analysis of *NEAT1* levels, lineage markers and Yamanaka factors.

**Gene expression manipulation with CRISPR–Cas9 (CRISPRi).** Gene expression of *NEAT1* or *GUARDIN* was transcriptionally repressed in HEK293T cells by co-transfecting a CAG-driven catalytically inactive version of *Streptococcus pyogenes* Cas9 (D10A and H840A) with N- and C-terminal SV40 nuclear localization signals fused to the KRAB domain of ZIM3 and a plasmid mixture expressing three (*NEAT1*) or two (*GUARDIN*) human U6 promoter-driven sgRNAs (20-nucleotide spacer) targeting the TSS of *NEAT1* or *GUARDIN*. An sgRNA mixture targeting the unrelated *MAPT* locus was co-transfected as a non-targeting control.

Gene expression was followed at 48 h post-transfection (medium change) for several hours by NLuc assay of the supernatant.

**Generation of stable cell lines carrying the INSPECT reporter system via CRISPR–Cas9.** We used plasmids expressing a mammalian codon-optimized *S. pyogenes* Cas9 (*Spy*Cas9) with two C-terminal SV40 nuclear localization signals (CBh promoter driven) and a human U6 promoter-driven sgRNA with a 19- to 21-bp spacer for genome targeting to generate stable INSPECT HEK293T, Jurkat E6.1 or H9 hESC lines (Extended Data Fig. 8). The insertion sites were chosen using the consensus NAG^GWW (where '^' indicates the insertion site). This insertion site leads to a functional splice donor and splice acceptor sequence in combination with the inserted INSPECT construct.

The efficiency of an sgRNA for a specific target site was tested by T7 endonuclease I assay (NEB) according to the manufacturer's protocol after 72 h post-transfection of cells with plasmids encoding Cas9 and the targeting sgRNA.

Optionally for knock-in, an i53 expression plasmid (a genetically encoded 53BP1 inhibitor)[55] was co-transfected in addition to the CRISPR–Cas9 plasmid to enhance homologous recombination.

In addition to the genetically encoded inhibition, the non-homologous end-joining pathway was blocked selectively by the DNA-dependent protein kinase inhibitor AZD7648 (ref. [56]). The medium was supplemented with 0.5 µM AZD7648 (HY-111783; MedChemExpress) 3 h before transfection.

The INSPECT donor DNA plasmid contains homology arms of at least 800 bp, flanking the reporter construct. It also contains a selection cassette to select for cells undergoing successful Cas9-mediated homologous recombination (Extended Data Fig. 9).

After knock-in, cells undergo selection, counterselection and monoclonalization. We assume an overall time of 8 weeks to generate monoclonal INSPECT cell lines (Extended Data Fig. 10).

*HEK293T and H9 selection.* Three days post-transfection (6-well format), the medium was supplemented with puromycin (10 µg ml$^{-1}$ for HEK293T cells; 0.5 µg ml$^{-1}$ for H9 cells). Cells were observed daily and detached with Accutase when surviving colonies reached a colony size of ~50 cells. After puromycin selection, the *EEF1A* promoter-driven

selection cassette was removed from the genome by three sequential transient expressions of CAG-driven Flp recombinase. Cells underwent counterselection with 2 µM ganciclovir (GCV) for 2 weeks.

Next, cells were monoclonalized by limited dilution into 96-well plates (culture media supplemented with 2 µM GCV for counterselection). Besides puromycin-*N*-acetyltransferase, the selection cassette encodes the herpes simplex virus thymidine kinase, which phosphorylates the prodrug GCV. The phosphorylated GCV is an analogue of guanosine nucleoside and terminates the elongation of DNA strands, resulting in cell death[57].

Cells were cultured until the colony size was big enough to expand over 48-well plates. After reaching confluence, half of the cell mass was used to isolate genomic DNA using a Wizard Genomic DNA Purification Kit (Promega). Genotyping of the genomic DNA was performed using Platinum SuperFi II PCR Master Mix (Thermo Fisher Scientific) or LongAmp Hot Start Taq 2X Master Mix (NEB) according to the manufacturer's protocol. Primer sequences and corresponding polymerase usage are indicated in Supplementary Table 2. The PCR products from clones, where the genotyping indicates homozygosity, were sent for Sanger sequencing to verify their sequence integrity. Genotyping results are shown in Supplementary Fig. 2.

*Jurkat E6.1 selection.* Three days post-nucleofection (12-well format), the medium was replaced with medium containing puromycin (2 µg ml$^{-1}$). Cells were observed daily and singularized by pipetting. This step was repeated until no substantial puromycin-mediated cell death could be observed. After selection, the *EEF1A*-driven selection cassette was removed from the genome by three sequential transient expressions of CMV promoter-driven Flp recombinase. Cells then underwent counterselection with 2 µM GCV for 2 weeks. Next, cells were monoclonalized by limited dilution into 96-well plates.

**Generation of *NEAT1* KO cell lines by activating the Cre recombinase-inducible KO switch.** Cells carrying the INSPECT construct underwent three transient transfections of plasmid-encoded Cre recombinase. Upon Cre expression, the inverted polyA sequence, together with an inverted upstream splice acceptor, was flipped irreversibly into its sense direction to induce an irreversible KO of the endogenous gene by transcription termination (Extended Data Fig. 6).

**Induction of TRE3G-driven transcription.** For transient transfection experiments with TRE3G promoter-driven constructs or stable cell lines containing a TRE3G promoter-driven sequence, the indicated amount of Dox was added to each well 16 h after transfection.

**Plasmid nucleofection.** Jurkat E6.1. For nucleofection, 1,000,000 Jurkat E6.1 cells were centrifuged (90 RCF for 10 min) and resuspended in 100 µl Amaxa Cell Line Nucleofector Kit V (Lonza) Nucleofector solution according to the manufacturer's protocol. The cell suspension was supplemented with 1 µg plasmid DNA and immediately transferred into cuvettes for electroporation. The Nucleofector program CL-120 of the 4D-Nucleofector (Lonza) was applied. After adding 500 µl pre-warmed RPMI media, cells were transferred carefully into a 12-well plate (2 ml final volume).

*H9.* A total of 1,000,000 H9 cells were detached and underwent nucleofection with the P3 Primary Cell 4D-Nucleofector X Kit (Lonza) using the CB-156 program of the 4D-Nucleofector (Lonza).

**Plasmid transfection.** HEK293T cells were transfected either with X-tremeGENE HP (Roche) or jetOPTIMUS DNA transfection reagent (Polyplus) according to the manufacturer's protocol. Total DNA amounts were kept constant across all samples in all transient experiments to yield comparable complex formation and thus reproducible results. For 96-well plates, 100 ng plasmid DNA was used, whereas for

6-well plates, 2.4 µg plasmid DNA was used per well. Cells were plated the day before transfection (25,000 cells per well in 200 µl for 96-well plates or 600,000 cells per well in 3 ml for 6-well plates).

**Viability/cytotoxicity assay.** HEK293T cells were seeded in a 96-well format (25,000 cells per well in 200 µl cultivation media). After 24 h, cells were transfected with plasmid DNA coding for mouse *Pgk1* promoter-driven HIV-1 gag and *Pgk1* promoter-driven Gag–PCP together with PP7-tagged INSPECT or control plasmid to mimic the cellular stress of transfection and transgene expression. After 48 h, cells were treated with the viability/cytotoxicity reagent (ApoTox-Glo Triplex Assay; Promega) according to the manufacturer's protocol. Signals were obtained at 505 nm (viability) and 520 nm (cytotoxicity) using a Varioskan LUX microplate reader (Thermo Fisher Scientific).

## Protein biochemical analysis

**ELISA.** A sandwich ELISA was performed to detect the secreted human IL-2 in supernatants of Jurkat E6.1 cells. The monoclonal IL-2 detection antibody (clone MQ1-17H12; rat isotype; 14-7029-81; Thermo Fisher Scientific) was diluted to 5 µg ml$^{-1}$ in phosphate-buffered saline (PBS) and immobilized on a 96-well plate (Nunc-Immuno; Merck) overnight at 4 °C (100 µl per well). After one washing step (all washing steps were performed with 250 µl DPBS supplemented with 0.05% Tween 20), the wells were incubated for 1 h with blocking buffer (2% Bovine Serum Albumin Fraction V (Sigma–Aldrich) in DPBS). After five washing steps, 100 µl supernatants, recombinant IL-2 control (diluted in PBS; PHC0023; Thermo Fisher Scientific) or RPMI media control were incubated for 1 h with gentle shaking. After five washing steps, 2 µg ml$^{-1}$ of the biotinylated polyclonal IL-2 detection antibody (diluted in blocking buffer; rabbit isotype; 13-7028-85; Thermo Fisher Scientific) was incubated for 2 h with gentle shaking (100 µl per well). Following another five washing steps, 100 µl of a 1:4,500 dilution of a streptavidin–horseradish peroxidase conjugate (N100; Thermo Fisher Scientific) was added to each well. After 30 min of gentle shaking, wells were washed five more times. For substrate solution, 100 mM 3,3′,5,5′-tetramethylbenzidine (Sigma–Aldrich) in DMSO was diluted 1:1,000 in 50 mM phosphate citrate buffer, then 30% hydrogen peroxide (Sigma–Aldrich) was added 1:5,000. The reaction was stopped using 1 M nitric acid. The 3,3′,5,5′-tetramethylbenzidine substrate intensity was measured at 450 nm using a Varioskan LUX microplate reader (Thermo Fisher Scientific).

## RNA analysis

**RNA smFISH.** HEK293T cells were seeded (300,000 cells per well) 24 h before fixation in 2-well µ-Slides (Ibidi). After three washing steps with DPBS (Gibco), cells were fixed for 10 min at room temperature using 10% neutral buffered formalin (Sigma–Aldrich). Following three additional washing steps with DPBS, the fixed cells were permeabilized in 70% ethanol overnight at 4 °C. After another three washing steps, the samples were incubated in a prehybridization buffer containing 2× saline sodium citrate solution and 10% deionized formamide (Merck) for 15 min. Subsequently, 50 µl hybridization buffer consisting of 50 µg competitor tRNA from *E. coli* (Roche), 10% dextran sulfate (VWR), 2 mg ml$^{-1}$ UltraPure BSA (Thermo Fisher Scientific), 10 mM ribonucleoside vanadyl complex (NEB) and 0.125 µM Stellaris FISH Probes, Human *NEAT1* 5′ Segment with Quasar 570 Dye (LGC; Biosearch Technologies) was added. The multiplicity of probes on a single RNA molecule increases the signal-to-noise ratio. The samples were sealed with parafilm and incubated at 37 °C in a humidified chamber for 6 h. The exposure to light was kept at a minimum to avoid photobleaching. After removing the parafilm, samples were incubated twice with prehybridization buffer for 15 min at 37 °C, followed by the last washing step with DPBS. Finally, 10 µl ProLong Gold Antifade Mountant with DAPI (Thermo Fisher Scientific) and a cover slide for imaging were added. Fluorescence microscopy was performed with a Zeiss

microscope (Axiovert 200M; Carl Zeiss Microscopy) using the 63× oil immersion objective.

**Isolation of RNA from cell lysates.** RNA extraction was performed using the Monarch Total RNA Miniprep Kit (NEB) according to the manufacturer's instructions. RNA was stored at −80 °C.

**Isolation of RNA from virus-like particle-containing supernatant.** At 48 h post-transfection, 130 µl supernatant per well from 96-well transfection experiments was transferred to a 96-well PCR plate and centrifuged at 500 RCF for 10 min. Subsequently, 100 µl was carefully removed and processed with the QiaAmp Viral RNA Mini Kit (QIAGEN) according to the manufacturer's protocol. RNA was stored at −80 °C.

**RT-qPCR.** RNA samples were subjected to DNAse digestion with DNAse I (NEB) with the following protocol changes: the DNAse concentration was doubled and the incubation regimen was extended to 30 min interspersed with short mixing steps, followed by inactivation at 75 °C for 15 min. For RT-qPCR reactions, TaqMan probes (FAM + BHQ1) and primers were ordered from IDT. The Luna Cell Ready Probe One-Step RT-qPCR Kit (NEB) was used according to the manufacturer's protocol. Reactions were run in 384-well plates (10 µl per well) in technical duplicates. Control amplification reactions without reverse transcriptase were carried out as a reference for each RT-qPCR run. The reactions were performed and monitored in an Applied Biosystems QuantStudioTM 12K Flex Real-Time PCR system. Primer and probe sequences are provided in Supplementary Table 2.

**Semi-quantitative RT-PCR.** Cells were collected for 5 min at 200 RCF after cell detachment using Accutase solution (Gibco; Thermo Fisher Scientific). The RNA was extracted with the RNeasy Mini Kit (QIAGEN) according to the manufacturer's protocol. Reverse transcription was performed using the SuperScript IV VILO Master Mix (Invitrogen, Thermo Fisher Scientific) according to the manufacturer's protocol with an extended incubation time (30 min at 50 °C), followed by a semi-quantitative PCR using LongAmp Hot Start Taq 2X Master Mix (NEB) according to the manufacturer's protocol. Primers were designed to amplify the full-length mRNA (*IL2* and *GUARDIN*). For long transcripts >20 kb (*NEAT1*), primers flank the INSPECT insertion site. The elongation time was adjusted to allow amplification of RNA splicing isoforms, which contain INSPECT.

## Fluorescence/bioluminescence detection and gamma counting

**Bioluminescence microscopy.** Bioluminescence microscopy images were obtained with an LV200 bioluminescence life-imaging system (Olympus) in 8-well Grid-500 µ-Slides (Ibidi). HEK293T cells were plated (51,000 cells per well) 24 h before imaging. For correlative bioluminescence imaging–smFISH analysis, the furimazine stock solution (Nano-Glo Luciferase Assay System; Promega) was used according to the manufacturer's protocol in a 1:100 final dilution but without the provided lysis buffer. After imaging, the cells were quickly fixed for smFISH (see RNA smFISH section).

Jurkat E6.1 cells (220,000 cells per well) were induced with 1 ng ml$^{-1}$ PMA, 1 µg ml$^{-1}$ PHA and 0.1 µM A23187 and subsequently imaged for 10 h at 37 °C. The RPMI media was supplemented with 20 mM HEPES, Nano-Glo Endurazine Live Cell Substrate (Promega) and Extracellular NanoLuc Inhibitor (Intracellular TE Nano-Glo Substrate/Inhibitor; Promega) according to the manufacturer's instructions.

**Bioluminescence quantification.** For bulk quantifications of cell bioluminescence, cells were plated and transfected in 96-well format. For bioluminescence detection of secreted NLuc, the supernatant was sampled (80 µl) at the indicated time points and detected using the Nano-Glo Luciferase Assay System (Promega) on the Centro LB 960

(Berthold Technologies) plate reader with 0.5-s acquisition time. For dual-luciferase readout using the Nano-Glo Dual-Luciferase Reporter Assay System (Promega), NLuc and FLuc signals were read out on-plate 48 h post-transfection. Signals were obtained with 0.5-s acquisition time 10 min after adding reagent 1 (ONE-Glo EX Luciferase) for FLuc and 10 min after adding reagent 2 (NanoDLR Stop & Glo) for NLuc. Reagent 2 includes FLuc inhibitor. The relative luminescence units (RLU) of FLuc and NLuc are expressed relative to the signals obtained by INSPECT$_{control}$, defined as 1 to make the $y$ axis range consistent across experiments.

**Epifluorescence microscopy.** Epifluorescence microscopy images were taken on an EVOS FL Auto Imaging System (Invitrogen, Thermo Fisher Scientific) with identical settings across all samples.

**Fluorescence flow cytometry analysis.** Adherent cells from 48-well plate transfections were dissociated with 200 µl Accutase and centrifuged at 200 RCF for 5 min at room temperature. Cell pellets were resuspended in ice-cold 500 µl DPBS containing 0.4% paraformaldehyde, washed with DPBS, resuspended, transferred into conical 5 ml polystyrene round-bottom tubes including a cell-strainer cap and kept on ice. In fluorescence-activated cell sorting (FACS), the main cell population was gated according to forward scatter and sideward scatter (Supplementary Fig. 3). Next, single cells were chosen according to their FSC-A and FSC-W values. Subsequently, the transfected cells were gated according to their green fluorescence (530 nm) or red fluorescence (586 nm). These gates were kept constant for all subsequent experimental conditions.

**γ-scintillator measurements.** Jurkat E6.1 cells carrying the INSPECT$_{NIS}$ reporter system homozygously in their genome (500,000 cells in 1 ml RPMI media per well) were activated for *IL2* expression. Analysis of intracellularly accumulated $^{131}I^-$ was performed 16 h post-induction. Each well was incubated with 100 kBq freshly calibrated $^{131}I^-$ in PBS for 2 h at 37 °C. After rinsing the wells three times with 10 ml PBS, cells were lysed in 1 ml 0.5 M NaOH. The radiation intensity was analysed 1 day later with a γ-scintillator (Wizard$^2$ 1-Detector Gamma Counter; PerkinElmer) for 1 min, resulting in an output of counts per minute.

**Software**

**Image analysis.** Image analysis was performed with Fiji with the BioVoxxel toolbox plugin[58]. Each image stack obtained from smFISH microscopy resulted in 21 layers of every channel, which were *z*-projected using maximum intensity projection. An ImageJ macro was created for automated feature recognition and counting. All images were analysed with the same settings. Only the variable prominence was adjusted for each batch.

For visualization of smFISH images, images underwent convoluted background subtraction (mean) with a radius of (100 pixels for Fig. 6c; 40 pixels for Fig. 6j). Images were linearly contrast enhanced only for visualization purposes (same settings for all images).

Images obtained by epifluorescence microscopy were analysed by segmenting the cells on the corresponding brightfield image and calculating the arithmetic mean of the fluorescence intensities per cell.

Images obtained from bioluminescence microscopy after IL-2 induction of Jurkat E6.1 INSPECT$_{IL2:SigP–NLuc}$ were analysed by randomly chosen ROIs from cells that showed responsiveness in the last frame, and were manually analysed in Fiji using the mean bioluminescence imaging intensity of each ROI at the indicated time points.

**Splice site predictions.** Splice site predictions were conducted using NetGene2 (ref. [59]).

**RT-qPCR analysis.** The acquired data were processed and exported using QuantStudio 12K Flex software. For some RT-qPCR traces, individual baseline correction was performed.

**Design considerations for INSPECT constructs**

The constructs and all plasmids used in this work were codon optimized for mammalian cell expression. Large gene fragments (gBlocks) and oligodeoxynucleotides were obtained from IDT. INSPECT features modular construction. We chose the exonic splice consensus sequence NAG^GWW for CRISPR–Cas9-mediated INSPECT insertion. The resulting flanking exonic sequences should have a minimum length of 30 bp to ensure precise splicing. For coding genes, INSPECT should not be introduced more than 50 bp downstream of the host CDS to avoid nonsense-mediated decay[60]. For lncRNAs, only restraints on the minimum exon size apply. Splice prediction software should further evaluate the chosen splice sites (for example, NetGene2 or Human Splicing Finder 3) after in silico insertion of the INSPECT reporter construct.

Insertion sites within native introns can be chosen according to sgRNA efficiency, but the distance to splice sites should be considered (>30 nucleotides).

The size of the INSPECT construct equipped with a secreted NLuc is 3,133 bp from splice donor to splice acceptor (3,076 bp for cytosolic NLuc).

We recommend that each INSPECT construct be validated by RT-(q)PCR flanking the insertion site, followed by Sanger sequencing to confirm the sequence integrity of the host RNA.

**Statistics and reproducibility**

Statistics were calculated with Prism 9 (GraphPad) as indicated. For clarity, not all statistical results are graphically presented but complete tabulated results are provided in Supplementary Table 1. All statistical analyses on RT-qPCR data were performed on the Ct (Cq), ΔCt (ΔCq) or ΔΔCt (ΔΔCq) values. Fold-changes (using the $2^{-\Delta Ct}$ or $2^{-\Delta\Delta Ct}$ method) were only used for graphical visualization and are displayed as geometrical means and geometrical s.d.

**Reporting summary**

Further information on research design is available in the Nature Research Reporting Summary linked to this article.

## Data availability

Unprocessed microscopy images are provided in the Supplementary Information. The results of genotyping are available in Supplementary Fig. 2. All of the other data supporting the findings of this study are available from the corresponding author on reasonable request. Source data are provided with this paper.

## Code availability

The ImageJ macro for automated cell area segmentation for fluorescence quantification, as well as feature recognition and the counting of paraspeckles after smFISH staining, is available in Supplementary Table 2.

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

## Acknowledgements
We acknowledge support from the Federal Ministry of Education and Research (BMBF) and the Free State of Bavaria under the Excellence Strategy of the Federal Government and the Länder through the ONE MUNICH project Munich Multiscale Biofabrication (to J.G. and G.G.W.). We thank V. Morath for help with the NIS experiments.

## Author contributions
D.-J.J.T. conceived of the INSPECT reporter mechanism, designed most of the constructs and co-conducted the cell and biochemical experiments. N.A. and J.G. co-designed and co-generated the constructs and co-conducted the cell and biochemical experiments E.-M.L., J.G., T.H.S., M.B., E.S., M. Dyka, G.R., T.P. and M.Ž. co-generated the constructs and co-conducted the cell and biochemical experiments. D.-J.J.T., J.G., and G.G.W. co-designed the RNA reporter secretion system. J.G. and N.A. conducted and co-analysed the respective experiments. M. Drukker advised on the *NEAT1* lncRNA study. D.-J.J.T., N.A., S.I., J.G., T.H.S., M.B., E.S., T.G. and M.G. conducted the *NEAT1* experiments. D.-J.J.T. and N.A. analysed the data with inputs from G.G.W. D.-J.J.T. and N.A. co-generated the figures with edits from G.G.W. D.-J.J.T., N.A. and G.G.W. wrote the manuscript. D.-J.J.T. coordinated the experimental activities. G.G.W. supervised the research.

## Funding

## Competing interests
D.-J.J.T. and G.G.W. have filed a patent application WO2022008510A3 on INSPECT's capabilities. The other authors declare no competing interests.

## Additional information
**Extended data** is available for this paper at https://doi.org/10.1038/s41556-022-00998-6.

**Correspondence and requests for materials** should be addressed to Gil Gregor Westmeyer.

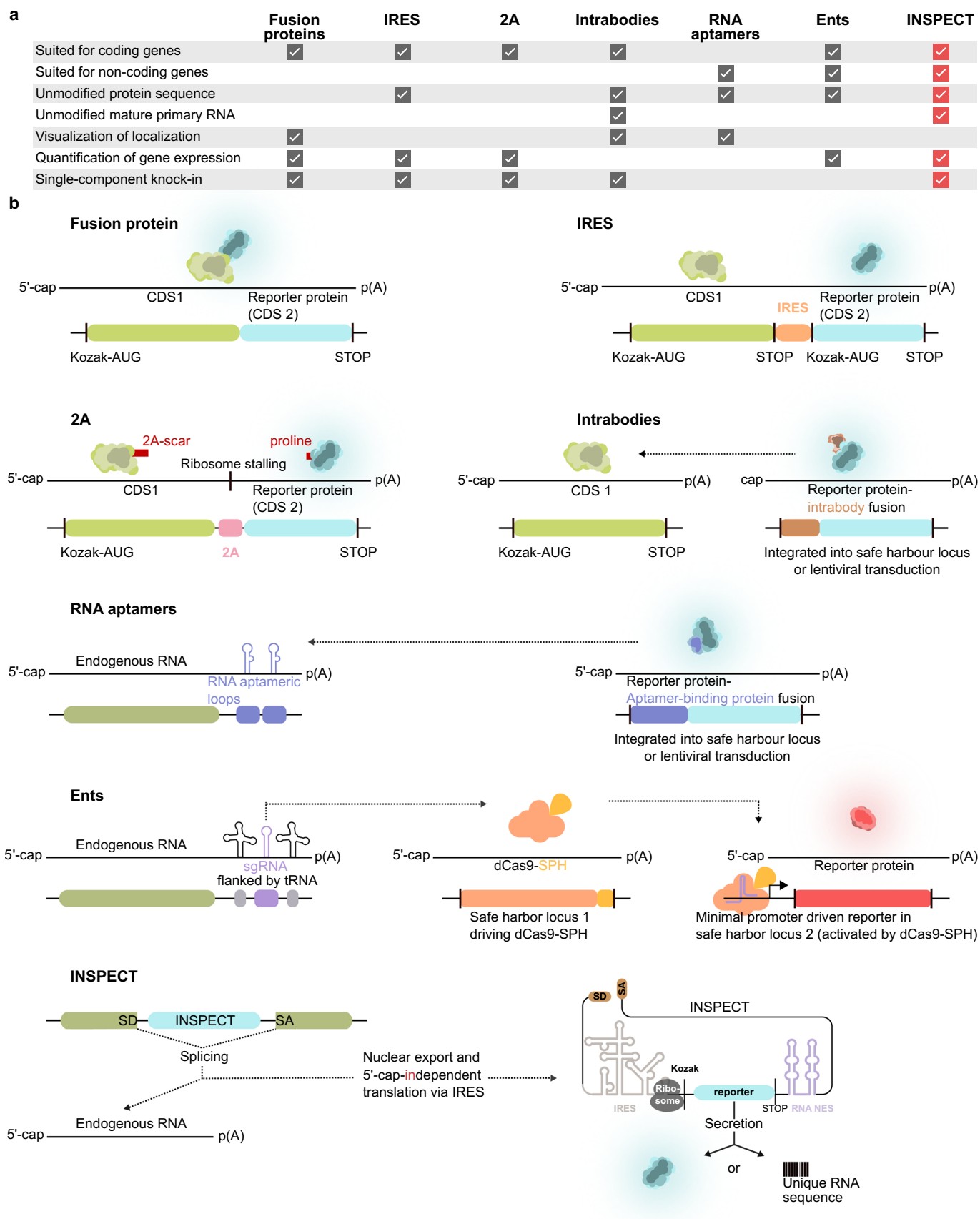

| | Fusion proteins | IRES | 2A | Intrabodies | RNA aptamers | Ents | INSPECT |
|---|---|---|---|---|---|---|---|
| Suited for coding genes | ✓ | ✓ | ✓ | ✓ | | ✓ | ✓ |
| Suited for non-coding genes | | | | | ✓ | ✓ | ✓ |
| Unmodified protein sequence | | ✓ | | ✓ | ✓ | ✓ | ✓ |
| Unmodified mature primary RNA | | | | ✓ | | | ✓ |
| Visualization of localization | ✓ | | | ✓ | ✓ | | |
| Quantification of gene expression | ✓ | ✓ | ✓ | | | ✓ | ✓ |
| Single-component knock-in | ✓ | ✓ | ✓ | ✓ | | | ✓ |

**Extended Data Fig. 1 | See next page for caption.**

**Extended Data Fig. 1 | Overview of existing genetically encoded approaches to monitor gene expression compared to INSPECT. a**, tabulation of the central capabilities and properties of currently available gene reporter methods compared to INSPECT. **b**, schematic depiction of each of the methods. Fusion protein: A direct fusion (here C-terminal) of a reporter protein (CDS2) resulting in a fusion protein to the native sequences (CDS1). IRES: Internal ribosome entry sites mediate cap-independent translation of the 3′-cistron proportional to CDS1 expression but modify the 3′-UTR of the endogenous mRNA. 2A: For stoichiometric translation of CDS1 and CDS2, 2A sequences use a ribosome stalling mechanism, leaving scars on the host protein. To visualize target proteins, a subset of nanobodies can be expressed in cells fused to fluorescent proteins (XFPs). RNA aptamer: Insertion of MS2/PP7 RNA aptamers into the UTR of an mRNA or a non-coding RNA enables visualization via an aptamer-binding protein (ABP)-XFP fusions. Ents (Endogenous transcription-gated switch): The tripartite system is composed of a sgRNA flanked by tRNAs, integrated into the 3′-UTR of a gene, which is released by endogenous RNAse Z/P, resulting in a poly(A)-deficient host transcript, a free poly(A)-tail and a free sgRNA that in turn induces the expression of a separate integrated reporter system via a dCas9 transactivator system which is also integrated into the genome. The host mRNA lacking the poly(A) tail then should be exported to the cytosolic environment. INSPECT: The Intron-encoded Scarless Programmable Extranuclear Cistronic Transcript is spliced, stabilized, exported from the nucleus into the cytosol for cap-independent translation or, alternatively, secreted from the cell as an RNA-reporter reporter.

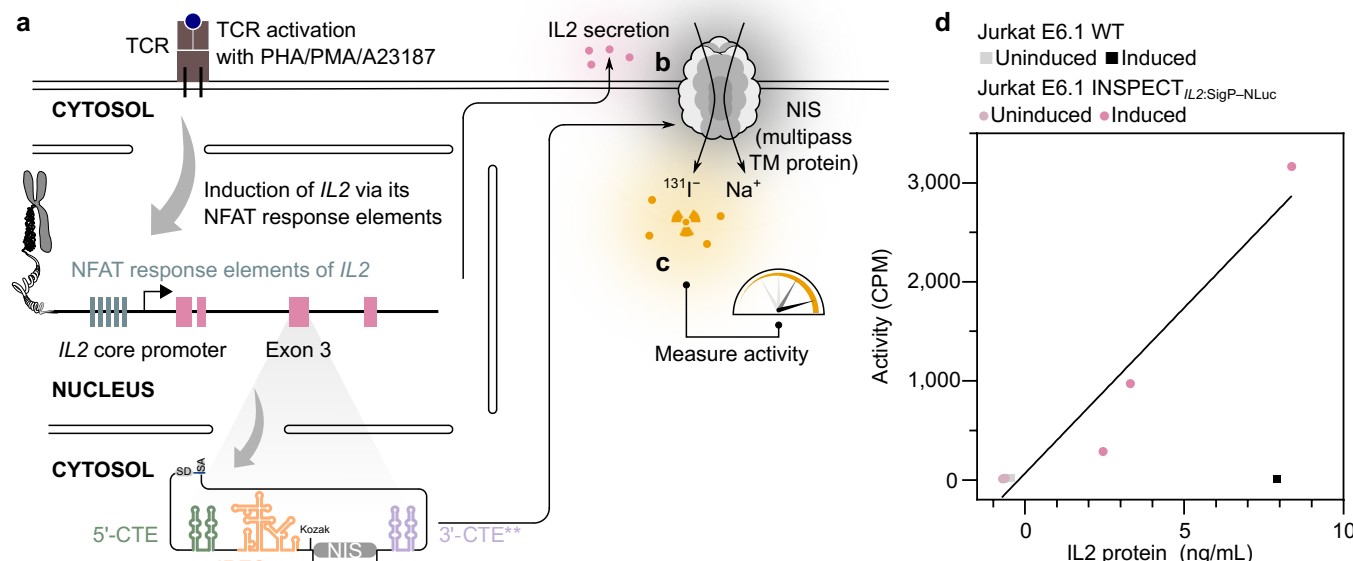

**Extended Data Fig. 2 | INSPECT reporter enables modular read-out of coding genes using NIS-mediated intracellular accumulation of the radioisotope I-131⁻. a**, TCR signaling was induced with the tripartite mixture of PHA (1 ng ml⁻¹), PMA (1 µg ml⁻¹), and the Ca²⁺ ionophore (Br)-A23187 (0.1 µM). The subsequent induction of *IL2* was then read out via INSPECT equipped with **b**, the sodium iodide symporter (NIS, 4,492 bp from splice donor to splice acceptor) knocked into exon 3 of the NFAT-controlled *IL2* locus in Jurkat E6.1 cells. **c**, NIS can be harnessed as a reporter modality by measuring the accumulation of the radioisotope ¹³¹I⁻. **d**, Counts per minute (CPM) of radioactive decay of the radioisotope I-131⁻ plotted against IL2 protein concentrations with the linear fit (Y = 334.8 X + 63.94; R² = 0.93) from 3 independent clones of INSPECT*IL2:NIS* Jurkat E6.1 cells analyzed after 16 hours of TCR activation and 2 hours incubation with 100 kBq ¹³¹I⁻. Source data are available online.

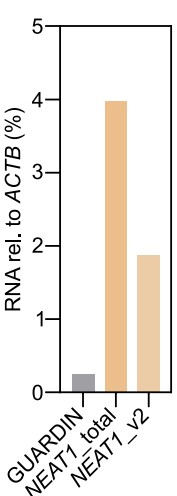

**Extended Data Fig. 3 | Relative expression strength of *GUARDIN* and *NEAT1* isoforms in HEK293T wild-type.** Bars represent average of technical duplicates. Source data are available online.

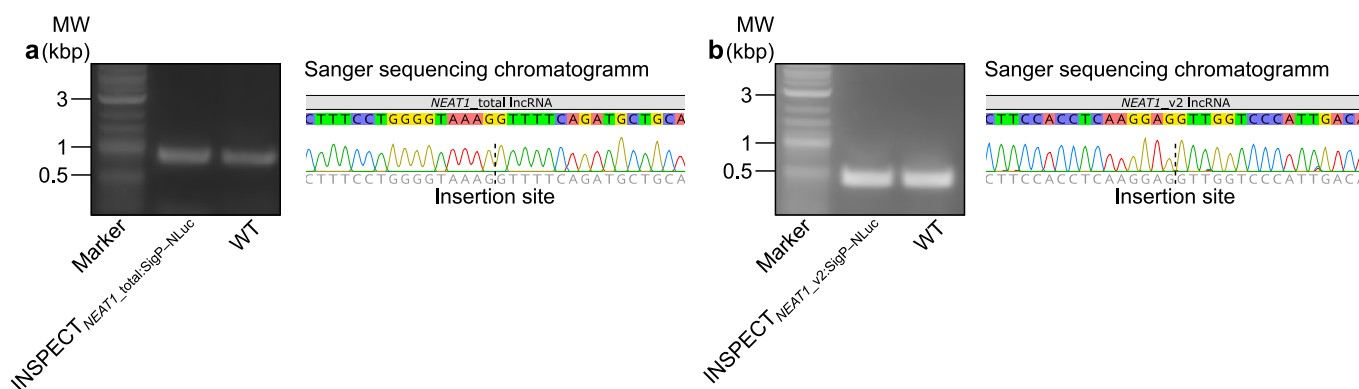

**Extended Data Fig. 4 | *NEAT1* RT-PCR and Sanger sequencing after INSPECT insertion. a**, Insertion into *NEAT1*_total, or **b**, insertion into *NEAT1*_v2. Semiquantitative RT-PCR was performed once for each insertion site, and corresponding bands were extracted from the agarose gel and sent for Sanger sequencing.

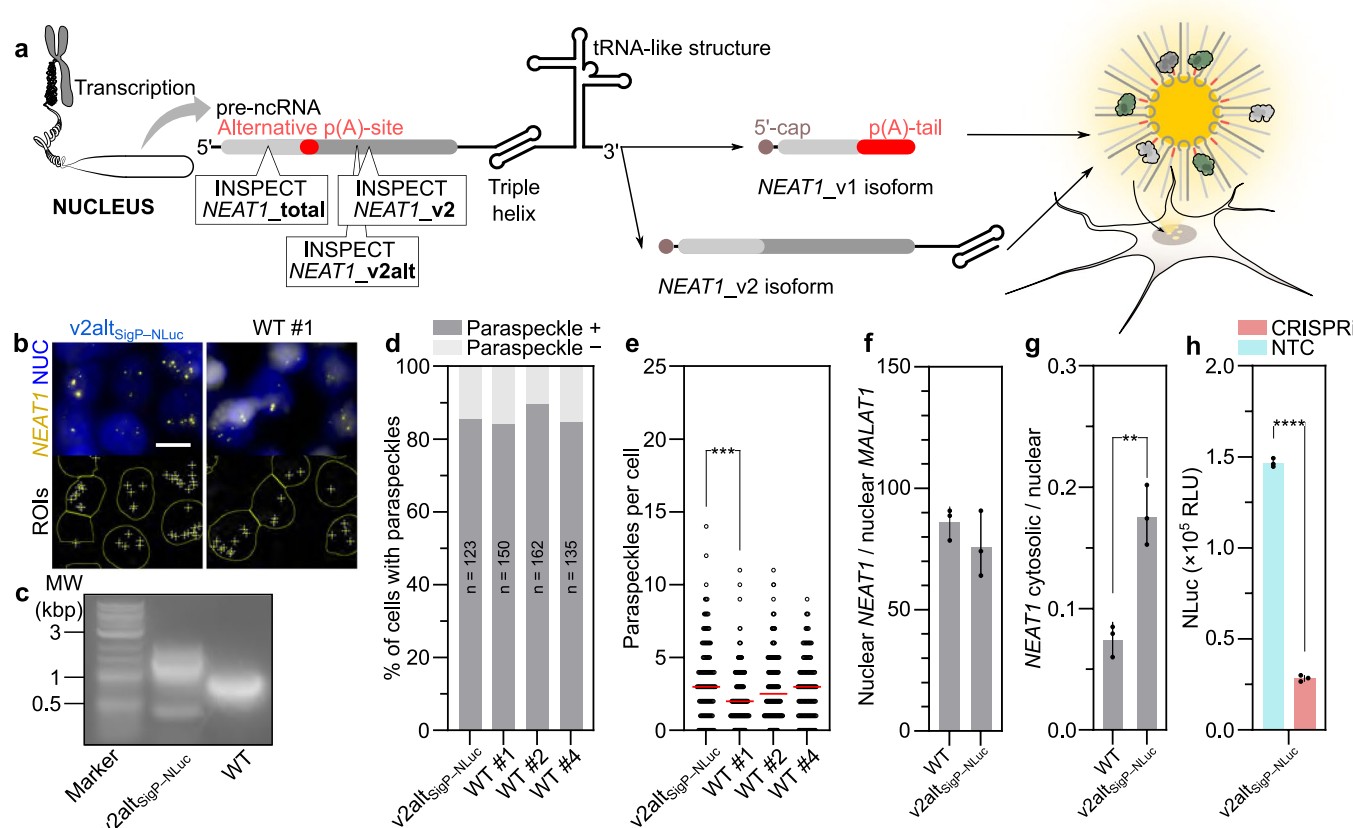

**Extended Data Fig. 5 | Alternative insertion site for monitoring of *NEAT1*_v2 (*NEAT1*_v2alt_SigP-NLuc).** The sub-panels follow the same comparisons as in Fig. 6. **a**, INSPECT equipped with a secreted NLuc was inserted via CRISPR-Cas9 into *NEAT1*_v2alt, which reports the long *NEAT1*_v2 isoform exclusively. **b**, Representative image of the DAPI and smFISH-probed (against *NEAT1*) channel. The images in the bottom row of each sub-panel illustrate which signals were identified as paraspeckles (+) inside the nucleus (circles) for automatic qualification. Scale bar, 10 μm. **c**, RT-PCR of *NEAT1*_v2 with primers flanking the v2alt insertion site reveals the splice isoforms in wild-type and transgenic HEK293T. **d,e**, Paraspeckle count in INSPECT cells and wild-type HEK293T (n indicates the number of cells analyzed regarding speckle count, full statistical analysis of Kruskal-Wallis-Test in Supplementary Table 1). **f**, Comparison of *NEAT1*

abundance relative to *MALAT1* in the nucleus after alternative INSPECT insertion in *NEAT1*_v2. **g**, Cytosolic/nuclear ratio of *NEAT1* RNA localization. **h**, NLuc signal (RLU) obtained from the supernatant of INSPECT cells 72 hours after transfection with plasmids for CRISPRi of *NEAT1*. CRISPRi was achieved via plasmids encoding a dCas9–transcriptional-repressor fusion chimera targeted with three sgRNAs against the *NEAT1* transcription start site. The medium was exchanged 24 hours before measurement to reset the secreted luciferase signal. The bars of **f-h** represent three biological replicates, with the error bar representing the s.d. Selected results of a two-tailed unpaired *t*-test are shown; $P**<0.01$; $****P<0.0001$. Full statistical results are given in Supplementary Table 1. Source data are available online. Wild-type reference data are duplicated from Fig. 6.

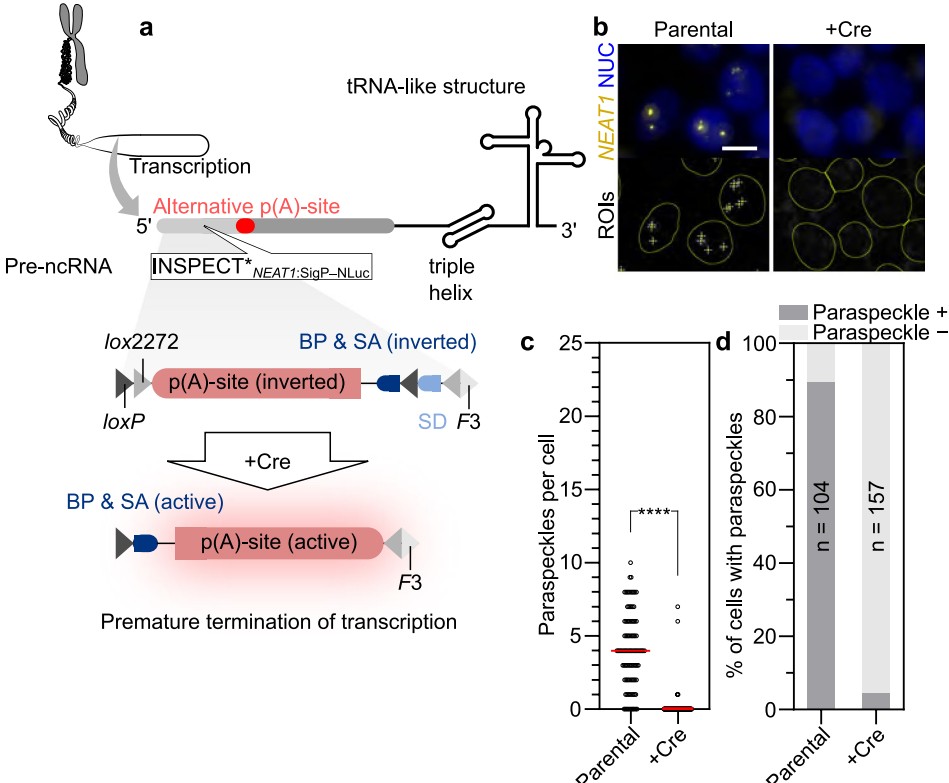

**Extended Data Fig. 6 | Cre-mediated off-switch leads to disruption of paraspeckle formation. a**, Cre-recombinase flips the inverted poly(A) sites leading to premature transcription termination. In addition, the branch point and the splice acceptor (BP & SA) ensure the usage of the poly(A). **b**, Representative images of smFISH against *NEAT1* in parental cells carrying INSPECT*$_{NEAT1\_total:SigP-NLuc}$ inserted upstream of the alternative polyA site and

after activation of the off-switch by expression of the Cre recombinase. **c, d**, Quantification of paraspeckles. The red line indicates the median of the distribution of paraspeckles per cell. **** denotes *P*<0.0001 in a Mann-Whitney test with n indicating the number of cells analyzed regarding speckle count. Full statistical results are given in Supplementary Table 1. Source data are available online.

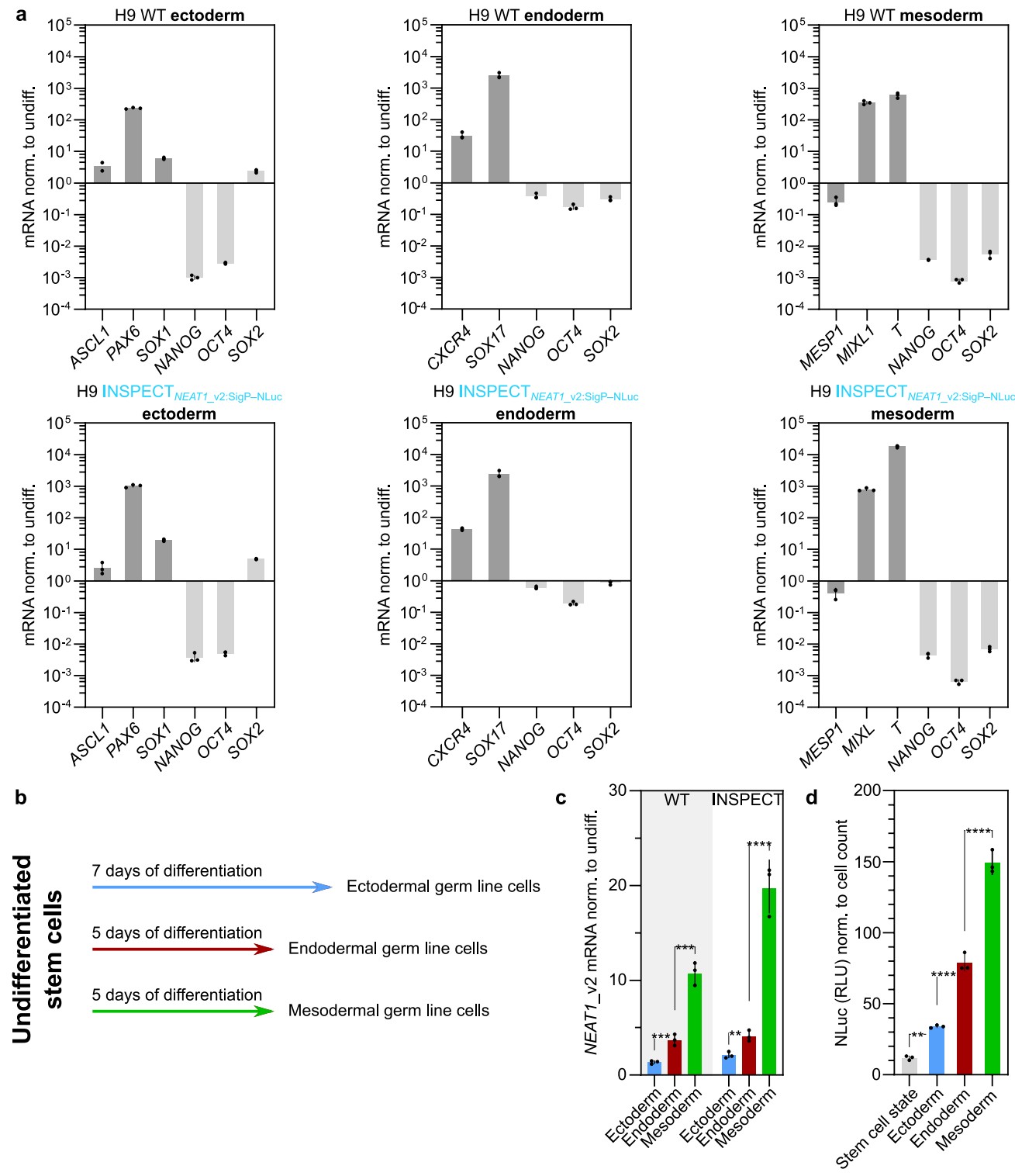

**Extended Data Fig. 7 | Monitoring of *NEAT1* upregulation during stem cell differentiation in H9 human embryonic stem cells with INSPECT<sub>SigP-NLuc</sub> knocked into *NEAT1*_v2. a**, RT-qPCR validation of H9 differentiation into ectoderm, endoderm, and mesoderm in wild-type and INSPECT<sub>NEAT1_v2:SigP-NLuc</sub> H9 stem cells. The fold change of lineage-specific markers, Yamanaka factors, and *NEAT1* RNA was analyzed for each differentiation process. **b**, Duration of each of the differentiation protocols. **c**, Fold change of *NEAT1*_v2 RNA level and **d**, relative luminescence signals obtained from the differentiated cells before RNA extraction at the final stage of each differentiation. For **a**,**c**, and **d** bars represent the mean of biological replicates (*n* = 3) with error bars showing the s.d. (geometric for **a**,**c**, and arithmetic for **d**). Selected results of one-way ANOVA with Bonferroni multiple comparisons test are shown; ****$P<0.0001$; ***$P<0.001$; **$P<0.01$. Full statistical results are given in Supplementary Table 1. Source data are available online.

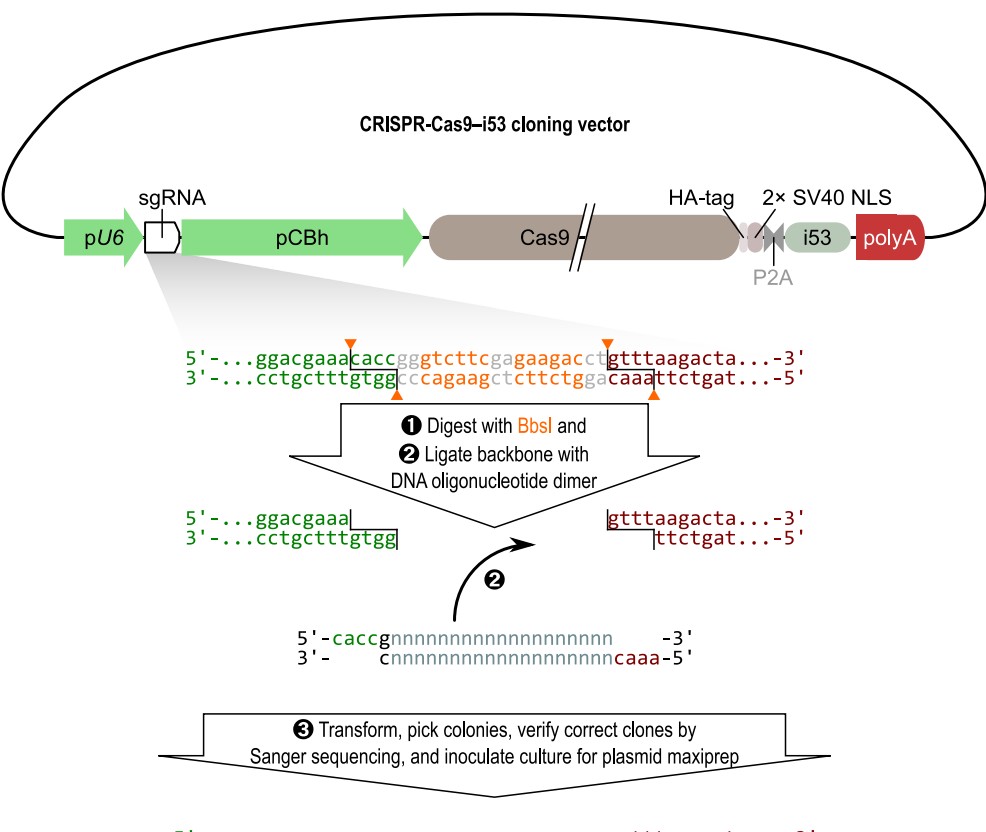

**Extended Data Fig. 8 | Cloning scheme of the CRISPR/Cas9 plasmid using a Type IIS restriction enzyme.** The empty CRISPR-Cas9–i53 cloning vector is digested with the type IIS restriction enzyme BbsI (isoschizomer: BpiI). The backbone fragment is used for subsequent ligation with a spacer fragment derived from the dimerization of two deoxyoligonucleotides. The ligation product is then used for transformation of competent *E.coli*.

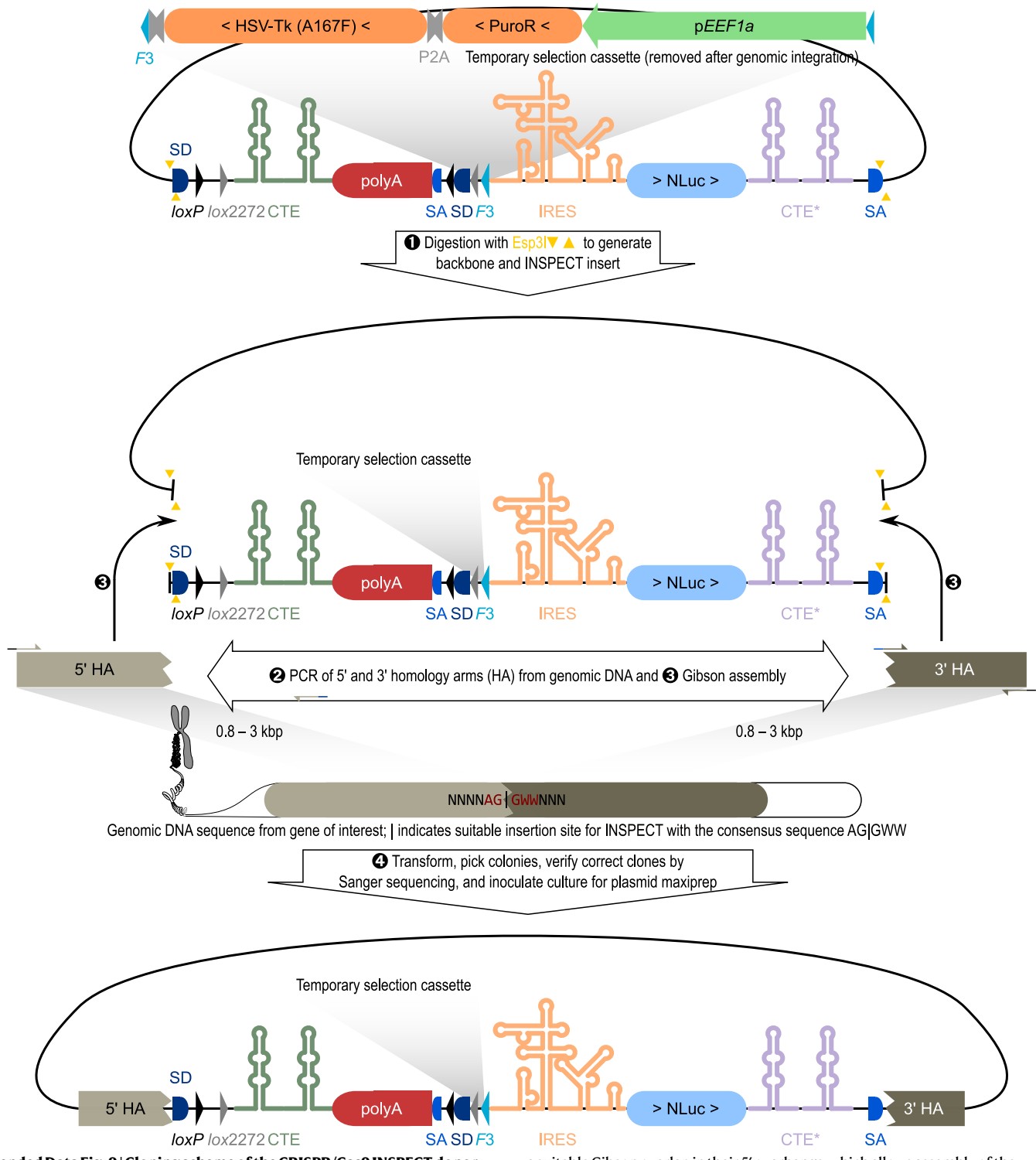

**Extended Data Fig. 9 | Cloning scheme of the CRISPR/Cas9 INSPECT donor plasmid for homology arm insertion.** The INSPECT donor plasmid is digested with the Type II S restriction enzyme Esp3I (isoschizomer: BsmBI) for a scar-free insertion of homology arms (HA). The insertion site within the gene of interest is chosen according to the consensus sequence of AG|GWW. The 5′ and 3′ HA of a gene of interest are amplified from genomic DNA with primers, including a suitable Gibson overlap in their 5′ overhangs, which allows assembly of the backbone fragment, the INSPECT insert, and both homology arms in a single Gibson isothermal assembly. The assembly mix is used for transformation of competent *E.coli*. Sequence integrity of the homology arms must be verified via Sanger sequencing.

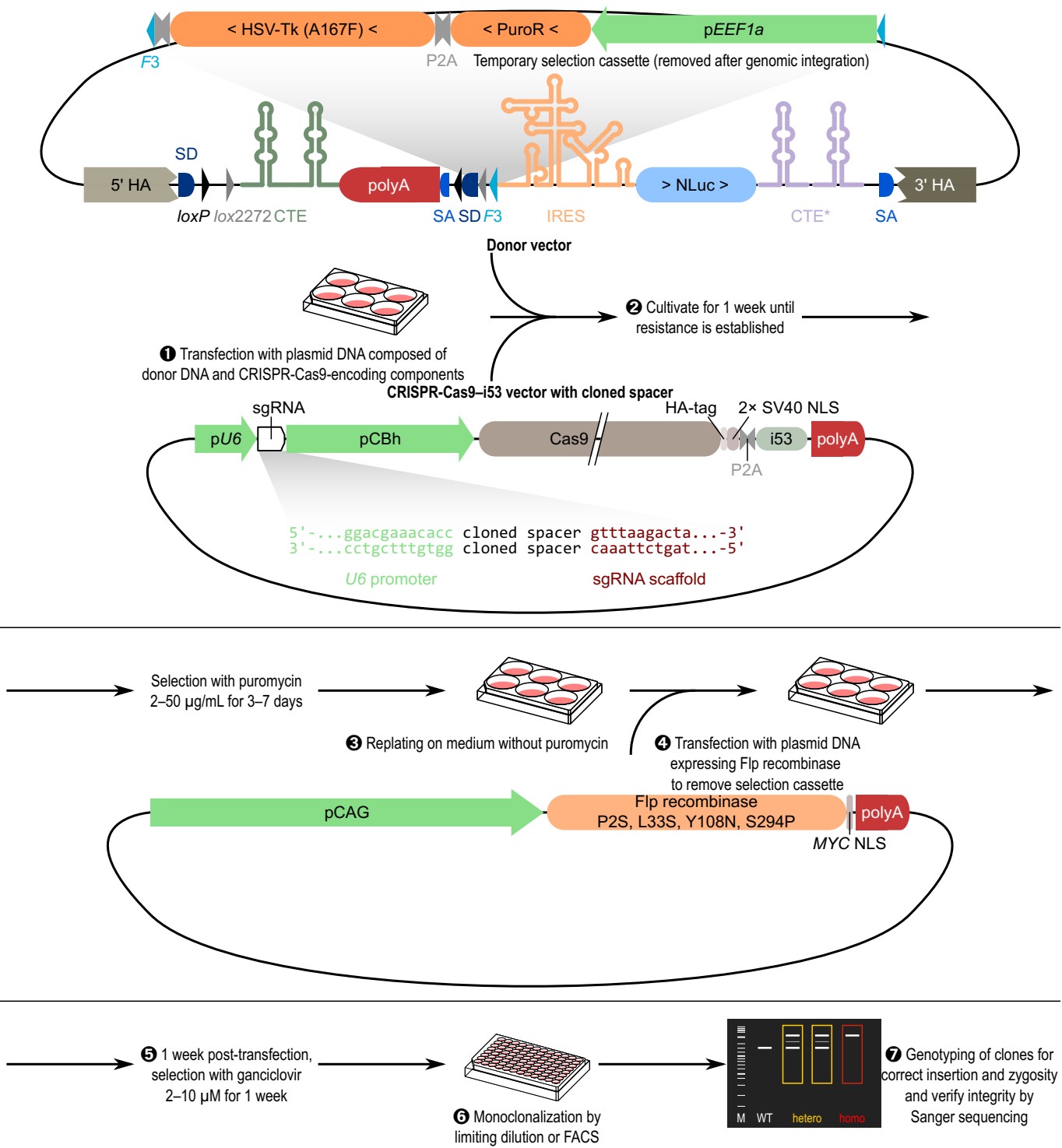

**Extended Data Fig. 10 | General workflow of establishing INSPECT-tagged cells by positive and negative selection.** Cells are transfected or electroporated for delivery of the CRISPR-Cas9–i53 plasmid as well as the INSPECT donor plasmid. Cells are then cultured until puromycin resistance is established. After up to one week of puromycin selection (concentration depends on cell line), cells are transfected or electroporated with an expression plasmid coding for CAG-promoter-driven mesostable Flp recombinase (P2S, L33S, Y108N, S294P) to remove the selection cassette (PuroR and HSV-TK) from the genome. Before monoclonalization, cells were counter-selected using ganciclovir (2-10 μM). The medium must be changed regularly to avoid the accumulation of toxic ganciclovir triphosphate. Cells are finally genotyped for determination of zygosity, and sequence integrity of the splice sites is verified via Sanger sequencing.

# Reporting Summary

Nature Research wishes to improve the reproducibility of the work that we publish. This form provides structure for consistency and transparency in reporting. For further information on Nature Research policies, see Authors & Referees and the Editorial Policy Checklist.

## Statistics

For all statistical analyses, confirm that the following items are present in the figure legend, table legend, main text, or Methods section.

| n/a | Confirmed | |
|---|---|---|
| ☐ | ☒ | The exact sample size (*n*) for each experimental group/condition, given as a discrete number and unit of measurement |
| ☐ | ☒ | A statement on whether measurements were taken from distinct samples or whether the same sample was measured repeatedly |
| ☐ | ☒ | The statistical test(s) used AND whether they are one- or two-sided<br>*Only common tests should be described solely by name; describe more complex techniques in the Methods section.* |
| ☒ | ☐ | A description of all covariates tested |
| ☐ | ☒ | A description of any assumptions or corrections, such as tests of normality and adjustment for multiple comparisons |
| ☐ | ☒ | A full description of the statistical parameters including central tendency (e.g. means) or other basic estimates (e.g. regression coefficient) AND variation (e.g. standard deviation) or associated estimates of uncertainty (e.g. confidence intervals) |
| ☐ | ☒ | For null hypothesis testing, the test statistic (e.g. *F*, *t*, *r*) with confidence intervals, effect sizes, degrees of freedom and *P* value noted<br>*Give P values as exact values whenever suitable.* |
| ☒ | ☐ | For Bayesian analysis, information on the choice of priors and Markov chain Monte Carlo settings |
| ☒ | ☐ | For hierarchical and complex designs, identification of the appropriate level for tests and full reporting of outcomes |
| ☒ | ☐ | Estimates of effect sizes (e.g. Cohen's *d*, Pearson's *r*), indicating how they were calculated |

*Our web collection on statistics for biologists contains articles on many of the points above.*

## Software and code

Policy information about availability of computer code

| Data collection | Geneious Prime 2019, MikroWin 2000 4.34, BD FACSDiva 6.1.3, QuantStudio 12K Flex |
|---|---|
| Data analysis | PRISM 9, Geneious Prime 2019, Fiji ImageJ (1.52p), FACSDiva (6.1.3), Image Lab software (v6.1.0 build 7, Bio-Rad), QuantStudio 12K Flex (v1.4), Human Splice Finder (v3.1), NetGene2 (v2.42) |

For manuscripts utilizing custom algorithms or software that are central to the research but not yet described in published literature, software must be made available to editors/reviewers. We strongly encourage code deposition in a community repository (e.g. GitHub). See the Nature Research guidelines for submitting code & software for further information.

## Data

Policy information about availability of data

All manuscripts must include a data availability statement. This statement should provide the following information, where applicable:

- Accession codes, unique identifiers, or web links for publicly available datasets
- A list of figures that have associated raw data
- A description of any restrictions on data availability

Source data are provided with this paper. All other data supporting the findings of this study are available from the corresponding author on request.

# Field-specific reporting

Please select the one below that is the best fit for your research. If you are not sure, read the appropriate sections before making your selection.

☒ Life sciences    ☐ Behavioural & social sciences    ☐ Ecological, evolutionary & environmental sciences

# Life sciences study design

All studies must disclose on these points even when the disclosure is negative.

| | |
|---|---|
| Sample size | Preliminary luciferase-based experiments showed that there was only small variation over independent replicates, so we decided for n>=3. |
| Data exclusions | Excluded data points were indicated in the raw data tables with a *. |
| Replication | All experiments were successfully replicated with independent biological samples. |
| Randomization | This technical report used cell culture as the primary method. Cell culture wells were allocated to the different treatment groups specified by the particular experiment without determining any specific property of the cells in a given well for selecting a specific condition. |
| Blinding | The experiments were performed, if possible, with master mixes and with multichannel pipettes to exclude unconscious biases during sample preparation. However, the individual human experimenter was not blinded for the conditions as this was not feasible. |

# Reporting for specific materials, systems and methods

We require information from authors about some types of materials, experimental systems and methods used in many studies. Here, indicate whether each material, system or method listed is relevant to your study. If you are not sure if a list item applies to your research, read the appropriate section before selecting a response.

### Materials & experimental systems

| n/a | Involved in the study |
|---|---|
| ☐ | ☒ Antibodies |
| ☐ | ☒ Eukaryotic cell lines |
| ☒ | ☐ Palaeontology |
| ☒ | ☐ Animals and other organisms |
| ☒ | ☐ Human research participants |
| ☒ | ☐ Clinical data |

### Methods

| n/a | Involved in the study |
|---|---|
| ☒ | ☐ ChIP-seq |
| ☐ | ☒ Flow cytometry |
| ☒ | ☐ MRI-based neuroimaging |

## Antibodies

| | |
|---|---|
| Antibodies used | IL-2 Monoclonal Antibody (MQ1-17H12): https://www.thermofisher.com/antibody/product/IL-2-Antibody-clone-MQ1-17H12-Monoclonal/14-7029-81<br>IL-2 Polyclonal Antibody, Biotin: https://www.thermofisher.com/antibody/product/IL-2-Antibody-Polyclonal/13-7028-85 |
| Validation | IL-2 Monoclonal Antibody (MQ1-17H12): Validation shown in 'Advanced verifcation' on https://www.thermofisher.com/antibody/product/IL-2-Antibody-clone-MQ1-17H12-Monoclonal/14-7029-81<br>IL-2 Polyclonal Antibody, Biotin: Validation shown in Bounab, et al. Dynamic single-cell phenotyping of immune cells using the microfluidic platform DropMap. (doi:10.1038/s41596-020-0354-0) |

## Eukaryotic cell lines

Policy information about cell lines

| | |
|---|---|
| Cell line source(s) | HEK293T cells (ECACC: 12022001, Sigma-Aldrich), Jurkat E6.1 cells (ECACC 88042803), H9 hESC (WiCELL Research Institute) |
| Authentication | All cells were purchased from trusted vendors. |
| Mycoplasma contamination | All cell lines were tested for mycoplasma contamination using MycoAlertTM Mycoplasma Detection Kit (LT07-318, Lonza). In addition, all cell lines were tested every 3 months for contamination by Hoechst 3334, which visualizes extranuclear speckles in case of contamination. All results shown in the manuscript were from cells tested negative for mycoplasma. |
| Commonly misidentified lines<br>(See ICLAC register) | No commonly misidentified cell lines were used in this study. |

# Flow Cytometry

## Plots

Confirm that:

☒ The axis labels state the marker and fluorochrome used (e.g. CD4-FITC).

☒ The axis scales are clearly visible. Include numbers along axes only for bottom left plot of group (a 'group' is an analysis of identical markers).

☐ All plots are contour plots with outliers or pseudocolor plots.

☒ A numerical value for number of cells or percentage (with statistics) is provided.

## Methodology

| | |
|---|---|
| Sample preparation | Adherent cells from 48-well plates transfection were dissociated with 200 μl Accutase™ and centrifuged at 200 r.c.f. for 5 min at RT, and cell pellets were resuspended in ice-cold 500 μl DPBS containing 0.4% PFA. After an additional washing step with DPBS, the fixated resuspended cells were transferred into conical 5 ml polystyrene round-bottom tubes, including a cell-strainer cap, and were kept on ice. |
| Instrument | BD FACSaria II (BD Biosciences) |
| Software | BD FACSDiva Software (Version 6.1.3, BD Biosciences) |
| Cell population abundance | For FACS analysis, at least 50,000 events were recorded per condition. The smallest fraction of analyzed cells was >5% of total events. |
| Gating strategy | The main cell population was gated according to their forward scatter and sideward scatter. Afterward, single cells were chosen according to their FSC-A and FSC-W. Subsequently, the transfected cells were gated according to their green fluorescence (530 nm) or red fluorescence (586 nm). These gates were not changed for all subsequent experimental conditions. |

☒ Tick this box to confirm that a figure exemplifying the gating strategy is provided in the Supplementary Information.

