## [Peer Review File · Nature Cell Biology]

Peer Review Information

Journal: Nature Cell Biology

Manuscript Title: Intron-encoded cistronic transcripts for minimally-invasive monitoring of coding and non-coding RNAs

Corresponding author name: Gil Gregor Westmeyer

Reviewer Comments & Decisions:

Decision Letter, initial version:

Subject: Decision on Nature Cell Biology submission NCB-W45968A
Message:

*Please delete the link to your author homepage if you wish to forward this email to co-authors.

Dear Professor Westmeyer,

Your manuscript, "Intron-encoded cistronic transcripts for minimally-invasive monitoring of (non-)coding RNAs", has now been seen by 3 referees, who are experts in noncoding RNAs (referees 1 and 3) and splicing (referee 2). As you will see from their comments (attached below) they find this work of potential interest, but have raised substantial concerns, which in our view would need to be addressed with considerable revisions before we can consider publication in Nature Cell Biology.

Nature Cell Biology editors discuss the referee reports in detail within the editorial team, including the chief editor, to identify key referee points that should be addressed with priority, and requests that are overruled as being beyond the scope of the current study. To guide the scope of the revisions, I have listed these points below.

I should stress that the referees' concerns point to a premature dataset and these points would need to be addressed with experiments and data, and reconsideration of the study for this journal and re-engagement of referees would depend on strength of these revisions.

In particular, it would be essential to:

a) demonstrate that INSPECT is really a minimally invasive method, as noted by:

Referee 1:

* Splicing of host gene.

While the elements of INSPECT were screened to maintain and not interfere of the host gene expression, the splicing status and expression patterns of genes to which INSPECT was inserted (IL2 and NEAT1) is not shown. No RT-qPCR comparing expression of host genes in INSPECT inserted vs. WT is shown. In this regard, another analysis which can be done with the collected NEAT1 smFISH data, is to

2compare the number of paraspeckles per cell in INSPECT inserted vs. WT cells (beyond the fraction of paraspeckle containing cells presented in fig. 3e).

Referee 2:

-It would be helpful to compare IL2 protein expression in Fig. 2d to Jurkat cells that have not undergone reporter insertion. This figure should also include formal statistical comparison (I realize the difference between non-activated and activated cells is very clearly different).

-Are the author's certain that accumulation of 131I- would not affect T-cell function? I realize this technique is being used a method to read out IL2 expression and presented as a proof-of-concept methodology, but if the protein encoded by the reporter affected cell function, then this would be a limitation of the specific approach presented.

Referee 3:

(1) The authors should be more careful when they claim that INSPECT acts "without altering the target of interest at either the RNA or protein level". The authors are adding an extra intron to the gene, which at minimum means an extra exon-junction complex has been deposited on the transcript. This is even more relevant when INSPECT is used to study NEAT1 as NEAT1 is normally an unspliced RNA. The authors' technique has now turned NEAT1 into a spliced RNA which may have unintended effects. Is it possible to add INSPECT into naturally present introns rather than adding a new intron?(3) Figure 2: The authors need to more completely demonstrate that insertion of the INSPECT element does not alter IL-2 transcription/RNA processing, etc. An ELISA is shown in Fig 2d, but such data do not address if the IL-2 RNA levels have been affected or if the dynamics of transcription induction have been changed. One could imagine that transcription induction might be slower as the gene length has been increased. It is also possible that some amount of the IL2 transcript is not fully spliced.

(6) Fig 3: To echo my points from #3 above, the authors need to more closely look at NEAT1 transcript levels in cells and whether INSPECT has changed NEAT1 at the RNA level in any way. They have only measured the number of cells with paraspeckles. They have not measured whether NEAT1 levels or processing is at all altered by the addition of the INSPECT elements. In Fig 3e, if INSPECT elements are not present, what do endogenous NEAT1 levels look like after CRISPRi? Same level of knockdown as when INSPECT elements are present? These are the sorts of important controls that are lacking in the current manuscript.

b) demonstrate the sensitivity of INSPECT, at what expression levels INSPECT reaches saturation, and sensitivity in an inducible manner and in a single-cell resolution, as noted by Referee 1:

*The resolution of INSPECT is not sufficiently explored and demonstrated

2As this is a method paper, it is imperative to delineate the upper and lower bound of systems sensitivity, i.e. the resolution of the reporter system is not properly demonstrated.

The authors nicely demonstrate that INSPECT properly reports the NEAT1 lncRNA, which is a rather abundant lncRNA. As this reporter system is promoted as suitable for lncRNAs of which many are expressed at very low levels (few copies per cell), it remains unanswered whether it is suitable for lncRNAs of low expression (whether such transcripts signal would be below threshold).

Furthermore, at what expression levels INSPECT reaches saturation is not demonstrated. This may cause a reduction in expression levels of a highly expressed transcript to generate an undetected or skewed signal (if after transcript reduction its levels remain near saturated levels of the reporter system).

Throughout the paper, three different reporter systems are employed within INSPECT (listed in order of increasing sensitivity): fluorescent proteins, luminescent proteins (luciferase) and radioactive decay (imported intracellularly by the reporter gene). It would be convincing to test the sensitivity of INSPECT with at least one of these reporter systems in an inducible manner, and at the single-cell rather than bulk level. Specifically, how does the proportion of expressing cells as detectable by FISH compare to the proportion of cells where the fluorescent protein is detectable?

c) provide more details about the parameters of the method, including insertion size/location and lag/persistence of the signal

Referee 1:

* Ease and efficiency of introducing INSPECT.

While INSPECT provides a direct and convenient route to assess its host genes expression levels relative to other methods (RT-qPCR, smFISH etc.), it requires establishing the system which can be non-trivial in terms of time and complexity of genome editing. Establishing INSPECT, I assume, entails cloning of INSPECT with homology arms, knocking-in the INSPECT and selecting/screening for correctly inserted clones. The size of the inserted element is particularly important for efficiency. Please specify the sizes of the inserts in each of the designs in all figures. Also, please elaborate on the time it takes from the beginning of the cloning process until single colonies are obtained, and on the efficiency of knocking-in INSPECT into cells (% knock-ins inserted at the correct locus).

* Lag time and persistence time of reporter signal.

INSPECT uses a reporter protein as a proxy for the levels of its host transcript which may complicate the interpreting the signal on the temporal dimension. Specifically the lag time and the persistence time of the reporter are crucial to determine the systems suitability for a research interest.

(i) Lag time: what is the lag time between the transcription until an increase in signal is detected (dependent on export and translation time).

(ii) Persistence time: How long after transcription is stalled does the signal persist (this may reflect both the stability of the INSPECT RNA and of the protein which it encodes).

Please perform pulse-chase experiments in the system to characterize the systems lag time and persistence time.

Referee 2:

-One major issue not described in detail is how the site of insertion of the reporter may influence reporter expression as well as parent gene expression. It seems possible that if the reporter were inserted next to an endogenous splice donor/acceptor site and/or regulatory element for a specific RNA isoform or splicing of the parent gene, that the reporter could influence splicing of the parent gene. Given these concerns, it would be helpful for the authors to (i) explain the rationale for the site of insertion of the reporter in exon 3 of IL2, and (ii) demonstrate that the reporter could be inserted into other sites within the IL2 locus (if that is possible) for similar effect. As a minor point, it would be helpful to modify Figure 2a to show exactly where in the IL2 locus the promoter was inserted.

Referee 3:

(2) Fig 1c: The exact location of the Cre based elements relative to the rest of the construct elements is somewhat ambiguous based on the writing. The authors should combine the drawings in Fig 1b/1c so that this is much clearer to the reader. It is also unclear at this point in the main text why the authors want the INSPECT method to have the potential to be like a gene trap as the introduction stresses the desire to make a “minimally invasive reporter system”, not any sort of gene knockout.

d) All other referee concerns pertaining to strengthening existing data, providing controls, methodological details, clarifications and textual changes, should also be addressed.

e) Finally please pay close attention to our guidelines on statistical and methodological reporting (listed below) as failure to do so may delay the reconsideration of the revised manuscript. In particular please provide:

- a Supplementary Table including all numerical source data in Excel format, with data for different figures provided as different sheets within a single Excel file. The file should include source data giving rise to graphical representations and statistical descriptions in the paper and for all instances where the

figures present representative experiments of multiple independent repeats, the source data of all repeats should be provided.

We would be happy to consider a revised manuscript that would satisfactorily address these points, unless a similar paper is published elsewhere, or is accepted for publication in Nature Cell Biology in the meantime.

- ensure that it conforms to our format instructions and publication policies (see below and <https://www.nature.com/nature/for-authors>).
- provide a point-by-point rebuttal to the full referee reports verbatim, as provided at the end of this letter.
- provide the completed Reporting Summary (found here <https://www.nature.com/documents/nr-reporting-summary.pdf>). This is essential for reconsideration of the manuscript will be available to editors and referees in the event of peer review. For more information see <http://www.nature.com/authors/policies/availability.html> or contact me.

When submitting the revised version of your manuscript, please pay close attention to our [Digital Image Integrity Guidelines](https://www.nature.com/nature-research/editorial-policies/image-integrity). and to the following points below:

Nature Cell Biology is committed to improving transparency in authorship. As part of our efforts in this direction, we are now requesting that all authors identified as 'corresponding author' on published papers create and link their Open Researcher and Contributor Identifier (ORCID) with their account on the Manuscript Tracking System (MTS), prior to acceptance. ORCID helps the scientific community achieve unambiguous attribution of all scholarly contributions. You can create and link your ORCID from the home page of the MTS by clicking on 'Modify my Springer Nature account'. For more information please visit www.springernature.com/orcid.

This journal strongly supports public availability of data. Please place the data used in your paper into a public data repository, or alternatively, present the data as Supplementary Information. If data can only be shared on request, please explain why in your Data Availability Statement, and also in the correspondence with your editor. Please note that for some data types, deposition in a public repository is mandatory - more information on our data deposition policies and available repositories appears below.

[REDACTED]

We would like to receive a revised submission within six months.

We hope that you will find our referees' comments, and editorial guidance helpful. Please do not hesitate to contact me if there is anything you would like to discuss.

Best wishes,

Jie Wang

Jie Wang, PhD
Senior Editor
Nature Cell Biology

6Tel: +44 (0) 207 843 4924
email: jie.wang@nature.com

Reviewers' Comments:

Reviewer #1:

Remarks to the Author:

In this paper, Truong et al. developed an intron-encoded reporter system for tracking gene expression (INSPECT). The independently translated intron encoded aspect of INSPECT renders it suitable as a reporter system for non-coding RNAs, a challenge which suffers from a paucity of such methods for these RNA species. The authors nicely demonstrate the system in two different settings, signal transduction and readout of lncRNA expression.

Several features of this system render it a nice and potentially widely useful system for tracking expression of transcripts. These include the system's modularity, its largely minimal effect on its host gene and the rather direct readout of reporter.

As the papers' novelty is a method – where the devil is in the (technical) details – the majority of the concerns relate to information that is not presented or is not addressed regarding the method.

Major points

*The resolution of INSPECT is not sufficiently explored and demonstrated

As this is a method paper, it is imperative to delineate the upper and lower bound of systems sensitivity, i.e. the resolution of the reporter system is not properly demonstrated.

The authors nicely demonstrate that INSPECT properly reports the NEAT1 lncRNA, which is a rather abundant lncRNA. As this reporter system is promoted as suitable for lncRNAs of which many are expressed at very low levels (few copies per cell), it remains unanswered whether it is suitable for lncRNAs of low expression (whether such transcripts signal would be below threshold).

Furthermore, at what expression levels INSPECT reaches saturation is not demonstrated. This may cause a reduction in expression levels of a highly expressed transcript to generate an undetected or skewed signal (if after transcript reduction its levels remain near saturated levels of the reporter system).

Throughout the paper, three different reporter systems are employed within INSPECT (listed in order of increasing sensitivity): fluorescent proteins, luminescent proteins (luciferase) and radioactive decay (imported intracellularly by the reporter gene). It would be convincing to test the sensitivity of INSPECT

7with at least one of these reporter systems in an inducible manner, and at the single-cell rather than bulk level. Specifically, how does the proportion of expressing cells as detectable by FISH compare to the proportion of cells where the fluorescent protein is detectable?

* Splicing of host gene.

While the elements of INSPECT were screened to maintain and not interfere of the host gene expression, the splicing status and expression patterns of genes to which INSPECT was inserted (IL2 and NEAT1) is not shown. No RT-qPCR comparing expression of host genes in INSPECT inserted vs. WT is shown. In this regard, another analysis which can be done with the collected NEAT1 smFISH data, is to compare the number of paraspeckles per cell in INSPECT inserted vs. WT cells (beyond the fraction of paraspeckle containing cells presented in fig. 3e).

* Ease and efficiency of introducing INSPECT.

While INSPECT provides a direct and convenient route to assess its host genes expression levels relative to other methods (RT-qPCR, smFISH etc.), it requires establishing the system which can be non-trivial in terms of time and complexity of genome editing. Establishing INSPECT, I assume, entails cloning of INSPECT with homology arms, knocking-in the INSPECT and selecting\screening for correctly inserted clones. The size of the inserted element is particularly important for efficiency. Please specify the sizes of the inserts in each of the designs in all figures. Also, please elaborate on the time it takes from the beginning of the cloning process until single colonies are obtained, and on the efficiency of knocking-in INSPECT into cells (% knock-ins inserted at the correct locus).

* Lag time and persistence time of reporter signal.

INSPECT uses a reporter protein as a proxy for the levels of its host transcript which may complicate the interpreting the signal on the temporal dimension. Specifically the lag time and the persistence time of the reporter are crucial to determine the systems suitability for a research interest.

(i) Lag time: what is the lag time between the transcription until an increase in signal is detected (dependent on export and translation time).

(ii) Persistence time: How long after transcription is stalled does the signal persist (this may reflect both the stability of the INSPECT RNA and of the protein which it encodes).

Please perform pulse-chase experiments in the system to characterize the systems lag time and persistence time.

Minor points

* Supplementary figure 3d. “24 h post-induction with different doxycycline (Dox) concentrations, msfGFP was expressed in a doxycycline-dependent manner, while mScarlet-I expression remained relatively stable.” In supplementary figure 3, there seems to be non-negligible difference in the fluorescent intensity of the constitutively expressed mScarlet. Are these differences across all samples statistically insignificant, such that you can say they are “relatively stable”?

* Supplementary figure 3d-e. What stands out from both these figure is the considerable difference in sensitivity of RT-qPCR detection of the reporter RNA, relative to that of the fluorescent signal from the reporter. This is especially true for conditions where Dox induction was at low concentrations (or no Dox addition). This seems to call into question the sensitivity of INSPECT for transcripts of low expression. (This feeds off the first listed major point).

* Suggestion for supplementary figure 4b: add the different images of this same cell line before Cre activation, for reference.

Reviewer #2:

Remarks to the Author:

This manuscript presents a very interesting to approach to monitor RNA expression through use of reporters expressed via synthetic intron sequences embedded within a gene where the reporter expression is dependent on the gene’s promoter. The stated benefit of this approach over other methods of tagging RNA or genes is that the reporter does not alter the coding sequence of the gene of interest.

-One major issue not described in detail is how the site of insertion of the reporter may influence reporter expression as well as parent gene expression. It seems possible that if the reporter were inserted next to an endogenous splice donor/acceptor site and/or regulatory element for a specific RNA isoform or splicing of the parent gene, that the reporter could influence splicing of the parent gene. Given these concerns, it would be helpful for the authors to (i) explain the rationale for the site of insertion of the reporter in exon 3 of IL2, and (ii) demonstrate that the reporter could be inserted into other sites within the IL2 locus (if that is possible) for similar effect. As a minor point, it would be helpful to modify Figure 2a to show exactly where in the IL2 locus the promoter was inserted.

-It would be helpful to compare IL2 protein expression in Fig. 2d to Jurkat cells that have not undergone reporter insertion. This figure should also include formal statistical comparison (I realize the difference between non-activated and activated cells is very clearly different).

-Are the author’s certain that accumulation of 131I- would not affect T-cell function? I realize this technique is being used a method to read out IL2 expression and presented as a proof-of-concept

9methodology, but if the protein encoded by the reporter affected cell function, then this would be a limitation of the specific approach presented.

-The manuscript title could simply state “RNAs” and not “(non)-coding RNAs” (because this latter phrasing is also confusing; I realized after reading the abstract that the authors have used this phrase to attempt to encapsulate both protein-coding as well as non-protein coding RNAs but this is not clear when reading the manuscript title alone).

Reviewer #3:

Remarks to the Author:

The authors have created a new way to track gene expression by encoding translatable elements in an intron they add to the gene. The design approach is overall straightforward and the authors have provided some data to suggest it works to look at endogenous transcripts. However, the authors have provided minimal data so it is hard to judge at this point how robust the system is and if it truly is a minimally invasive approach. I have made some specific suggestions below as to the sorts of data that should be included in a revised manuscript.

(1) The authors should be more careful when they claim that INSPECT acts “without altering the target of interest at either the RNA or protein level”. The authors are adding an extra intron to the gene, which at minimum means an extra exon-junction complex has been deposited on the transcript. This is even more relevant when INSPECT is used to study NEAT1 as NEAT1 is normally an unspliced RNA. The authors’ technique has now turned NEAT1 into a spliced RNA which may have unintended effects. Is it possible to add INSPECT into naturally present introns rather than adding a new intron?

(2) Fig 1c: The exact location of the Cre based elements relative to the rest of the construct elements is somewhat ambiguous based on the writing. The authors should combine the drawings in Fig 1b/1c so that this is much clearer to the reader. It is also unclear at this point in the main text why the authors want the INSPECT method to have the potential to be like a gene trap as the introduction stresses the desire to make a “minimally invasive reporter system”, not any sort of gene knockout.

(3) Figure 2: The authors need to more completely demonstrate that insertion of the INSPECT element does not alter IL-2 transcription/RNA processing, etc. An ELISA is shown in Fig 2d, but such data do not address if the IL-2 RNA levels have been affected or if the dynamics of transcription induction have been changed. One could imagine that transcription induction might be slower as the gene length has been increased. It is also possible that some amount of the IL2 transcript is not fully spliced.

10(4) Key details of the INSPECT intron sometimes change between panels, especially the reporter gene that is translated from the intron (NLucPEST in Fig 1, NIS in Fig 2, SecNLuc in Fig 3, mScarlet in Fig S2, etc). It is not clear why the authors have used such a variety of reporters across panels. Perhaps it is because the system is very robust and any combination works, or it could be that the system is hard to make work well and only certain designs work in certain contexts. It is not possible to judge this important point based on the data currently presented.

(5) The use of gag-PCP to detect INSPECT transcripts does not seem to be much better than current aptamer based approaches. Yes, a different transcript than the mRNA/ncRNA of interest is being detected, but you still have to change the cells quite a bit by putting gag-PCP into them.

(6) Fig 3: To echo my points from #3 above, the authors need to more closely look at NEAT1 transcript levels in cells and whether INSPECT has changed NEAT1 at the RNA level in any way. They have only measured the number of cells with paraspeckles. They have not measured whether NEAT1 levels or processing is at all altered by the addition of the INSPECT elements. In Fig 3e, if INSPECT elements are not present, what do endogenous NEAT1 levels look like after CRISPRi? Same level of knockdown as when INSPECT elements are present? These are the sorts of important controls that are lacking in the current manuscript.

(7) The Discussion section is written in an unusual format. It is largely bullet points with one sentence per paragraph.

(8) The points in the Discussion are overstated. For example, the authors discuss how INSPECT allows for “high-throughput”, “multi-time point assessment”, and “multiplexing” but the authors have not provided data proving any of these points in the current manuscript.

(9) The full, exact sequences of the complete INSPECT cassettes need to be included in the methods section so that this work can be replicated in the future by any interested party.

Minor points:

(1) Yellow is very difficult to see in Fig 2d.

(2) P. 6: The authors write that INSPECT “offers the possibility to create a full knockout of the tagged gene” but it is not really a full knockout but instead the creation of a shorten, prematurely terminated transcript.

ABSTRACT AND MAIN TEXT – please follow the guidelines that are specific to the format of your manuscript, as listed in our Guide to Authors (http://www.nature.com/ncb/pdf/ncb_gta.pdf) Briefly, Nature Cell Biology Articles, Resources and Technical Reports have 3500 words, including a 150 word abstract, and the main text is subdivided in Introduction, Results, and Discussion sections. Nature Cell

Biology Letters have up to 2500 words, including a 180 word introductory paragraph (abstract), and the text is not subdivided in sections.

Methods should be written concisely, but should contain all elements necessary to allow interpretation and replication of the results. As a guideline, Methods sections typically do not exceed 3,000 words. The Methods should be divided into subsections listing reagents and techniques. When citing previous methods, accurate references should be provided and any alterations should be noted. Information

must be provided about: antibody dilutions, company names, catalogue numbers and clone numbers for monoclonal antibodies; sequences of RNAi and cDNA probes/primers or company names and catalogue numbers if reagents are commercial; cell line names, sources and information on cell line identity and authentication. Animal studies and experiments involving human subjects must be reported in detail, identifying the committees approving the protocols. For studies involving human subjects/samples, a statement must be included confirming that informed consent was obtained. Statistical analyses and information on the reproducibility of experimental results should be provided in a section titled "Statistics and Reproducibility".

All Nature Cell Biology manuscripts submitted on or after March 21 2016 must include a Data availability statement as a separate section after Methods but before references, under the heading "Data Availability". . For Springer Nature policies on data availability see <http://www.nature.com/authors/policies/availability.html>; for more information on this particular policy see <http://www.nature.com/authors/policies/data/data-availability-statements-data-citations.pdf>. The Data availability statement should include:

- Accession codes for primary datasets (generated during the study under consideration and designated as "primary accessions") and secondary datasets (published datasets reanalysed during the study under consideration, designated as "referenced accessions"). For primary accessions data should be made public to coincide with publication of the manuscript. A list of data types for which submission to community-endorsed public repositories is mandated (including sequence, structure, microarray, deep sequencing data) can be found here <http://www.nature.com/authors/policies/availability.html#data>.
- Unique identifiers (accession codes, DOIs or other unique persistent identifier) and hyperlinks for datasets deposited in an approved repository, but for which data deposition is not mandated (see here for details <http://www.nature.com/sdata/data-policies/repositories>).
- At a minimum, please include a statement confirming that all relevant data are available from the authors, and/or are included with the manuscript (e.g. as source data or supplementary information), listing which data are included (e.g. by figure panels and data types) and mentioning any restrictions on availability.
- If a dataset has a Digital Object Identifier (DOI) as its unique identifier, we strongly encourage including this in the Reference list and citing the dataset in the Methods.

We recommend that you upload the step-by-step protocols used in this manuscript to the Protocol Exchange. More details can be found at www.nature.com/protocolexchange/about.

All imaging data should be accompanied by scale bars, which should be defined in the legend. Cropped images of gels/blots are acceptable, but need to be accompanied by size markers, and to retain visible background signal within the linear range (i.e. should not be saturated). The boundaries of panels with low background have to be demarked with black lines. Splicing of panels should only be considered if unavoidable, and must be clearly marked on the figure, and noted in the legend with a statement on whether the samples were obtained and processed simultaneously. Quantitative comparisons between samples on different gels/blots are discouraged; if this is unavoidable, it should only be performed for samples derived from the same experiment with gels/blots were processed in parallel, which needs to be stated in the legend.

The total number of Supplementary Figures (not including the “unprocessed scans” Supplementary Figure) should not exceed the number of main display items (figures and/or tables (see our Guide to Authors and March 2012 editorial <http://www.nature.com/ncb/authors/submit/index.html#suppinfo>; <http://www.nature.com/ncb/journal/v14/n3/index.html#ed>). No restrictions apply to Supplementary Tables or Videos, but we advise authors to be selective in including supplemental data.

GUIDELINES FOR EXPERIMENTAL AND STATISTICAL REPORTING

REPORTING REQUIREMENTS – We are trying to improve the quality of methods and statistics reporting in our papers. To that end, we are now asking authors to complete a reporting summary that collects information on experimental design and reagents. The Reporting Summary can be found here <https://www.nature.com/documents/nr-reporting-summary.pdf> If you would like to reference the guidance text as you complete the template, please access these flattened versions at <http://www.nature.com/authors/policies/availability.html>.

We strongly recommend the presentation of source data for graphical and statistical analyses as a separate Supplementary Table, and request that source data for all independent repeats are provided when representative experiments of multiple independent repeats, or averages of two independent experiments are presented. This supplementary table should be in Excel format, with data for different figures provided as different sheets within a single Excel file. It should be labelled and numbered as one of the supplementary tables, titled “Statistics Source Data”, and mentioned in all relevant figure legends.

Author Rebuttal to Initial comments

General reply to all Reviewers (1st revision)

We thank the Reviewers for their constructive and thorough feedback on our manuscript. We have now conducted a series of new experiments to address all requests in detail.

As a short overview of the improvements and expansions of the figures and analyses, we have included a short list here, ordered by the categories highlighted in the Editor's email.

We have now confirmed the minimal invasiveness of INSPECT for 3 genes in 3 cell lines according to the following parameters.

For the coding gene *IL2*:

- RT-PCR and Sanger sequencing for the insertion site → **new Fig. 2b,c**
- RT-qPCR of *IL2* mRNA and quantification of secreted IL2 protein in comparison to INSPECT RNA and NLuc luminescence of INSPECT_{*IL2*:SigP-NLuc} cells → **new Fig. 2d and related text**
- Time courses during TCR activation and 'pulse-chase' after TCR activation → **new Fig. 2e,f**
- bioluminescence imaging of INSPECT_{*IL2*:NLuc} cells with cellular resolution → **new Fig. 2g,h**

For the lncRNA *NEAT1*:

- number of paraspeckles per cell including statistical analysis in addition to % of cells with paraspeckles → **new Fig. 3d,e, Supplementary Table 1**
- RT-PCR and Sanger sequencing for the insertion site → **new Extended Data Fig. 5a,b**
- unchanged nuclear abundance and cytoplasmic/nuclear-ratio → **new Fig. 3f,g**
- monitoring of *NEAT1* during stem cell differentiation also assessed by all main biomarkers for the different cell lineages → **new Extended Data Fig. 8**

For the lncRNA *GUARDIN*:

- RT-PCR and Sanger sequencing of insertion site → **new Extended Data Fig. 4b,c**

19We have now further characterized the sensitivity and dynamic range of INSPECT in the following models:

Monitoring of induction of *IL2*:

- 3 orders of magnitudes of *IL2* mRNA abundance can be followed by INSPECT with single-cell resolution, if required, demonstrating high dynamic range while maintaining maximum sensitivity. → **new Fig. 2e–h**
- we have also followed *IL2* induction and decrease after TCR activation over time and also with single-cell resolution → **new Fig. 2h**

Monitoring of lncRNAs:

- *NEAT1* can still be detected after strong CRISPRi suppression,
- detection of basal *NEAT1* levels with single-cell resolution using bioluminescence microscopy → **new Fig. 3j**
- the lncRNA *GUARDIN* (only ~6% abundance compared to *NEAT1*) could be detected at baseline levels and even after CRISPRi suppression. → **new Extended Data Fig. 4**

We have given the insertion site and size for all reporter lines in the respective figures and have added detailed information on our streamlined protocol to generate INSPECT reporter lines.

Please find our detailed point-by-point responses below.

We like to thank the Reviewers again for the thoughtful comments, which tremendously helped us to improve the manuscript's quality.

Reviewer #1

R1P0

In this paper, Truong et al. developed an intron-encoded reporter system for tracking gene expression (INSPECT). The independently translated intron encoded aspect of INSPECT renders it suitable as a reporter system for non-coding RNAs, a challenge which suffers from a paucity of such methods for these RNA species. The authors nicely demonstrate the system in two different settings, signal transduction and readout of lncRNA expression.

Several features of this system render it a nice and potentially widely useful system for tracking expression of transcripts. These include the system's modularity, its largely minimal effect on its host gene and the rather direct readout of reporter.

As the papers' novelty is a method – where the devil is in the (technical) details – the majority of the concerns relate to information that is not presented or is not addressed regarding the method.

Response to R0P0:

We thank the Reviewer for acknowledging the value of INSPECT for minimally invasive monitoring of gene expression, including non-coding genes.

Major points

R1P1:

The resolution of INSPECT is not sufficiently explored and demonstrated

As this is a method paper, it is imperative to delineate the upper and lower bound of systems sensitivity, i.e. the resolution of the reporter system is not properly demonstrated.

The authors nicely demonstrate that INSPECT properly reports the NEAT1 lncRNA, which is a rather abundant lncRNA. As this reporter system is promoted as suitable for lncRNAs of which many are expressed at very low levels (few copies per cell), it remains unanswered whether it is suitable for lncRNAs of low expression (whether such transcripts signal would be below threshold).

Furthermore, at what expression levels INSPECT reaches saturation is not demonstrated. This may cause a reduction in expression levels of a highly expressed transcript to generate an undetected or skewed signal (if after transcript reduction its levels remain near saturated levels of the reporter system).

Throughout the paper, three different reporter systems are employed within INSPECT (listed in order of increasing sensitivity): fluorescent proteins, luminescent proteins (luciferase) and radioactive decay (imported intracellularly by the reporter gene). It would be convincing to test the sensitivity of INSPECT with at least one of these reporter systems in an inducible manner, and at

the single-cell rather than bulk level. Specifically, how does the proportion of expressing cells as detectable by FISH compare to the proportion of cells where the fluorescent protein is detectable?

Response to R1P1:

We have followed the Reviewer's request to characterize the dynamic range of INSPECT further.

To determine the upper and lower bound of the reporter signals, we used the inducible *IL-2* gene, which covers a broad spectrum of expression strength from basically undetectable basal levels to an expression level reaching the expression of the housekeeping gene beta-actin (*ACTB*).

The **new Fig. 2d** shows a robust T-cell-receptor-mediated induction of *IL2* mRNA in the parenteral cells and the INSPECT reporter line from close to the RT-qPCR detection limit before induction over ~3 orders of magnitude after 16 hours of TCR activation.

We found that this substantial signal increase was tracked well by the NLuc luciferase signal of the INSPECT reporter, while the corresponding ELISA to determine the IL2 protein concentration secreted to the supernatant contained protein levels of $39.8 \pm 0.5 \text{ ng ml}^{-1}$ for wildtype cells and $67 \pm 1.8 \text{ ng ml}^{-1}$ for the INSPECT line.

When we then followed this end-point measurement up with longitudinal monitoring of 24 h, we found that already after 4 h of T cell stimulation, we could detect a significant induction of *IL2* expression, which was highly correlated (Pearson's $r = 1$, $P < 0.0001$) with the RT-qPCR measurements, again reaching a similar NLuc signal ($\sim 10^5$ RLU) after 16 hours of *IL2* induction (**Fig. 2e**), replicating the end-point measurement in Fig. 2d,

In addition, we could faithfully recapitulate *IL2* expression with single-cell resolution by using bioluminescence microscopy, a high signal-to-noise variant of microscopy, which allowed us to assess the variability of *IL2* induction over the cell population. While baseline *IL2* expression was not detectable, we were able to identify Jurkat cells responding already after 4 hours. (**Fig. 2g,h**).

In addition to monitoring of inducible gene expression with INSPECT, we also expanded on the characterization of the INSPECT reporter for the lncRNA *NEAT1*, which is expressed in a short and a long variant, of which the latter is a crucial RNA constituent of the cellular structures referred to as paraspeckles. We demonstrate that the INSPECT reporter line has unaltered paraspeckle abundance with respect to the fraction of cells with paraspeckles and the number of paraspeckles per cell (**Fig. 3d,e**). Suppression of *NEAT1* expression (*NEAT1*_{total} and *NEAT1*_{v2}) via CRISPRi reduces the luminescence reporter signal in line with the RNA level (**Fig. 3h,i**).

Again, we used bioluminescence microscopy to also demonstrate the correspondence between the luminescence signal from the INSPECT_{*NEAT1*:NLuc} reporter in live cells and paraspeckles, made visible via smFISH staining against *NEAT1* of the corresponding cells after fixation (**Fig. 3j**).

We furthermore challenged the sensitivity of INSPECT by applying the INSPECT reporter on the lncRNA *GUARDIN* (doi:10.1038/s41556-018-0066-7) (RP3-510D11.2-1), which exhibits a magnitude lower expression than *NEAT1_v2* (**Supplementary Fig. 2**).

Supplementary Fig. 2 | Relative expression strength of *GUARDIN* and *NEAT1* isoforms.

We still obtained a robust NLuc bioluminescence signal for baseline expression levels of *GUARDIN* as well as still after CRISPRi gene suppression by (8 fold decrease, ~0.1% of ACTB expression level; **Extended Data Fig. 4d**).

R1P2:

Splicing of host gene.

While the elements of INSPECT were screened to maintain and not interfere of the host gene expression, the splicing status and expression patterns of genes to which INSPECT was inserted (IL2 and NEAT1) is not shown. No RT-qPCR comparing expression of host genes in INSPECT inserted vs. WT is shown. In this regard, another analysis which can be done with the collected NEAT1 smFISH data, is to compare the number of paraspeckles per cell in INSPECT inserted vs. WT cells (beyond the fraction of paraspeckle containing cells presented in fig. 3e).

Response to R1P2:

We have followed the Reviewer's suggestions and have added both RT-qPCR data for *IL2* and *NEAT1* and a more detailed analysis of the abundance of paraspeckles and *NEAT1*'s subcellular localization.

We added RT-PCR and sequencing data confirming the splice status of *NEAT1* (**Extended Data Fig. 5**), *IL2* (**Fig. 2b,c**), and the new target *GUARDIN* (**Extended Data Fig. 4b,c**) after the insertion of the respective INSPECT reporter.

Furthermore, we analyzed RNA concentrations of WT and INSPECT_{SigP-NLuc} cell lines regarding *NEAT1* (**Fig. 3i**) and *GUARDIN* (**Extended Data Fig. 4d**) after CRISPRi perturbation and *IL2* (**Fig. 2d,e,f**) after TCR activation via RT-qPCR.

For *NEAT1*, we analyzed the nuclear/cytoplasmic ratio and the nuclear abundance relative to the lncRNA *MALAT1* to show that the localization of the lncRNA *NEAT1* was not altered by INSPECT insertion (**Fig. 3f,g**).

We also now provide a more detailed quantification of the paraspeckles per cell in addition to the fraction of cells with paraspeckles and confirm that both measures are not significantly altered between the INSPECT-modified cell lines and the parental wild-type cells (**Fig. 3e**, **Supplementary Table1**).

R1P3:

Ease and efficiency of introducing INSPECT.

While INSPECT provides a direct and convenient route to assess its host genes expression levels relative to other methods (RT-qPCR, smFISH etc.), it requires establishing the system which can be non-trivial in terms of time and complexity of genome editing. Establishing INSPECT, I assume, entails cloning of INSPECT with homology arms, knocking-in the INSPECT and selecting/screening for correctly inserted clones. The size of the inserted element is particularly important for efficiency. Please specify the sizes of the inserts in each of the designs in all figures. Also, please elaborate on the time it takes from the beginning of the cloning process until single colonies are obtained, and on the efficiency of knocking-in INSPECT into cells (% knock-ins inserted at the correct locus).

Response to R1P3:

We thank the Reviewer for this suggestion. We have now generated several schematics to illustrate all cloning steps of the INSPECT plasmids and our optimized procedure to establish monoclonal cell lines.

We created a convenient all-in-one CRISPR/Cas9 plasmid with improved codon optimization and nuclear localization and a strong mammalian promoter, and an optimized sgRNA expression cassette. This CRISPR plasmid is available with and without transient co-expression of i53, a 53BP1 inhibitor, which dramatically increases HDR efficiency when applied in parallel with a small molecule DNA-PKcs inhibitor (**Supplementary Fig. 3 and Materials & Methods section**).

The cloning of the homology arms is simple. It can be achieved in a single step, using a type IIS restriction enzyme to separate the INSPECT insert from the plasmid backbone and then inserting the 5' and 3' homology arms amplified via PCR from your desired cell line into the plasmid via Gibson assembly (**Supplementary Fig. 4**).

Our donor plasmid has been equipped with a transient positive/negative selection marker, which allows us to enrich the transgenic cells even at low knock-in efficiency. After that, the cassette is removed using the Fip recombinase, and the absence of the cassette is selected with ganciclovir. HSV-Tk metabolizes ganciclovir into a toxic product, and thus, cells still containing the cassette are depleted from the population. The size of the INSPECT_{SigP-NLuc} reporter cassette is 6,236 bp (without homology arms); after Fip-mediated excision of the selection cassette, the INSPECT has a length of 3,133 bp.

We also extended the section 'Generation of stable cell lines carrying the INSPECT reporter system via CRISPR/Cas9', where we describe the cell-line creation. We assume an overall time of 8 weeks to generate monoclonal INSPECT cell lines (**Supplementary Fig. 5**).

With the above methods, we achieve a KI rate of 86% (64/74) related to at least one correct transgenic allele. However, for our manuscript, we selected only homozygous clones with an overall efficiency of 47% (35/74). Although homozygosity is not necessary to obtain reporter signals, we chose only homozygous INSPECT clones to rigorously exclude potential invasive effects obscured by the remaining wild-type allele; thus, the exclusive usage of homozygous clones allowed us to reliably state the non-invasiveness of INSPECT. 7% (10/74) contained either a random integration of the construct or the selection cassette was not yet removed.

More precisely, we had a 39% (16/41) homozygous KI-rate in HEK293T cells, 48% (12/25) in Jurkat E6.1 cells, and 87.5% (7/8) in embryonic H9 stem cells using our optimized components. All of these details are also given in the Material and Methods section and can be included in an accompanying protocol.

R1P4:

Lag time and persistence time of reporter signal.

INSPECT uses a reporter protein as a proxy for the levels of its host transcript which may complicate the interpreting the signal on the temporal dimension. Specifically the lag time and the persistence time of the reporter are crucial to determine the systems suitability for a research interest.

(i) Lag time: what is the lag time between the transcription until an increase in signal is detected (dependent on export and translation time).

(ii) Persistence time: How long after transcription is stalled does the signal persist (this may reflect both the stability of the INSPECT RNA and of the protein which it encodes).

Please perform pulse-chase experiments in the system to characterize the systems lag time and persistence time.

Response to R1P4:

IL2 is one of the genes with the fastest induction known in TCR signaling and thus lends itself well to measuring the temporal resolution available with INSPECT.

(i) Lag time

We have induced WT and INSPECT_{*IL2*:SigP-NLuc} Jurkat E6.1 cells and sampled cells for RNA extraction and the corresponding supernatant for luciferase measurement every 2 hours in the first 6 h and then again after 24 hours.

After 4 hours, we found a significant increase in the NLuc reporter signal in line with the *IL2* mRNA levels for the INSPECT reporter cell line and the parental wild-type cells. Overall, the NLuc signal trajectories were highly correlated with the *IL2* mRNA time courses (Pearson's $r = 1$, $P < 0.0001$; **Fig. 2e**).

At this sampling rate, we did not find a substantial lag between mRNA levels and RLUs.

We followed up the bulk NLuc measurements with bioluminescence microscopy to obtain single-cell resolution at a higher temporal sampling rate. We found that the first responses also occurred ~4 hours after induction.

(ii) Persistence time

We have also conducted pulse-chase experiments as requested by the Reviewer:

We mildly activated TCR signaling for 6 hours before exchanging the media to a media with the cell-permeable long-term NLuc substrate endurazine. In addition, we suppressed the extracellular signals with a cell-impermeable NLuc inhibitor. Only the steady-state intracellularly active NLuc enzyme during its ER/Golgi passage is captured in this condition.

We observed a rapid decrease of the *IL2* mRNA levels and the NLuc signal after 1 hour of TCR activation, again with a high correlation of the signal time courses (Pearson's $r = 0.9847$, $P =$

0.0003), indicating that pulsed gene expression can be reliably “chased” via the INSPECT reporter signal (**Fig. 2f**).

Minor points

R1P5:

Supplementary figure 3d. “24 h post-induction with different doxycycline (Dox) concentrations, msfGFP was expressed in a doxycycline-dependent manner, while mScarlet-I expression remained relatively stable.” In supplementary figure 3, there seems to be non-negligible difference in the fluorescent intensity of the constitutively expressed mScarlet. Are these differences across all samples statistically insignificant, such that you can say they are “relatively stable”?

Response to R1P5:

We thank the Reviewer for prompting us to look more closely at the constitutively expressed mScarlet expression with high doxycycline concentrations.

We apologize for the imprecise phrasing of ‘relatively stable,’ which was meant to illustrate the substantial discrepancy between massive induction of the INSPECT-encoded msfGFP fluorescence over orders of magnitude and the comparably moderate changes in mScarlet-I expression.

We have, however, now computed a one-way ANOVA to compare the effects of doxycycline concentrations on the mScarlet-I fluorescence signal and the INSPECT RNA reporter levels.

We found that the decreasing trend of mScarlet-I fluorescence was not statistically significant ($F(5,12) = 2.24$, $P = 0.1172$) but that the accompanying reduction of the secreted RNA reporter levels was significantly different ($F(4,10) = 90.35$, $P < 0.0001$) for 5 and 500 ng ml⁻¹ dox (post-hoc test with Bonferroni multiple-comparisons tests (MCT)) as shown in the following graph, where we regrouped the data of the **old Supplementary Fig. 3** according to the promoter type (dox-inducible (left) vs. constitutive (right)).

We think that this effect is caused by doxycycline-related exhaustion of the cellular transcription and translation capacity due to massive induction of the Dox-dependent transcription at high doxycycline concentrations. Such behavior of strong inducible promoters suppressing the expression of co-expressed genes has been noticed before and termed ‘squenching’ (doi:10.1038/334721a0, doi:10.1038/335683a0, doi:10.1038/s41467-019-14147-5). We hypothesize that by concurrent squenching of Gag-PCP expression, secreted RNA reporter levels are more strongly reduced than the fluorescent signal accumulation inside the cytosol.

To avoid such confounding factors from squenching, we have thus repeated the experiment with Dox concentrations up to 1 ng ml⁻¹ and optimized transfection conditions (jetOPTIMUS vs. XtremeGene).

As can be seen in the **new Extended Data Fig. 3** below, we still observed a significant induction of the msfGFP fluorescence and accompanying increase in the INSPECT RNA reporter abundance (**Extended Data Fig. 3d**).

As in the previous experiment, neither the mScarlet-I fluorescence ($F(3,8) = 0.7020$, $P = 0.5769$) nor the respective RNA reporter ($F(2,6) = 3.021$, $P = 0.1237$) were significantly suppressed for these Dox concentrations. The no-Gag-PCP control showed no secretion of the RNA reporter ($P < 0.0001$, two-tailed unpaired t -test (**Extended Data Fig. 3e**, **Supplementary Table 1**)).

These data thus demonstrate that the Gag-mediated secretion of INSPECT-encoded RNA reporters can be used as an alternative INSPECT reporter module while avoiding undesired

effects of unnecessarily high Dox concentrations.

Extended Data Fig. 3 | INSPECT reporter enables modular read-out of coding genes using RNA reporters.

R1P6:

Supplementary figure 3d-e. What stands out from both these figure is the considerable difference in sensitivity of RT-qPCR detection of the reporter RNA, relative to that of the fluorescent signal from the reporter. This is especially true for conditions where Dox induction was at low concentrations (or no Dox addition). This seems to call into question the sensitivity of INSPECT for transcripts of low expression. (This feeds off the first listed major point).

Response to R1P6:

Yes, it is correct that fluorescence reporters intrinsically have a lower intensity than reporter enzymes such as luciferases or PCR-amplified detection schemes that can be deployed on the RNA reporters.

The INSPECT system is theoretically compatible with any reporter (we have shown 4 classes in the manuscript), which can be chosen with respect to the requirements for the detection modality (plate reader, imaging, etc.) and the expression levels of the gene of interest.

As detailed in our response to your first major point, we have shown that for genes of interest with low expression levels, such as *IL2* at baseline levels or *GUARDIN*, the NLuc luciferase is an attractive choice since it offers high sensitivity, a high dynamic range and can be detected in high-throughput multiwell-plate assays and/or via bioluminescence microscopy with single-cell resolution.

R1P7:

Suggestion for supplementary figure 4b: add the different images of this same cell line before Cre activation, for reference.

Response to R1P7:

We thank the Reviewer for this suggestion. We added smFISH images of the same cell line before Cre activation and provided the corresponding speckle count analysis. (**Extended Data Fig. 7**).

Reviewer #2:

R2P0

Remarks to the Author:

This manuscript presents a very interesting approach to monitor RNA expression through use of reporters expressed via synthetic intron sequences embedded within a gene where the reporter expression is dependent on the gene's promoter. The stated benefit of this approach over other methods of tagging RNA or genes is that the reporter does not alter the coding sequence of the gene of interest.

Response to R2P0:

We thank Reviewer 2 for the positive overall assessment of INSPECT.

R2P1:

-One major issue not described in detail is how the site of insertion of the reporter may influence reporter expression as well as parent gene expression. It seems possible that if the reporter were inserted next to an endogenous splice donor/acceptor site and/or regulatory element for a specific RNA isoform or splicing of the parent gene, that the reporter could influence splicing of the parent gene. Given these concerns, it would be helpful for the authors to (i) explain the rationale for the site of insertion of the reporter in exon 3 of IL2, and (ii) demonstrate that the reporter could be inserted into other sites within the IL2 locus (if that is possible) for similar effect. As a minor point, it would be helpful to modify Figure 2a to show exactly where in the IL2 locus the promoter was inserted.

Response to R2P1:

We thank Reviewer 2 for these detailed suggestions on how to further detail the design considerations of INSPECT constructs.

(i)

Regarding the choice of the insertion site, we have added the following paragraph to the Materials & Methods section.

"We chose the exonic splice consensus sequence NAG|GWW ('|' indicates the insertion site) for CRISPR/Cas9-mediated INSPECT insertion. The resulting flanking exonic sequences should have a minimum length of 30 bp to ensure precise splicing.

For coding genes, INSPECT should not be introduced more than 50 bp downstream of the host CDS to avoid nonsense-mediated decay (doi:10.1016/j.cell.2016.05.053). For long non-coding RNAs, only restraints on minimum exon size apply. A splice prediction software should further

31evaluate chosen splice sites (e.g., NetGene2 or Human Splicing Finder 3) after *in silico* insertion of the INSPECT reporter construct.”

This heuristic was also used to insert the INSPECT_{NLuc} reporter into exon 3 of *IL2*. We have now extended the validation experiments to confirm that the NLuc reporter has the dynamic range to track the strong *IL2* induction and can longitudinally track *IL2* expression faithfully.

(ii)

Since, in the case of *IL2*, alternative insertion sites do not carry additional information on *IL2* expression, we instead inserted INSPECT at two locations in the lncRNA *NEAT1* that can differentiate between the total *NEAT1* (*NEAT1*_{total}, composed of *NEAT1*_{v1+v2}) or exclusively the long isoform (*NEAT1*_{v2}) (**Fig. 3**).

For both reporter variants, we have now extended our validation experiments of the respective INSPECT reporter lines to – in addition to the fraction of cells with paraspeckles – also quantify the number of paraspeckles per cell, the nuclear abundance of *NEAT1* relative to the nuclear lncRNA *MALAT1*, as well as the relative nuclear/cytosolic abundance of *NEAT1* (**Fig. 3e,f,g**). These reporter lines could then faithfully track the modulation of *NEAT1* expression (**Fig. 3h**) and could also be used to observe *NEAT1* expression with single-cell resolution in live cells (**Fig. 3j**). We also analyzed an alternative insertion site for monitoring *NEAT1*_{v2}. While this reporter line showed the same fraction of cells with paraspeckles, our extended validation experiments now showed a slight increase in the number of paraspeckles per cell and an alteration in the nuclear/cytosolic abundance of *NEAT1* (**Extended Data Fig. 6**).

Due to the cryptic splicing, we did not continue with further experiments using this insertion site (**Extended Data Fig. 8**).

We used the same heuristics for the insertion into the lncRNA *GUARDIN* and found no alteration in the splicing behavior.

As with any reporter system, it is advisable to double-check that the processing of the gene of interest is not affected at the RNA or protein level before starting large-scale analyses.

R2P2:

It would be helpful to compare *IL2* protein expression in Fig. 2d to Jurkat cells that have not undergone reporter insertion. This figure should also include formal statistical comparison (I realize the difference between non-activated and activated cells is very clearly different).

Response to R2P2:

As can be seen in the old **Fig. 2d** (now **new Extended Data Fig. 1**), the variability of the IL2 protein induction between clonal cells selected from the parental Jurkat pool is considerable, while there was a linear relationship to the INSPECT reporter signal (here uptake of ^{131}I via the NIS importer). While all clones were homozygous, one clone showed a stronger induction of *IL2* than wild-type Jurkat E6.1 cells, while the two other clones were less inducible.

This behavior is in line with the well-known variable gene expression in Jurkat T cells, both in range and dynamics, which is highly dependent on the specific clonal population and even varies with subclones from originally clonal populations (doi:10.1016/0014-4827(87)90171-6).

The Jurkat E6.1 clone (described in Weiss, A. et al. (1984). The role of T3 Surface Molecules in the Activation of Human T Cells: a Two-Stimulus Requirement for IL-2 Production Reflects Events Occurring at a Pre-translational Level. J. Immunol. 133: 123-128) is derived from the parental Jurkat JM clone and exhibits a particularly high variability (doi:10.1002/ijc.2910190505).

We have now substantially expanded the INSPECT monitoring of *IL2* with a new reporter cell line based on the luciferase NLuc as a reporter enzyme (INSPECT_{IL2:SigP-NLuc}). During the selection process for this reporter line, we again found a substantial variability of the *IL2* induction and we thus carefully screened for a clone with the most similar increase in induction as compared to our parental cells.

33We have then followed the *IL2* trajectory (**new Fig. 2e**) of the selected INSPECT_{*IL2:SigP-NLuc*} line and found that non-induced levels are non-significantly different from the wt Jurkat cells over the entire monitoring window of 24 hours.

After TCR activation, *IL2* induction time courses overlap for the first ~5 hours before the wt Jurkat cells decrease in induction speed slightly relative to INSPECT_{*IL2:SigP-NLuc*} although the overall correlation between the trajectories was still high (Pearson's $r = 0.9997$, $P < 0.0001$; **new Fig. 2e**).

We also used an established ELISA against IL2, but its sensitivity was too weak to measure IL2 secretion without TCR activation and only sufficient to detect secreted IL2 protein accumulated over 16 hours of strong induction, yielding $39.8 \pm 0.5 \text{ ng ml}^{-1}$ and $67 \pm 1.8 \text{ ng ml}^{-1}$. While these values from three technical replicates are significantly different ($P < 0.0001$, unpaired two-tailed *t*-test), the factor of ~2 difference in the accumulated IL2 protein is small compared with the massive induction over orders of magnitude.

We thus moved on with the INSPECT_{*IL2:SigP-NLuc*} line and followed *IL2* decrease after cessation of TCR activation, again showing a high correlation (Pearson's $r = 0.9195$, $P = 0.0095$) to the parental cells (**new Fig. 2f**).

We also used bioluminescence microscopy to monitor milder *IL2* induction over time at single-cell resolution, which allowed us to resolve the variability in the induction time over the population cell, revealing a substantial heterogeneity although all the cells we used are of clonal nature (**new Fig. 2h**).

We thus conclude from these characterization experiments that the selected INSPECT_{*IL2:SigP-NLuc*} demonstrated reasonably representative induction behavior as compared to wildtype Jurkat cells such that it can be a valuable reporter for convenient multiwell screening of e.g., pharmacological modulators of *IL2* expression with high sensitivity and over time.

R2P3:

Are the author's certain that accumulation of 131I- would not affect T-cell function? I realize this technique is being used a method to read out IL2 expression and presented as a proof-of-concept methodology, but if the protein encoded by the reporter affected cell function, then this would be a limitation of the specific approach presented.

Response to R2P3:

NIS is an established reporter in the molecular imaging field for tracking CAR-T cells, and there are real concerns that the tracer is affecting cell function (doi:10.1016/j.omto.2021.03.003).

We now have moved the graphs to the extended data figure section since, as the Reviewer pointed out, the main point was to show as a proof-of-concept that also complex transmembrane

proteins can be used as INSPECT reporter modules and that high sensitivity can also be achieved for an imaging modality that has translational potential (doi:10.1126/scitranslmed.aag2196). Using engineered HSV-Tk enzymes (doi:10.1016/j.omto.2021.03.003) instead of NIS for PET readout would be another feasible option.

R2P4:

The manuscript title could simply state “RNAs” and not “(non)-coding RNAs” (because this latter phrasing is also confusing; I realized after reading the abstract that the authors have used this phrase to attempt to encapsulate both protein-coding as well as non-protein-coding RNAs but this is not clear when reading the manuscript title alone).

Response to R2P4:

Yes, this was our intent. We thank the Reviewer for the suggestions and would leave it to the editor to help guide us on which notation is preferred in the title and whether the expression should be expanded.

Reviewer #3:

R3P0

Remarks to the Author:

The authors have created a new way to track gene expression by encoding translatable elements in an intron they add to the gene. The design approach is overall straightforward and the authors have provided some data to suggest it works to look at endogenous transcripts. However, the authors have provided minimal data so it is hard to judge at this point how robust the system is and if it truly is a minimally invasive approach. I have made some specific suggestions below as to the sorts of data that should be included in a revised manuscript.

Response to R3P0:

We thank the Reviewer for appreciating the overall design of the reporter system.

As detailed in the responses to your respective questions below, we have now added substantial additional characterizations to confirm that the INSPECT reporters can track gene expression of several genes in different cell lines without changing the target mRNA or mature lncRNA.

R3P1:

(1) The authors should be more careful when they claim that INSPECT acts “without altering the target of interest at either the RNA or protein level”. The authors are adding an extra intron to the gene, which at minimum means an extra exon-junction complex has been deposited on the transcript. This is even more relevant when INSPECT is used to study *NEAT1* as *NEAT1* is normally an unspliced RNA. The authors’ technique has now turned *NEAT1* into a spliced RNA which may have unintended effects. Is it possible to add INSPECT into naturally present introns rather than adding a new intron?

Response to R3P1:

We thank the Reviewer for pointing out this phrasing in the abstract. In other paragraphs, we more correctly write that INSPECT is designed not to alter the mature mRNA and thus the protein.

To capture the more general case in the abstract, we now changed the respective phrase to “without altering the mature lncRNA or mRNA of the target of interest.”

We have now provided additional validation experiments for *NEAT1*, which we will also detail in the responses to your specific questions below.

To experimentally show that the localization of the unspliced nuclear lncRNA *NEAT1* is not changed by the insertion of INSPECT and the resulting formation of an exon-exon-junction complex at the splice site, we obtained the nuclear/cytoplasmic ratio of *NEAT1* abundance for

wild-type HEK293T and the INSPECT cell lines (**Fig. 3g**). In addition, we confirmed that the insertion of INSPECT did not alter the nuclear abundance of *NEAT1* in comparison to unspliced wild-type *NEAT1* (**Fig. 3f**). Moreover, a more detailed analysis of the paraspeckle count per cell was not significantly different from unmodified WT cells (**Fig. 3e**). We also validated by RT-PCR followed by Sanger sequencing to ensure that the mature transcripts were unchanged from wild-type (**Extended Data Fig. 5**).

Although INSPECT insertion into a natural intron is certainly also possible, we chose the more challenging case to show the generalizability of INSPECT to other lncRNAs by choosing the low-abundance lncRNA *GUARDIN* as an additional target.

GUARDIN is a p53-responsive lncRNA involved in maintaining genomic integrity (RP3-510D11.2-1, doi:10.1038/s41556-018-0066-7), which has about a magnitude lower abundance as compared with *NEAT1_v2* in HEK293T cells (**Supplementary Fig 2**).

We showed that INSPECT insertion into *GUARDIN* did not modify the native expression level (RT-qPCR, **Extended Data Fig. 4d**) nor the RNA sequence (RT-PCR followed by Sanger sequencing, **Extended Data Fig. 4b,c**). We then also showed via CRISPRi that the NLuc reporter signal in the supernatant we obtained from INSPECT_{*GUARDIN*:SigP-NLuc} is truly driven by the endogenous *GUARDIN* promoter (**Extended Data Fig. 4e**).

In theory, it would be possible to add these elements into a native constitutive intron without *de novo* introduction of a new splice acceptor and donor. Since different introns have different splicing kinetics and the export of the intron is subsequently mediated by the RNA-encoded nuclear export elements, the export might be faster than some slow-splicing introns and, thus, inadvertently lead to the export of unspliced transcript. Therefore, to ensure fast and correct splicing, we preferred to insert *de novo* splice sites independent if the gene already bears native introns.

R3P2:

(2) Fig 1c: The exact location of the Cre based elements relative to the rest of the construct elements is somewhat ambiguous based on the writing. The authors should combine the drawings in Fig 1b/1c so that this is much clearer to the reader. It is also unclear at this point in the main text why the authors want the INSPECT method to have the potential to be like a gene trap as the introduction stresses the desire to make a “minimally invasive reporter system”, not any sort of gene knockout.

Response to R3P2:

We thank Reviewer 3 for the request to illustrate better that the Cre-recombinase-inducible KO element is located upstream of the IRES and flanked by *loxP* sites. The KO element itself is silent because the polyadenylation sites and the splice acceptor are not oriented in sense direction. Upon transient delivery of Cre. *e.g.*, via plasmid transfection, the KO element is flipped into its sense direction to terminate the transcribed RNA prematurely. We have also added the description “Embedded Cre-inducible off-switch” to **Fig. 1c** to highlight the genetic elements for the Cre-inducible off-switch.

Incorporating the Cre-inducible KO within the INSPECT cassette enables the experimenter to create a reporter cell line and a KO cell line with a single knock-in.

This feature is entirely optional and not contradictory to the minimally invasive monitoring of gene expression. We, however, think that it can be a useful add-on functionality for complementing promoter-dependent interrogation with tissue-specific or developmental-stage-dependent KOs of coding and non-coding genes.

Suppose an experimenter likes to use Cre for controlling a different gene. In that case, the feature can be quickly removed by a simple *in vitro* restriction digestion of the INSPECT donor plasmid, followed by ligation.

R3P3:

(3) Figure 2: The authors need to more completely demonstrate that insertion of the INSPECT element does not alter IL-2 transcription/RNA processing, etc. An ELISA is shown in Fig 2d, but such data do not address if the IL-2 RNA levels have been affected or if the dynamics of transcription induction have been changed. One could imagine that transcription induction might be slower as the gene length has been increased. It is also possible that some amount of the IL2 transcript is not fully spliced.

Response to R3P3:

We have now added RT-qPCR data comparing response time and RNA levels of *IL2* from parental Jurkat E6.1 and INSPECT_{*IL2*:SigP-NLuc} after T cell induction with 0.5 $\mu\text{g ml}^{-1}$ PHA and 0.5 ng μl^{-1} PMA (**Fig. 2e**).

We show that the exon 3 of *IL2*, which we split into two new exons by the introduction of INSPECT, was functionally reconstituted since RT-qPCR which span this corresponding region, gave comparable quantities of RNA compared to wild-type and INSPECT_{*IL2*:SigP-NLuc} Jurkat E6.1 (**Fig. 2d**). Correct splicing of the *IL2* mRNA was also verified via RT-PCR using primers spanning the full-length *IL2* transcript and subsequent Sanger analysis (**Fig. 2b,c**). In addition, the sandwich ELISA revealed similar IL-2 secretion after TCR activation in unmodified and INSPECT_{*IL2*:SigP-NLuc}

cells ($39.8 \pm 0.5 \text{ ng ml}^{-1}$ for WT and $67 \pm 1.8 \text{ ng ml}^{-1}$ for INSPECT_{IL2:SigP-NLuc} (**Supplementary Table 1**)).

R3P4:

(4) Key details of the INSPECT intron sometimes change between panels, especially the reporter gene that is translated from the intron (NLuc-PEST in Fig 1, NIS in Fig 2, SecNLuc in Fig 3, mScarlet in Fig S2, etc). It is not clear why the authors have used such a variety of reporters across panels. Perhaps it is because the system is very robust and any combination works, or it could be that the system is hard to make work well and only certain designs work in certain contexts. It is not possible to judge this important point based on the data currently presented.

Response to R3P4:

The luciferase NLuc is the main INSPECT reporter module that works robustly for all three genes that we assessed (*IL2*, *NEAT1*, *GUARDIN*) both in multi-well bulk bioluminescence measurements with high sensitivity (**Fig. 2d**, **Fig 3h**, **Extended Data Fig. 4e**) and dynamic range as well as in bioluminescence microscopy providing single-cell resolution (**Fig. 2g,h**, and **Fig. 3j**). As the Reviewer alluded to, we also found it essential to demonstrate in this ‘Technical Report’ that INSPECT is versatile in allowing for different reporter modules, given the different experimental needs and specific advantages and disadvantages of the different detection and imaging methods.

We thus also show data for the transmembrane sodium iodide symporter (NIS) (**Extended Data Fig. 1**), fluorescence proteins (**Extended Data Fig. 2**), and secretion of RNA barcodes, which we present in a separate section entitled “Alternative INSPECT reporter modules” (**Extended Data Fig. 3**).

We have also added detailed technical notes on the different features of the respective reporter modules in the figure legends and ‘Material & Methods’ section.

R3P5:

(5) The use of gag-PCP to detect INSPECT transcripts does not seem to be much better than current aptamer based approaches. Yes, a different transcript than the mRNA/ncRNA of interest is being detected, but you still have to change the cells quite a bit by putting gag-PCP into them.

Response to R3P5:

The Reviewer is correct in that a second genetic component for Gag-mediated export has to be introduced into this variant of the INSPECT reporter system, similar to an aptamer system.

However, the main differentiating feature is the increased multiplexing capacity by using unique RNA sequences of a secreted RNA reporter instead of secreted luciferases (or fluorescent proteins) while still allowing for convenient repeated sampling of the supernatant. These capabilities are different from those of existing tagging system by inserting aptamer into the lncRNA or the 3'-UTR of a coding gene, while a fluorescent protein fused to an aptamer-binding protein is constitutively expressed, allowing localization of the RNA of interest, which INSPECT is not designed to do. Moreover, direct modification of the target RNA with aptamers may influence the half-life of the target RNA and thus, can be considered to be unfortunately invasive (doi:10.1038/nmeth.4502).

With respect to the effects of Gag-PCP on cell physiology, we confirmed that there was no effect on cell viability (**Extended data Fig. 3c**). In a more extensive version of the reporter module, Gag-PCP may also be integrated into a safe harbor locus and its expression rendered inducible via doxycyclin or a Tet-On 3G system.

R3P6:

(6) Fig 3: To echo my points from #3 above, the authors need to more closely look at NEAT1 transcript levels in cells and whether INSPECT has changed NEAT1 at the RNA level in any way. They have only measured the number of cells with paraspeckles. They have not measured whether NEAT1 levels or processing is at all altered by the addition of the INSPECT elements. In Fig 3e, if INSPECT elements are not present, what do endogenous NEAT1 levels look like after CRISPRi? Same level of knockdown as when INSPECT elements are present? These are the sorts of important controls that are lacking in the current manuscript.

Response to R3P6:

As also stated in our response to your point R3P1 above, we have now added additional data to show that INSPECT does not perturb the native behavior of the parental transcripts. We refined the figures and added data supporting that the subcellular localization of NEAT1 is not changed after INSPECT insertion. We analyzed the *NEAT1* level of the nucleus and cytoplasm via RT-qPCR after subcellular fractionation (**Fig. 3f,g**). To verify efficient splicing, we performed an RT-PCR and verified the sequence integrity at the insertion site by Sanger sequencing (**Extended Data Fig. 5**). As requested, we also added RT-qPCR data for the CRISPRi of *NEAT1* in wild-type HEK293T. (**Fig. 3h,i**).

R3P7 & R3P8:

(8) The points in the Discussion are overstated. For example, the authors discuss how INSPECT allows for “high-throughput”, “multi-time point assessment”, and “multiplexing” but the authors have not provided data proving any of these points in the current manuscript.

Response to R3P7 & R3P8:

We had written the respective part of the Discussion with the intention to provide an outlook on which experiments are technically feasible based on the specifications of INSPECT. We now provide multi-time point data for monitoring IL2 expression in a multi-well format in the new **Fig. 2e,f** and also with single-cell and multi-time point resolution in **Figure 2 h,i**.

As compared with the labor- and time-intensive smFISH stainings against *NEAT1*, the multi-well luminescence detection of *NEAT1* is intrinsically easier to scale up to higher throughputs and can also be measured non-destructively in live cells (**Fig. 3j**).

However, since “high-throughput assessment” is a phrase that can maybe be understood with a variable “scale” in mind, we now write “multi-well assessment” instead.

“As compared with consumptive sequence analyses or staining techniques, INSPECT can allow for multi-time point and convenient multi-well assessment of modulators of lncRNA expression for correlation with resulting phenotypes⁴⁹.”

We used the term “multiplexing” in the section on the RNA reporter module to point out that the different modular reporter proteins we implemented (Nluc, NIS, GFP) can provide different signal channels that can be expanded by using RNA as opposed to proteins as reporters.

To make it clearer that we meant an extension of the present work we now made the following edit in the wording.

“Analysis of secreted INSPECT transcripts by RT-qPCR as opposed to a reporter protein signal can **replaced :[further extend the] with** enable multiplexing capabilities of INSPECT.”

R3P9:

(9) The full, exact sequences of the complete INSPECT cassettes need to be included in the methods section so that this work can be replicated in the future by any interested party.

Response to R3P9:

We have now included all INSPECT sequences in **Supplementary Table 2**.

We will also submit the cloning plasmids for the INSPECT modules with full documentation to Addgene.

Minor points:

R3P10:

(1) Yellow is very difficult to see in Fig 2d.

Response to R3P10:

We have now moved this figure to **Extended Data Fig. 1**. We changed the yellow color to pink, which is more perceivable.

R3P11:

(2) P. 6: The authors write that INSPECT “offers the possibility to create a full knockout of the tagged gene” but it is not really a full knockout but instead the creation of a shorten, prematurely terminated transcript.

Response to R3P11:

As the Reviewer writes above, the method is more akin to a gene trap. We used the term “knockout” in the broadest sense, e.g., “A knockout, as related to genomics, refers to the use of genetic engineering to inactivate or remove one or more specific genes from an organism.” (National Human Genome Research Institute)

Targeted introductions of a polyA signal have already been used to inactive genes (doi:10.1186/s12575-018-0086-5, doi:10.1261/rna.069484.118). Moreover, gene and also promoter traps, which inactivate a trapped gene via premature transcriptional termination, have been used routinely to study genotype-phenotype correlations (doi:10.1186/s12575-018-0086-5). We have now deleted the term “full knockout” and instead wrote: “the reporter system also features an optional Cre-recombinase-dependent switch that renders the target gene inoperative, *i.e.*, effectively generates a knockout (KO).”

Decision Letter, first revision:

Subject: Your manuscript, NCB-W45968B

Message:

42Our ref: NCB-W45968B

11th July 2022

Dear Dr. Westmeyer,

Thank you for submitting your revised manuscript "Intron-encoded cistronic transcripts for minimally-invasive monitoring of (non-)coding RNAs" (NCB-W45968B). It has now been seen by the original referees and their comments are below. The reviewers find that the paper has improved in revision, and therefore we'll be happy in principle to publish it in Nature Cell Biology, pending minor revisions to satisfy the referees' final requests and to comply with our editorial and formatting guidelines.

Since the current version of your manuscript is in a PDF format, please email us a copy of the file in an editable format (Microsoft Word or LaTeX)-- we can not proceed with PDFs at this stage.

We are now performing detailed checks on your paper and will send you a checklist detailing our editorial and formatting requirements in about a week. Please do not upload the final materials and please not yet make any revisions until you receive this additional information from us.

Thank you again for your interest in Nature Cell Biology Please do not hesitate to contact me if you have any questions.

Sincerely,

Daryl J.V. David, PhD

Senior Editor, Nature Cell Biology
Consulting Editor, Nature Communications
Nature Portfolio

Heidelberger Platz 3, 14197 Berlin, Germany
Email: daryl.david@nature.com
ORCID: <https://orcid.org/0000-0002-9253-4805>

43Reviewer #1 (Remarks to the Author):

The authors have substantially improved the manuscript, and the description of this important and powerful method is now better supported by data. I can therefore recommend publication, once the authors address these small points:

Figure 1e: It seems from the raw data provided that the comparison in the top and bottom panel are between induced and non-induced INSPECT cell lines. I would clarify this both in the figure legend ('luminescence signal ... from INSPECT Jurkat or unmodified Jurkat' whereas the raw data suggests it is between induced and non-induced.) as well as in the figure (e.g. replace 'control' with 'non-induced')

Figure 1f: Clarify in the legend what the r value presented on the top of the panels is.

Reviewer #2 (Remarks to the Author):

The authors have replied to my initial comments and questions well. I have no further issues with the manuscript.

Reviewer #4 (Remarks to the Author):

The authors have done a nice job and have largely addressed my prior concerns. I have a few remaining polishing comments for the text.

(1) In the abstract, the authors write that INSPECT results “in unmodified mRNA” and acts “without altering the mature lncRNA or mRNA of the target of interest”. To be most accurate, the authors should clarify that the “mature RNA sequence” is unaltered. There should be an additional exon-junction complex on the RNA due to insertion of INSPECT, so the RNP (which is the form in which all RNAs function) is altered.

(2) P.2: The following phrase is confusing: “mimicking viral motifs to trigger nuclear RNA export instead of intron lariat formation and degradation.” Intron lariats are still being formed with the INSPECT system and I assume it is the intron lariats that are being exported to the cytoplasm, but the manuscript

formally does not show whether it is a linear or circular form that is exported. Please clarify the language here and comment on the form of RNA that is exported.

(3) P.2: Please provide reference for “3’ stabilizing element (triple helix)”. None has been provided in text or figure legend.

(4) Extended Data Fig 6 is an important example of the need for validating that insertion of the INSPECT elements does not alter the endogenous gene. Given that this is a methods paper, I feel it would be very appropriate in the discussion section to highlight the potential pitfalls/shortcomings of the system so that naïve users do not fall into such traps.

(5) Methods section is very long. I defer to Editor on journal policy, but it seems like the authors can move some of the very common methods (e.g. DNA digestion, agarose gel running, etc) into a supplemental file.

Decision Letter, final requests:

Subject: NCB: Your manuscript, NCB-W45968B

Message:

Our ref: NCB-W45968B

26th July 2022

Dear Dr. Westmeyer,

Thank you for your patience as we’ve prepared the guidelines for final submission of your Nature Cell Biology manuscript, "Intron-encoded cistronic transcripts for minimally-invasive monitoring of (non-)coding RNAs" (NCB-W45968B). Please carefully follow the step-by-step instructions provided in the attached file, and add a response in each row of the table to indicate the changes that you have made. Please also check and comment on any additional marked-up edits we have proposed within the text. Ensuring that each point is addressed will help to ensure that your revised manuscript can be swiftly handed over to our production team.

We would like to start working on your revised paper, with all of the requested files and forms, as soon as possible (preferably within one week). Please get in contact with us if you anticipate delays.

45If you have not done so already, please alert us to any related manuscripts from your group that are under consideration or in press at other journals, or are being written up for submission to other journals (see: <https://www.nature.com/nature-research/editorial-policies/plagiarism#policy-on-duplicate-publication> for details).

In recognition of the time and expertise our reviewers provide to Nature Cell Biology's editorial process, we would like to formally acknowledge their contribution to the external peer review of your manuscript entitled "Intron-encoded cistronic transcripts for minimally-invasive monitoring of (non-)coding RNAs". For those reviewers who give their assent, we will be publishing their names alongside the published article.

Nature Cell Biology offers a Transparent Peer Review option for new original research manuscripts submitted after December 1st, 2019. As part of this initiative, we encourage our authors to support increased transparency into the peer review process by agreeing to have the reviewer comments, author rebuttal letters, and editorial decision letters published as a Supplementary item. When you submit your final files please clearly state in your cover letter whether or not you would like to participate in this initiative. Please note that failure to state your preference will result in delays in accepting your manuscript for publication.

Cover suggestions

As you prepare your final files we encourage you to consider whether you have any images or illustrations that may be appropriate for use on the cover of Nature Cell Biology.

Nature Cell Biology has now transitioned to a unified Rights Collection system which will allow our Author Services team to quickly and easily collect the rights and permissions required to publish your work. Approximately 10 days after your paper is formally accepted, you will receive an email in providing you with a link to complete the grant of rights. If your paper is eligible for Open Access, our Author Services team will also be in touch regarding any additional information that may be required to arrange payment for your article.

Please note that Nature Cell Biology is a Transformative Journal (TJ). Authors may publish their research with us through the traditional subscription access route or make their paper immediately open access through payment of an article-processing charge (APC). Authors will not be required to make a final decision about access to their article until it has been accepted. Find out more about Transformative Journals

Authors may need to take specific actions to achieve compliance with funder and institutional open access mandates. If your research is supported by a funder that requires immediate open access (e.g. according to Plan S principles) then you should select the gold OA route, and we will direct you to the compliant route where possible. For authors selecting the subscription publication route, the journal's standard licensing terms will need to be accepted, including self-archiving policies. Those licensing terms will supersede any other terms that the author or any third party may assert apply to any version of the manuscript.

For information regarding our different publishing models please see our Transformative Journals page. If you have any questions about costs, Open Access requirements, or our legal forms, please contact ASJournals@springernature.com.

47[REDACTED]

Best regards,

Nyx Hills
Staff
Nature Cell Biology

On behalf of

Daryl J.V. David, PhD

Senior Editor, Nature Cell Biology
Consulting Editor, Nature Communications
Nature Portfolio

Heidelberger Platz 3, 14197 Berlin, Germany
Email: daryl.david@nature.com
ORCID: <https://orcid.org/0000-0002-9253-4805>

Reviewer #1:

Remarks to the Author:

The authors have substantially improved the manuscript, and the description of this important and powerful method is now better supported by data. I can therefore recommend publication, once the authors address these small points:

Figure 1e: It seems from the raw data provided that the comparison in the top and bottom panel are between induced and non-induced INSPECT cell lines. I would clarify this both in the figure legend ('luminescence signal ... from INSPECT Jurkat or unmodified Jurkat' whereas the raw data suggests it is between induced and non-induced.) as well as in the figure (e.g. replace 'control' with 'non-induced')

48Figure 1f: Clarify in the legend what the r value presented on the top of the panels is.

Reviewer #2:

Remarks to the Author:

The authors have replied to my initial comments and questions well. I have no further issues with the manuscript.

Reviewer #4:

Remarks to the Author:

The authors have done a nice job and have largely addressed my prior concerns. I have a few remaining polishing comments for the text.

(1) In the abstract, the authors write that INSPECT results “in unmodified mRNA” and acts “without altering the mature lncRNA or mRNA of the target of interest”. To be most accurate, the authors should clarify that the “mature RNA sequence” is unaltered. There should be an additional exon-junction complex on the RNA due to insertion of INSPECT, so the RNP (which is the form in which all RNAs function) is altered.

(2) P.2: The following phrase is confusing: “mimicking viral motifs to trigger nuclear RNA export instead of intron lariat formation and degradation.” Intron lariats are still being formed with the INSPECT system and I assume it is the intron lariats that are being exported to the cytoplasm, but the manuscript formally does not show whether it is a linear or circular form that is exported. Please clarify the language here and comment on the form of RNA that is exported.

(3) P.2: Please provide reference for “3’ stabilizing element (triple helix)”. None has been provided in text or figure legend.

(4) Extended Data Fig 6 is an important example of the need for validating that insertion of the INSPECT elements does not alter the endogenous gene. Given that this is a methods paper, I feel it would be very appropriate in the discussion section to highlight the potential pitfalls/shortcomings of the system so that naïve users do not fall into such traps.

49(5) Methods section is very long. I defer to Editor on journal policy, but it seems like the authors can move some of the very common methods (e.g. DNA digestion, agarose gel running, etc) into a supplemental file.

Author Rebuttal, first revision:

Responses to Reviewers (2nd revision)

General remark to all Reviewers.

We thank all reviewers again for their very valuable comments, which we address point-by-point below.

Since we were asked to rearrange the figures quite substantially, also we generated the look-up table below for revision 1.

Old Figure	New Figure
Main Figures	
Old Fig. 1	New Fig. 1
Old Extended Data Fig. 2	New Fig. 2
Old Extended Data Fig. 3	New Fig. 3
Old Fig. 2	New Fig. 4
Old Extended Data Fig. 4	New Fig. 5
Old Fig. 3	New Fig. 6
Extended Data Figures	

Old Supplementary Fig. 1	New Extended Data Fig. 1
Old Extended Data Fig. 1	New Extended Data Fig. 2
Old Supplementary Fig. 2	New Extended Data Fig. 3
Old Extended Data Fig. 5	New Extended Data Fig. 4
Old Extended Data Fig. 6	New Extended Data Fig. 5
Old Extended Data Fig. 7	New Extended Data Fig. 6
Old Extended Data Fig. 8	New Extended Data Fig. 7
Old Supplementary Fig. 3	New Extended Data Fig. 8
Old Supplementary Fig. 4	New Extended Data Fig. 9
Old Supplementary Fig. 5	New Extended Data Fig. 10
Supplementary Figures	
Old Supplementary Fig. 5	New Supplementary Fig. 1
Old Supplementary Fig. 6	Source Data

Reviewer #1 (Remarks to the Author):

The authors have substantially improved the manuscript, and the description of this important and powerful method is now better supported by data. I can therefore recommend publication, once the authors address these small points:

51Figure 1e: It seems from the raw data provided that the comparison in the top and bottom panel are between induced and non-induced INSPECT cell lines. I would clarify this both in the figure legend ('luminescence signal ... from INSPECT Jurkat or unmodified Jurkat' whereas the raw data suggests it is between induced and non-induced.) as well as in the figure (e.g. replace 'control' with 'non-induced').

Figure 1f: Clarify in the legend what the r value presented on the top of the panels is.

R1P0_rev2

We thank Reviewer #1 for the constructive comments that helped us improve our manuscript substantially.

We clarified the caption by modifying the following sentence for Fig 4e:

“Corresponding mRNA levels of *IL2* (middle) or INSPECT (bottom) were quantified by RT-qPCR and normalized to ACTB expression strength. ** $P < 0.01$; *** $P < 0.001$; **** $P < 0.0001$ (two-tailed, unpaired t-test for induced vs control condition after 4 hours of induction)”

We clarified r in the caption of Fig 4e: “ r indicates the Pearson correlation coefficient. Full statistical results are given in Supplementary Table 1.”

Reviewer #2 (Remarks to the Author):

The authors have replied to my initial comments and questions well. I have no further issues with the manuscript.

R2P0_rev2

We thank Reviewer #2 for the helpful comments during the first revision.

Reviewer #3 (Remarks to the Author):

The authors have done a nice job and have largely addressed my prior concerns. I have a few remaining polishing comments for the text.

(1) In the abstract, the authors write that INSPECT results “in unmodified mRNA” and acts “without altering the mature lncRNA or mRNA of the target of interest”. To be most accurate, the authors should clarify that the “mature RNA sequence” is unaltered. There should be an additional exon-junction complex on the RNA due to insertion of INSPECT, so the RNP (which is the form in which all RNAs function) is altered.

(2) P.2: The following phrase is confusing: “mimicking viral motifs to trigger nuclear RNA export instead of intron lariat formation and degradation.” Intron lariats are still being formed with the INSPECT system and I assume it is the intron lariats that are being exported to the cytoplasm, but the manuscript formally does not show whether it is a linear or circular form that is exported. Please clarify the language here and comment on the form of RNA that is exported.

(3) P.2: Please provide reference for “3’ stabilizing element (triple helix)”. None has been provided in text or figure legend.

(4) Extended Data Fig 6 is an important example of the need for validating that insertion of the INSPECT elements does not alter the endogenous gene. Given that this is a methods paper, I feel it would be very appropriate in the discussion section to highlight the potential pitfalls/shortcomings of the system so that naïve users do not fall into such traps.

(5) Methods section is very long. I defer to Editor on journal policy, but it seems like the authors can move some of the very common methods (e.g. DNA digestion, agarose gel running, etc) into a supplemental file.

R3P0_rev2

We thank the Reviewer for the detailed comments on how to improve the text.

(1) We followed the suggestion and changed the respective sentence in the abstract to: “Post-transcriptional excision of INSPECT results in the mature endogenous RNA without sequence alterations and an additional engineered transcript that leaves the nucleus by hijacking the nuclear export machinery for subsequent translation into a reporter or effector protein.”

(2) We deleted the words “lariat formation and” since we did not determine specifically if the exported intron is circular or already debranched (linear).

(3) We apologize that we did not give a reference for the Malat1 triple helix. We have now added the following two references:

- Wilusz, J.E., JnBaptiste, C.K., Lu, L.Y., Kuhn, C.D., Joshua-Tor, L. and Sharp, P.A., 2012. A triple helix stabilizes the 3’ ends of long noncoding RNAs that lack poly (A) tails. *Genes & development*, 26(21), pp.2392-2407.
- Brown, J.A., Valenstein, M.L., Yario, T.A., Tycowski, K.T. and Steitz, J.A., 2012. Formation of triple-helical structures by the 3’-end sequences of MALAT1 and MEN β noncoding RNAs. *Proceedings of the National Academy of Sciences*, 109(47), pp.19202-19207.

(4) We already provided a section in the Materials & Methods clarifying optimal design: 'Design considerations for INSPECT constructs'. We have now extended the paragraph to allude to potential pitfalls and how to check/validate the host RNA's integrity. We have also added a similar section in the discussion, recommending testing multiple insertion sites.

(5) We thank R#3 for the suggestion to shorten the M&M section and would leave it to the editor to help guide us on this matter.

Final Decision Letter:

Subject: Decision on Nature Cell Biology submission NCB-W45968C
Message:

Dear Dr Westmeyer,

I am pleased to inform you that your manuscript, "Intron-encoded cistronic transcripts for minimally-invasive monitoring of coding and non-coding RNAs", has now been accepted for publication in Nature Cell Biology.

Thank you for sending us the final manuscript files to be processed for print and online production, and for returning the manuscript checklists and other forms. Thank you for providing both the "Tracked changes" and "cleaned" version of your manuscript text file - please do note that we have verified and will be using only your "cleaned" Article text file.

Your manuscript will now be passed to our production team who will be in contact with you if there are any questions with the production quality of supplied figures and text.

Please note that Nature Cell Biology is a Transformative Journal (TJ). Authors may publish their research with us through the traditional subscription access route or make their paper immediately open access through payment of an article-processing charge (APC). Authors will not be required to make a final decision about access to their article until it has been accepted. Find out more about Transformative Journals

Authors may need to take specific actions to achieve compliance with funder and institutional open access mandates. If your research is supported by a funder that requires immediate open access (e.g. according to Plan S principles) then you should select the gold OA route, and we will direct you to the compliant route where possible. For authors selecting the subscription publication route, the journal's standard licensing terms will need to be accepted, including self-archiving policies. Those licensing terms will supersede any other terms that the author or any third party may assert apply to any version of the manuscript.

If you have not already done so, we strongly recommend that you upload the step-by-step protocols used in this manuscript to the Protocol Exchange (www.nature.com/protocolexchange), an open online resource established by Nature Protocols that allows researchers to share their detailed experimental know-how. All uploaded protocols are made freely available, assigned DOIs for ease of citation and are fully searchable through nature.com. Protocols and Nature Portfolio journal papers in which they are used can be linked to one another, and this link is clearly and prominently visible in the online versions of both papers. Authors who performed the specific experiments can act as primary authors for the Protocol as they will be best placed to share the methodology details, but the Corresponding Author of the present research paper should be included as one of the authors. By uploading your Protocols to Protocol Exchange, you are enabling researchers to more readily reproduce or adapt the methodology you use, as well as increasing the visibility of your protocols and papers. You can also establish a dedicated page to collect your lab Protocols. Further information can be found at www.nature.com/protocolexchange/about

With kind regards,
Daryl

Daryl J.V. David, PhD

Senior Editor, Nature Cell Biology
Consulting Editor, Nature Communications
Nature Portfolio

Heidelberger Platz 3, 14197 Berlin, Germany
Email: daryl.david@nature.com
ORCID: <https://orcid.org/0000-0002-9253-4805>